# Loss of WIPI4 in neurodegeneration causes autophagy-independent ferroptosis

Ye Zhu[1,2,4], Motoki Fujimaki[1,2,4], Louisa Snape[3], Ana Lopez[1,2,3],
Angeleen Fleming [1,2,3] & David C. Rubinsztein [1,2] ✉

β-Propeller protein-associated neurodegeneration (BPAN) is a rare
X-linked dominant disease, one of several conditions that manifest with
neurodegeneration and brain iron accumulation. Mutations in the *WD repeat
domain 45* (*WDR45*) gene encoding WIPI4 lead to loss of function in BPAN
but the cellular mechanisms of how these trigger pathology are unclear. The
prevailing view in the literature is that BPAN is simply the consequence of
autophagy deficiency given that WIPI4 functions in this degradation pathway.
However, our data indicate that WIPI4 depletion causes ferroptosis—a type
of cell death induced by lipid peroxidation—via an autophagy-independent
mechanism, as demonstrated both in cell culture and in zebrafish. WIPI4
depletion increases ATG2A localization at endoplasmic reticulum–
mitochondrial contact sites, which enhances phosphatidylserine import
into mitochondria. This results in increased mitochondrial synthesis of
phosphatidylethanolamine, a major lipid prone to peroxidation, thus
enabling ferroptosis. This mechanism has minimal overlap with classical
ferroptosis stimuli but provides insights into the causes of neurodegeneration
in BPAN and may provide clues for therapeutic strategies.

β-Propeller protein-associated neurodegeneration (BPAN) is a rare
X-linked dominant disease that predominantly affects females. It is one
of several diseases that manifest with neurodegeneration and brain iron
accumulation. The gene that is mutated leading to loss of function in
BPAN is *WDR45*, encoding WIPI4 (refs. 1,2). Given that WIPI4 is involved
in autophagy, it has been assumed that BPAN pathology is primarily due
to incapacitation of this intracellular clearance pathway[3,4].

## Results

### WIPI4 loss of function induces ferroptosis
Given that mutations of WIPI4 result in reduced stability of the protein
in patients with BPAN[1], we first depleted WIPI4 using small interfering
RNA (siRNA) targeting *WD repeat domain 45* (*WDR45*) to mimic the
disease condition in cultured cancer cell lines. Cell death, as measured
by lactate dehydrogenase (LDH) release, was increased in SH-SY5Y
neuroblastoma cells depleted of WIPI4 (Fig. 1a and Extended Data
Fig. 1a). Similar changes in cell viability were seen with another siRNA

oligonucleotide targeting WIPI4 in HeLa (Extended Data Fig. 1b) and
SH-SY5Y cells (Extended Data Fig. 1c).

This increased cell death was abrogated by ferroptosis inhibi-
tors (liproxstatin-1 and ferrostatin-1, Fer-1) but not by inhibitors of
apoptosis (Z-VAD-fmk) or necrosis (necrostatin-1; Fig. 1a). In addition,
the viability of both SH-SY5Y (Fig. 1b) and HeLa cells (Extended Data
Fig. 1d) was decreased by WIPI4 knockdown, which could only be res-
cued by ferroptosis inhibitors. WIPI4 knockdown did not increase the
levels of cleaved caspase 3, a marker of apoptosis induction (Extended
Data Fig. 1e). Transient clustered regularly interspaced short palin-
dromic repeats (CRISPR) knockout (KO) of *WDR45* in HeLa cells also
increased cell death, which was rescued by the iron chelator deferoxam-
ine and the reactive oxygen species (ROS) scavenger *N*-acetyl-ʟ-cysteine
(Fig. 1c and Extended Data Fig. 1f, which illustrates KO efficiency).

Ferroptosis is characterized by increased cell death and lipid per-
oxidation. Lipid peroxidation is usually initiated by the abstraction of
hydrogen atoms from methylene carbons in polyunsaturated fatty

[1]Department of Medical Genetics, University of Cambridge, Cambridge Institute for Medical Research, Cambridge, UK. [2]UK Dementia Research Institute,
University of Cambridge, Cambridge Institute for Medical Research, Cambridge, UK. [3]Department of Physiology, Development and Neuroscience,
University of Cambridge, Cambridge, UK. [4]These authors contributed equally: Ye Zhu, Motoki Fujimaki. ✉e-mail: dcr1000@cam.ac.uk

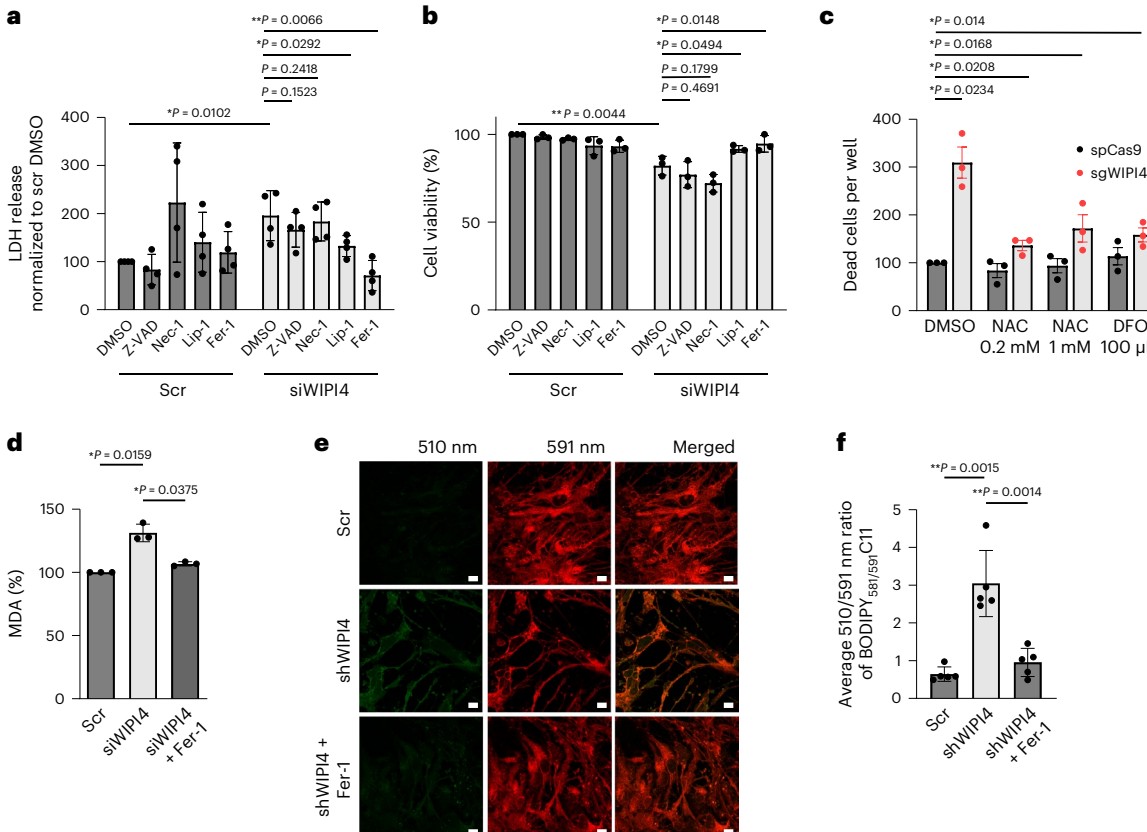

**Fig. 1 | WIPI4 loss of function induces ferroptosis. a**, LDH release, as a measure of cytotoxicity (*n* = 5), from control and WIPI4-knockdown SH-SY5Y cells following treatment with dimethylsulfoxide (DMSO; control), 1 μM Z-VAD-fmk (Z-VAD), 10 μM necrostatin-1 (Nec-1), 100 nM liproxstatin-1 (Lip-1) or 1 μM Fer-1 for 48 h. Unless specified otherwise, oligonucleotide J-019758 was used as the siWIPI4 in all experiments. **b**, Cell viability, measured using AquaBluer (*n* = 3), of control and WIPI4-knockdown SH-SY5Y cells treated as in **a. c**, HeLa cells transiently transfected with pSpCas9(BB)-2A-GFP with single guide RNA (sgRNA) targeting WIPI4 exon 3 (sgWIPI4) or its control construct were cultured for 24 h before treatment with *N*-acetyl-ʟ-cysteine (NAC) or deferoxamine (DFO), at the indicated concentrations, in media containing IncuCyte red dye. The cells were imaged 72 h post transfection. The number of dead cells were determined as the number of red objects per well normalized to that of the scramble and spCas9

control construct-transfected cells treated with DMSO (*n* = 3). **d**, Levels of MDA of control and WIPI4-knockdown SH-SY5Y cells treated with DMSO or 1 μM Fer-1 for 24 h (*n* = 3). **e**, Images of BODIPY$_{581/591}$C11-stained i³ neurons. On Day 14 of differentiation, the i³ neurons were treated with scramble and anti-human WIPI4 smartpool shRNA (shWIPI4) delivered by PLKO.1 packaged in lentivirus for 24 h, followed by treatment with 5 μM Fer-1 or DMSO in fresh media without virus for another 24 h before imaging. Scale bars, 10 μm. **f**, Average ratio of the intensities of the 510 and 591 nm emissions quantified from the images in **e** (*n* = 5; ≥5 fields imaged per experiment). **a**–**d**,**f**, Data are the normalized mean ± s.d.; two-tailed one-sample Student's *t*-test for comparisons with the control and two-tailed paired Student's *t*-test for comparisons with other samples (**a**–**d**); two-tailed paired Student's *t*-test (**f**); *P < 0.05 and **P < 0.01. Scr, scramble; *n*, number of biologically independent experiments. Source numerical data are provided.

---

acids (PUFAs)[5]. Lipid peroxidation is sufficient to cause membrane damage, without the involvement of any specific pore-forming proteins, and leads to cell death[5]. The generation of secondary product aldehydes, including malondialdehyde (MDA), is a commonly used biochemical marker of lipid peroxidation[5]. The levels of MDA were significantly increased in SH-SY5Y cells treated with siRNA targeting WIPI4 (siWIPI4; Fig. 1d); Fer-1 treatment rescued these back to normal levels. The levels of peroxidised lipids were also assessed with BODIPY$_{581/591}$C11, a probe that stains both reduced and oxidized lipids, and emits the respective signals at different wavelengths. The 510/591 nm signal strength ratio, which provides an indication of the level of lipid peroxidation, was increased for SH-SY5Y cells treated with siWIPI4, as measured by flow cytometry (Extended Data Fig. 1g,h), suggesting that lipid peroxidation is induced and the observed cell death is due to ferroptosis.

Similar changes were observed in human induced pluripotent stem cells (iPSC)-derived neurons (i³ neurons) treated with anti-WIPI4 short hairpin RNAs (shRNAs) delivered by lentiviruses. All five anti-WIPI4 shRNAs induced more neuron death compared with the control and this increase was rescued by treatment with the ferroptosis inhibitor Fer-1 (Extended Data Fig. 1i). The 510/591 nm ratio of BODIPY$_{581/591}$C11

in i³ neurons was increased by WIPI4 depletion and rescued by Fer-1 treatment (Fig. 1e,f and Extended Data Fig. 1j, which shows the knockdown efficiencies and staining of neuron markers). A similar increase in cytotoxicity was observed for primary neurons treated with shRNA targeting mouse WIPI4 (Extended Data Fig. 1k).

In *rho*:EGFP zebrafish, injection of CRISPR guides targeting *wdr45* was confirmed to silence *wdr45* expression (Extended Data Fig. 1l) and resulted in photoreceptor degeneration at 10 days post fertilization (d.p.f.; Extended Data Fig. 1m) as well as a reduced lifespan (Fig. 2a). Treatment with Fer-1 from 5 d.p.f. to 10 d.p.f. rescued the loss of photoreceptors induced by *wdr45*-targeting CRISPR knockdown but Fer-1 did not affect the photoreceptors in the uninjected fish (Fig. 2b,c). Injection of *wdr45*-targeting CRISPR guides increased the concentrations of MDA in larvae at 5 d.p.f. compared with their uninjected siblings (Fig. 2d). Therefore, *wdr45* depletion induced ferroptosis and photoreceptor degeneration in zebrafish, and also resulted in reduced survival.

## WIPI4-mediated ferroptosis is autophagy independent but ATG2 dependent

Unlike WIPI4, the loss of function of another WIPI family protein, WIPI2 (ref. 6), which also acts in autophagy, did not induce cytotoxicity (Fig. 3a

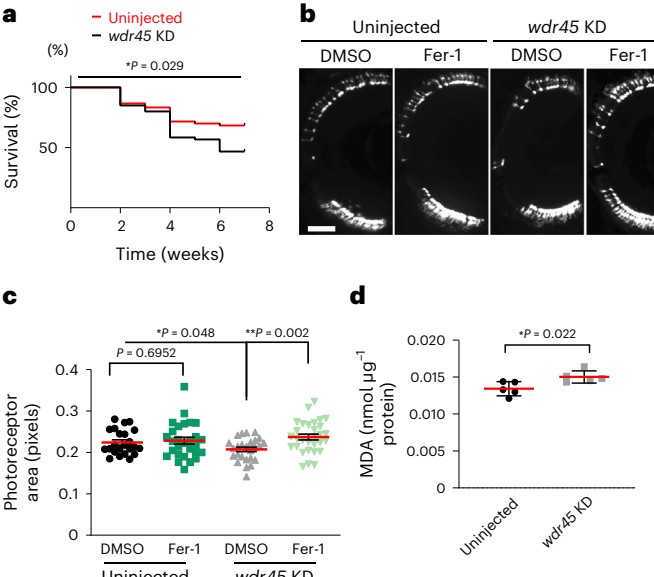

**Fig. 2 | WIPI4 loss of function induces ferroptosis in zebrafish. a**, Zebrafish juveniles with *wdr45* knockdown (*wdr45* KD) by CRISPR injection had reduced lifespans compared with their uninjected siblings from the age of 4 weeks. Survival rate of WIPI4 CRISPR mutants and their uninjected siblings over a period of 7 weeks (*n* = 60 fish per group); log-rank Mantel–Cox test. **b**, Treatment of *rho*:EGFP transgenic fish with 10 μM Fer-1 from 5 to 10 d.p.f. rescued the loss of photoreceptors following *wdr45* KD by CRISPR injection, whereas Fer-1 did not cause any change in the fluorescence rod area of uninjected fish. Representative images of sections across the eye of 10 d.p.f. *rho*:EGFP fish injected with *wdr45*-targeting CRISPRs and their uninjected siblings treated with DMSO or 10 μM Fer-1, respectively. Scale bar, 50 μm. **c**, Photoreceptor areas from the images in **b**; *n* ≥ 23 eyes per group. **d**, Increase in lipid peroxidation, represented by higher concentrations of MDA, in wild-type fish subjected to *wdr45* KD by CRISPR injection at 5 d.p.f. compared with their uninjected siblings; *n* = 5 biologically independent experiments, 30 fish each. Data are the mean ± s.d. *P < 0.05 and **P < 0.01; two-tailed unpaired Student's *t*-test. Source numerical data are provided.

and Extended Data Fig. 2a) in SH-SY5Y cells. Similarly, the knockdown of core autophagy genes—that is, *ATG7/10* (to simultaneously deplete both ATG7 and ATG10), *ATG16L1*, and *ATG2A* and *ATG2B* (*ATG2A/B*)—did not reduce the viability of SH-SY5Y cells (Extended Data Fig. 2b,c). Cytotoxicity was induced by WIPI4 knockdown in ATG16L1-null (autophagy-null cells lacking LC3-II and autophagosomes; Fig. 3b and Extended Data Fig. 2d) and Beclin 1-null (Extended Data Fig. 2e) cells as well as cells treated with an ULK1/2 inhibitor, which targets these core autophagy enzymes (Extended Data Fig. 2f). In addition, the autophagy inducer rapamycin did not abrogate cell death caused by WIPI4 KO (Extended Data Fig. 2g).

WIPI4 knockdown in combination with ATG16L1 or ATG7/10 knockdown decreased cell viability to a similar level observed for WIPI4 knockdown alone (Extended Data Fig. 2c). Co-injection of zebrafish with CRISPR guides targeting the critical autophagy gene *atg7* (as described previously[7]) and *wdr45* reduced the viability of photoreceptors compared with fish injected with only *atg7* guides (Fig. 3c,d). Thus, WIPI4 depletion induces ferroptosis independent of autophagy.

The homologous proteins ATG2A/B bind to WIPI4 to enable tethering of ATG2A/B to the sites of autophagosome formation, which allows them to transfer lipids from the endoplasmic reticulum (ER) to nascent autophagosomes[8]. We therefore wondered whether WIPI4 depletion caused ATG2A/B mislocalization, thereby changing the lipid composition of certain subcellular compartments to enhance lipid peroxidation and ferroptosis. To test this, we first assessed whether ferroptosis caused by WIPI4 depletion was dependent on ATG2. Knockout (Fig. 3e,f and Extended Data Fig. 3a) or knockdown (Extended Data Fig. 3b) of

ATG2A/B in SH-SY5Y cells prevented the cytotoxicity and elevated MDA levels (Extended Data Fig. 3c) caused by WIPI4 depletion (Extended Data Fig. 3d shows the WIPI4- and ATG2A-knockdown efficiencies), whereas ATG2A/B knockdown by itself did not change cell viability (Extended Data Fig. 2c). ATG2A overexpression was sufficient to reduce cell viability, which could be rescued by ferroptosis inhibitors (Fig. 3g). ATG2A overexpression did not induce apoptosis, as shown by the levels of cleaved caspase 3 (Extended Data Fig. 1e).

Transfection of WIPI4-knockdown cells with GFP–WIPI4-RS (siRNA-resistant) recovered the WIPI4 protein levels (Extended Data Fig. 3e) and cytotoxicity (Fig. 3h) compared with the scramble-treated control cells. However, an ATG2A/B binding-deficient mutant of WIPI4 (GFP-WIPI4-RS-ΔLOOP3; Extended Data Fig. 3f) did not rescue the cytotoxicity caused by WIPI4 depletion[9] (Fig. 3h), suggesting that ferroptosis buffering by WIPI4 depends on its interaction with ATG2. Several pathways have been reported to regulate the activation of ferroptosis, including those regulated by the cyst(e)in/glutathione peroxidase-4 (GPX4), ferroptosis suppressor protein-1 (FSP1)/coenzyme Q10 (ref. [10]) and FSP1/Vitamin K hydroquinone (VKH$_2$)[11]. PUFA biosynthesis also regulates ferroptosis sensitivity, and arachidonic acid- and adrenic acid-conjugated phospholipids are favoured ferroptosis substrates[12]. To understand whether the WIPI4–ATG2 axis falls into any of the above pathways, we challenged ATG2A/B-KO HeLa cells with inducers of the GPX4 (RSL3, erastin, sorafenib and FIN56), FSP1-QH$_2$/VKH$_2$ (iFSP1) and PUFA biosynthetic (by overloading cells with PUFAs) pathways. RSL3, erastin and iFSP1 reduced cell viability (Extended Data Fig. 4a) and induced cytotoxicity (Extended Data Fig. 4b) to similar levels in ATG2-KO and wild-type control cells, with Lip-1 successfully rescuing the cell death caused by these stimuli. FIN56 and sorafenib showed similar effects on cytotoxicity (Extended Data Fig. 4b). However, overloading cells with arachidonic acid (a PUFA) failed to reduce cell viability in ATG2A/B-double-KO cells compared with wild-type cells (Extended Data Fig. 4c).

### The ATG2–WIPI4 axis regulates ferroptosis dependent on mitochondria

The results above suggest that WIPI4–ATG2-regulated ferroptosis overlaps with the PUFA biosynthesis pathway but acts in parallel with other major ferroptosis pathways. We hypothesized that some ATG2-binding membranes are hotspots for lipid peroxidation and ferroptosis. We tested this by immunoprecipitating ATG2A-associated membranes, which were stained with BODIPY$_{581/591}$C11 and imaged to quantify the 510/591 nm ratio (Extended Data Fig. 4d). Interestingly, there was increased peroxidation of ATG2-interacting membranes in siWIPI4-treated cells but this was normalized when the cells were treated with Fer-1 (Extended Data Fig. 4e).

Mainly mitochondrial proteins were pulled down by ATG2A (Extended Data Fig. 5a). Although ER proteins were also pulled down, this occurred at lower levels compared with mitochondrial proteins (Sec23A compared with NUDFA9; Extended Data Fig. 5a). There may be other compartments associated with ATG2A/B, including lipid droplets. We hypothesised that the 'hotspot' membrane for ferroptosis should be one associated with more ATG2 in the absence of WIPI4. Consistent with its role in autophagy, WIPI4 depletion destabilized the localization of ATG2 on autophagosomes: GFP–ATG2A localized for shorter time periods on autophagosomes decorated with RFP–LC3 in HeLa cells treated with siWIPI4 compared with scramble-treated cells (Fig. 4a and Extended Data Fig. 5b). An ATG2A mutant, ATG2A-mLIR[13], which has compromised LC3 interaction and defective autophagosome biogenesis induced more ferroptosis compared with wild-type ATG2A (Extended Data Fig. 5c), consistent with the autophagy independence of this phenomenon (Fig. 3a–d and Extended Data Fig. 2).

The localization of ATG2A on mitochondria (super resolution microscopy in Extended Data Fig. 5d and confocal imaging in Extended Data Fig. 5e,f) was increased in WIPI4-depleted cells, but

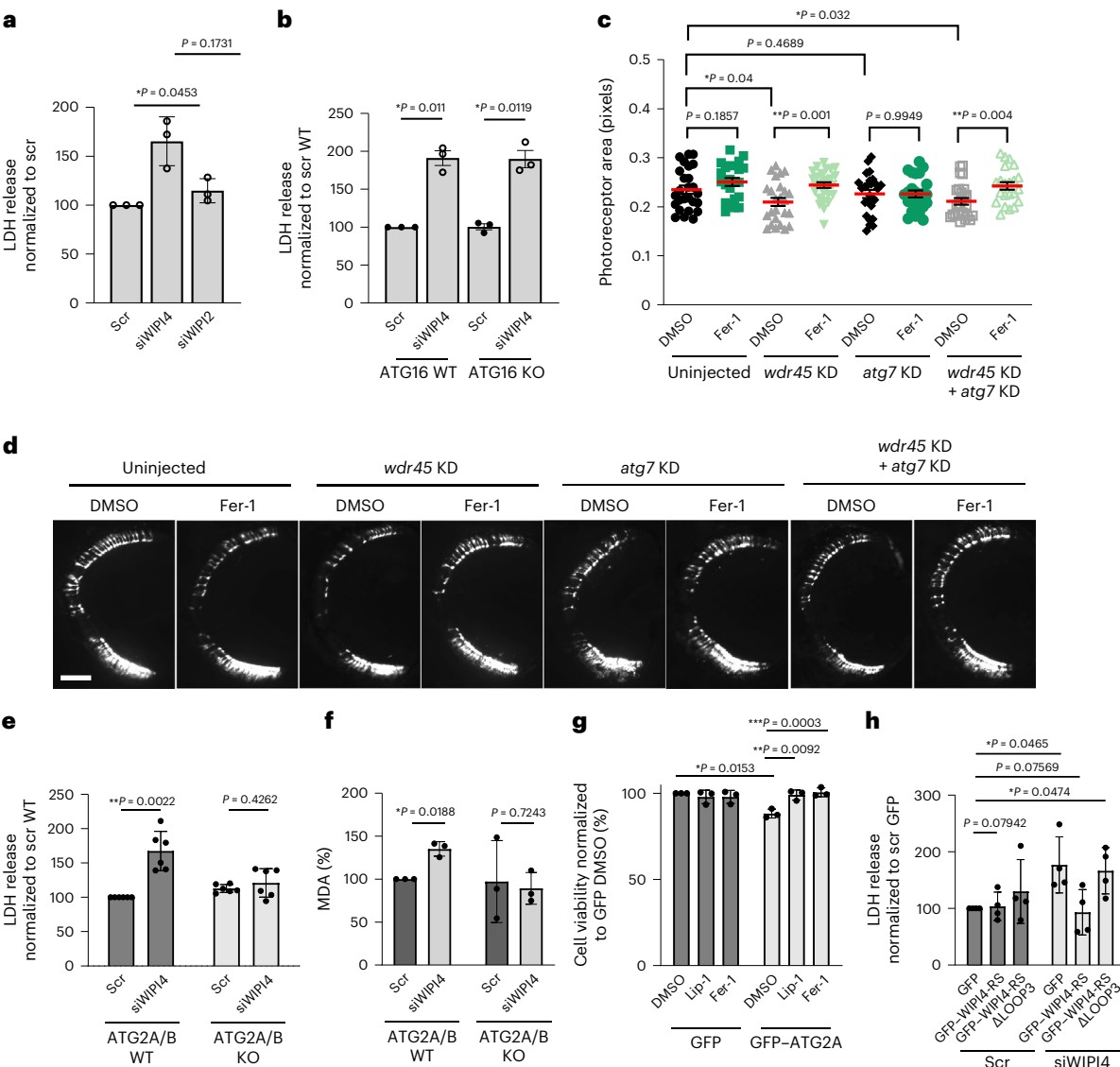

**Fig. 3 | WIPI4-mediated ferroptosis is autophagy independent but ATG2 dependent. a**, LDH release from SH-SY5Y cells following WIPI4-knockdown and WIPI2-knockdown via siRNA treatment (siWIPI4 and siWIPI2, respectively) in comparison to control cells (*n* = 3). **b**, LDH release from siWIPI4- and scramble siRNA (scr)-treated ATG16L1-KO and control (wild-type, WT) HeLa cells (*n* = 3). **c**, Rod photoreceptor area of *rho*:EGFP zebrafish retinas at 10 d.p.f. showing the autophagy-independent effect of *wdr45*-targeting CRISPR knockdown via injection with *wdr45*-targeting CRISPR guides (*wdr45* KD). We analysed cryosections across fish retinas of uninjected zebrafish larvae and their siblings injected with either *wdr45* KD or *atg7*-targeting CRISPR guides (*atg7* KD), or both in combination, followed by treatment with DMSO or 10 μM Fer-1 at 5–10 d.p.f. Data are the mean ± s.d.; *n* = 22 eyes per group; two-tailed unpaired Student's *t*-test. **d**, Representative images of *rho*:EGFP zebrafish retinas at 10 d.p.f. showing

the autophagy-independent effect of *wdr45* KD. Scale bar, 50 μm. **e**, ATG2A/B WT or CRISPR KO (ATG2A/B KO) HeLa cells were transfected with control siRNA or siWIPI4 for 72 h and subjected to an LDH assay (*n* = 3). **f**, MDA analysis of the cells in **e** (*n* = 3). Controls normalised to 100. **g**, Cell viability of ATG2A/B KO HeLa cells transfected with GFP or GFP–ATG2A for 24 h and treated with 100 nM Lip-1 or 1 μM Fer-1 for 20 h following a medium change (*n* = 3). **h**, LDH release from control and WIPI4-knockdown HeLa cells transfected with GFP, GFP–WIPI4 WT or LOOP3 siRNA-resistant mutant (ΔLOOP3; *n* = 4). **a**,**b**,**e**–**h**, Data are the normalized mean ± s.d.; *n*, number of biologically independent experiments; two-tailed one-sample Student's *t*-test for comparisons to the control and two-tailed paired Student's *t*-test for comparisons between other samples. *\*P* < 0.05 and *\*\*P* < 0.01. Source numerical data are provided.

was not significantly different on lipid droplets (Extended Data Fig. 5g). Similarly, fractionation experiments confirmed that WIPI4 depletion enriched ATG2 in the mitochondrial fraction but not in the post-nuclear-supernatant and post-mitochondrial fractions (Fig. 4b and Extended Data Fig. 5h). WIPI4 depletion also increased ATG2A localization at ER–mitochondria contact sites; this was also observed in ATG16L1-KO cells (Fig. 4c and Extended Data Fig. 6a). The integrity of ER–mitochondria contact sites was not obviously compromised in WIPI4-silenced cells (Extended Data Fig. 6b).

ATG2 localizes on ER–mitochondria contact sites by interacting with the ER protein TMEM41b[14] and the mitochondrial protein

TOMM40 (ref. [8]). WIPI4 depletion resulted in increased interactions between TOMM40 or TMEM41b (Extended Data Fig. 7a) and GFP–ATG2A, and TOMM40 and TMEM41b (Fig. 4d and Extended Data Fig. 7b). No TOMM40 was pulled down by TMEM41b in ATG2A/B-KO cells, possibly because the TOMM40/TMEM41b interaction is dependent on ATG2A[8]. Interestingly, siWIPI4-induced cell death is abrogated by TOMM40 depletion (Fig. 5a). These results further suggest that increased ATG2 localization at ER–mitochondria contact sites in the absence of WIPI4 causes ferroptosis.

If increased localization of ATG2 onto ER–mitochondria contact sites in siWIPI4 conditions is responsible for ferroptosis induction,

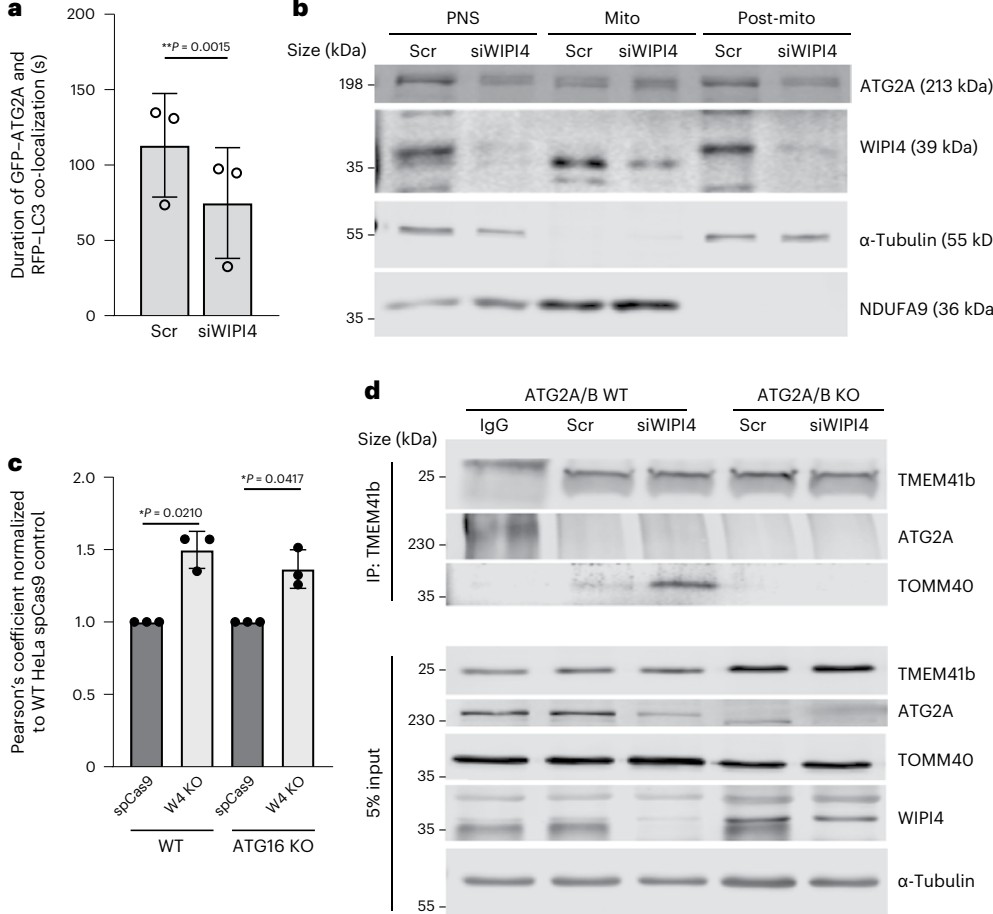

**Fig. 4 | WIPI4 depletion mislocalizes ATG2. a**, Average duration of GFP–ATG2 and RFP–LC3 co-localization in HeLa cells stably expressing RFP–LC3 quantified from videos taken of living cells (*n* = 3 independent biological repeats; ≥3 co-localization events in different cells imaged per experiment). The maximum length of each video was 140 s. Data are the mean ± s.d.; two-tailed paired Student's *t*-test. **b**, HeLa cells transfected with control siRNA or siWIPI4 were subjected to cell fractionation and immunoblotted with antibodies to the indicated proteins. For fractionation, isotonic buffer was used. Mitochondria were collected as pellets resuspended in PBS. This may explain the different mobility of WIPI4 in these different fractions. PNS, post-nuclear supernatant;

mito, mitochondrial fraction; post-mito, post mitochondrial fraction. **c**, Pearson's coefficient of endogenous ATG2A with calnexin in ATG16-KO and wild-type control cells (*n* = 3 biologically independent experiments; ≥20 fields imaged in each experiment). Data are the normalized mean ± s.d.; two-tailed one-sample Student's *t*-test for comparisons to the control and two-tailed paired Student's *t*-test for comparisons between other samples. **d**, ATG2A/B-WT and -KO HeLa cells transfected with control siRNA or siWIPI4 were immunoprecipitated (IP) with endogenous anti-TMEM41b and blotted for ATG2A and TOMM40. *P < 0.05 and **P < 0.01. Source numerical data and unprocessed blots are provided.

stopping the mitochondrial localization of ATG2 should abolish its role in promoting ferroptosis. We made an ER–mitochondria contact site localization-deficient ATG2 mutant, GFP-ATG2A-ΔMLD, with a deletion of the MLD domain that is required for mitochondrial localization[8]. In ATG2A/B-KO HeLa cells, there was less association in the mitochondrial fraction for GFP-ATG2A-ΔMLD (Fig. 5b,c) compared with GFP–ATG2A wild-type (GFP–ATG2A-WT). In contrast, WIPI4 interaction-deficient ATG2A, GFP-ATG2A-YFS (in which the three amino acids required for ATG2A's interaction with WIPI4—Y, F and S[9]—were mutated to A) was more enriched in the mitochondrial fraction (Fig. 5b,c). The interaction of ATG2A with TOMM40 was abrogated by the ΔMLD mutation but increased by the YFS mutation (Fig. 5d), which again suggests that ATG2A tends to localize more onto mitochondria when it cannot interact with WIPI4 (or in the absence of WIPI4). Overexpression of GFP-ATG2A-ΔMLD in ATG2A/B KO cells failed to increase cytotoxicity compared with GFP–ATG2A-WT (Fig. 5e). GFP-ATG2A-YFS induced the production of MDA to levels higher than those observed for GFP-ATG2A (Fig. 5f), whereas GFP-ATG2A-ΔMLD failed to change MDA levels.

We hypothesised that mitochondrial ATG2 localization caused dysregulation of lipids initially in mitochondria, which resulted in increased lipid peroxidation. A higher 510/591 nm ratio, determined

from confocal images, was observed for primary neurons that were treated with anti-mouse-WIPI4 smartpool shRNA and stained with MitoPerOx, a mitochondria-specific BODIPY$_{581/591}$C11 probe[15] (Fig. 5g and representative fluorescence staining of primary neurons in Extended Data Fig. 7c), suggesting more mitochondrial membrane peroxidation.

If lipid peroxidation on mitochondria is required for ferroptosis in siWIPI4-treated cells, then removing the lipid ROS on mitochondria or depleting cells of mitochondria should rescue ferroptosis. MitoTEMPO, a mitochondria-specific ROS scavenger[16], rescued the increase of cytotoxicity of siWIPI4 treatment of SH-SY5Y cells (Fig. 5h). In human iPSC-derived neurons, the increase in cytotoxicity by infection with ATG2A lentivirus was rescued by co-treatment with Mito-TEMPO (Extended Data Fig. 7d,e).

To specifically remove mitochondria in cells, we generated a stable HeLa cell line expressing PARKIN–FLAG. Treatment of this cell line with carbonyl cyanide 3-chlorophenylhydrazone (CCCP) induces mitophagy and depletes mitochondria[17]. In cells depleted of mitochondria, siWIPI4 failed to induce cell death, suggesting that WIPI4–ATG2-induced ferroptosis is mitochondria dependent (Fig. 5i and Extended Data Fig. 7f). However, in cells where the mitophagy receptor BCL2 interacting protein 3 like (BNIP3L, also known as NIX; Extended Data Fig. 7g) was

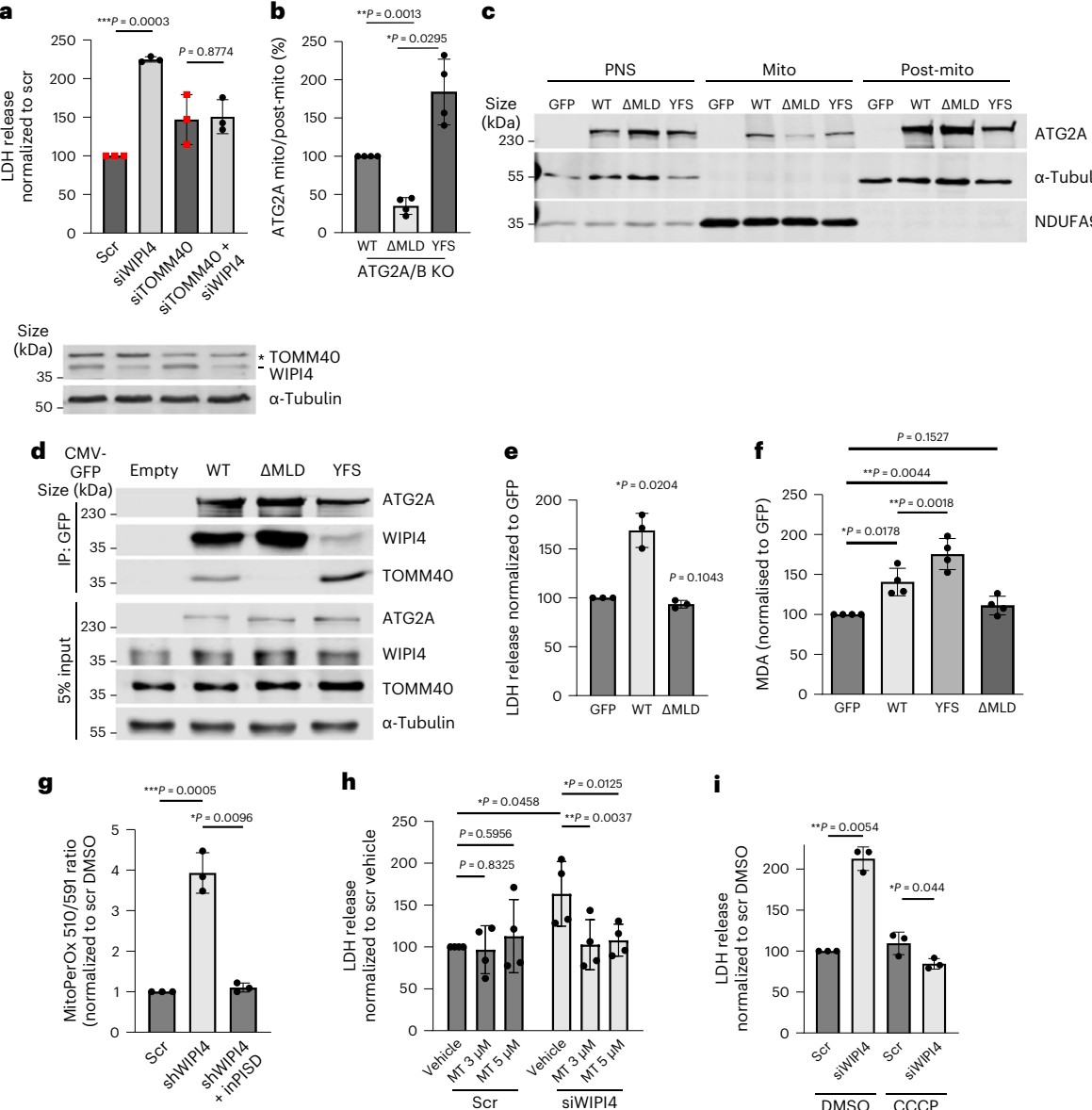

**Fig. 5 | The ATG2–WIPI4 axis regulates ferroptosis dependent on mitochondria. a**, LDH release from HeLa cells transfected with the indicated siRNAs (*n* = 3). LDH release was measured 72 h post transfection (top). The western blots show the silencing efficiency of siWIPI4 and siRNA targeting TOMM40 (siTOMM40). **b**, Ratio of ATG2A levels in the mitochondrial versus post-mitochondrial fractions (*n* = 4). **c**, ATG2A/B-KO HeLa cells transfected with GFP, GFP–ATG2A-WT, GFP-ATG2A-ΔMLD or GFP-ATG2A-YFS were subjected to cell fractionation and immunoblotted. **d**, HeLa cells transfected with GFP, GFP–ATG2A-WT, GFP-ATG2A-ΔMLD or GFP-ATG2A-YFS were immunoprecipitated by GFP-Trap beads and blotted for TOMM40 and WIPI4. **e**, LDH release from ATG2A/B-KO HeLa cells transfected with GFP, GFP-ATG2A-WT or GFP-ATG2A-ΔMLD (*n* = 3). **f**, Levels of MDA in ATG2A/B-KO HeLa cells transfected with GFP, GFP-ATG2A-WT, GFP-ATG2A-ΔMLD or GFP-ATG2A-YFS (*n* = 4). Control normalised to 100. **g**, Lipid peroxidation evaluated by MitoPerOx staining of primary neurons (*n* = 3; ≥5 fields were imaged per sample). The peroxidation levels were calculated as the fluorescence intensities of peroxidized (510 nm emission) signal

normalized to total signal (591 nm emission). The values were then normalized to that of scramble DMSO. Mouse primary neurons were cultured for five days before infection with scramble or anti-hWIPI4 smartpool shRNA delivered by lentivirus for 24 h, and then treated with 5 µM inPISD or DMSO in fresh media for one day. **h**, LDH release of SH-SY5Y cells transfected with control siRNA or siWIPI4 for 72 h, followed by incubation with the indicated concentration of MitoTEMPO (MT) for 48 h (*n* = 4). **i**, Mitophagy was induced in HeLa cells stably expressing PARKIN–FLAG with 12.5 µM CCCP for 48 h before transfection with siWIPI4. The cells were then cultured for 24 h in fresh media and LDH release was assessed (*n* = 3). **a**,**b**,**e**–**i**, Data are the normalized mean ± s.d.; two-tailed one-sample Student's *t*-test for comparisons to the control and two-tailed paired Student's *t*-test for comparisons between other samples; *n*, number of biologically independent experiments. WT, GFP–ATG2A-WT; ΔMLD, GFP-ATG2A-ΔMLD; YFS, GFP-ATG2A-YFS. *$P < 0.05$, **$P < 0.01$ and ***$P < 0.001$. Source numerical data and unprocessed blots are provided.

silenced, WIPI4 depletion still induced cell death and mitochondrial mass was unchanged in WIPI4-depleted HeLa cells (Extended Data Fig. 7h). These data suggest that WIPI4 depletion-mediated ferroptosis is not the consequence of defective mitophagy, which is consistent with our assertion that this process is independent of autophagy flux (Fig. 3 and Extended Data Fig. 2).

## Mitochondrial PE levels are increased by ATG2 and depend on its lipid transfer function

As a lipid transport protein[18], ATG2 might change the lipid composition of mitochondrial membranes. Phosphatidylethanolamine (PE) levels in the mitochondrial fractions, but not in whole cells, were increased by siWIPI4 (Fig. 6a,b); the levels of phosphatidylcholine (PC) and phosphatidylserine

(PS) in mitochondria were unchanged (Fig. 6c,d). The levels of unsaturated fatty acids were increased in mitochondria but not in total cell lysates (Fig. 6e). Importantly, unsaturated PE has been reported to be the main substrate of lipid peroxidation in ferroptosis[12,19,20].

To understand whether this is directly mediated by the ATG2 lipid transfer function, we made a lipid-transfer-deficient mutant of ATG2A, GFP–ATG2A-LTD[18] (Extended Data Fig. 8a). In contrast to GFP–ATG2A-WT, GFP–ATG2A-LTD failed to increase mitochondrial PE levels (Fig. 6f). Neither construct increased the PE levels of whole cells (Fig. 6g). As expected, overexpression of wild-type ATG2A (GFP–ATG2A-WT) increased cytotoxicity, whereas this was not observed for GFP–ATG2A-LTD (Fig. 6h). Similarly, GFP–ATG2A-LTD and GFP–ATG2A-ΔMLD did not cause the excessive morphological defects in zebrafish that were seen for GFP–ATG2A-WT and GFP–ATG2A-YFS (Fig. 6i, Extended Data Fig. 8b and Supplementary Table 1). This confirmed that mitochondrial localization of ATG2 and its lipid transfer function are required for its ferroptotic mediation in vivo. Thus, ATG2 mislocalization increased mitochondrial PE levels and cytotoxicity dependent on its lipid transfer function.

We then investigated whether reducing mitochondrial PE levels is sufficient to rescue WIPI4–ATG2-regulated ferroptosis. In i[3] neurons (Fig. 6j) and mouse primary neurons (Extended Data Fig. 8c), shWIPI4-induced (Extended Data Fig. 8d) cytotoxicity was rescued by an inhibitor of the mitochondrial PE synthase phosphatidylserine decarboxylase (PISD), inPISD[21]; as expected[22], cytotoxicity was also rescued by the mitochondrial ROS scavengers MitoTEMPO and reduced ubiquinone (MitoQH$_2$).

There are two isoforms of PISD in mammalian cells. One is specifically localized on mitochondria (mitoPISD) and the other localizes on both lipid droplets and mitochondria[23]. To specifically disable the PISD activity on mitochondria, we designed siRNA oligonucleotides targeting the mitochondrial isoform of PISD (Extended Data Fig. 9a). Silencing with this siRNA oligonucleotide lowered the PISD levels in the mitochondrial fraction but not in the other fractions of cell lysates (Extended Data Fig. 9b). Knockdown of mitoPISD in both HeLa (Fig. 7a and Extended Data Fig. 9c) and SH-SY5Y cells (Extended Data Fig. 9d) rescued the cell death induced by siWIPI4. Similarly, knockdown of mitochondrial PISD in ATG2A/B-KO HeLa cells rescued the increase in cytotoxicity (Fig. 7b), and MDA (Fig. 7c) and PE levels (Fig. 7d) induced by the expression of GFP–ATG2A-WT. Overexpression of mitochondria-targeted PISD (mitoPISD-myc-DDK) induced ferroptosis (Fig. 7e and validated in Extended Data Fig. 9e,f). These results suggest mitoPISD regulates cytotoxicity downstream of WIPI4–ATG2.

### Phosphatidylserine is transported into mitochondria by ATG2 and then locally converted to PE

We wondered how inhibition of mitochondrial PE synthesis rescued WIPI4–ATG2-induced ferroptosis. As the levels of total PISD protein

were not changed by WIPI4 depletion (Extended Data Fig. 9g), we considered that ATG2 might transport more PS (the substrate of PE synthesis by PISD) into mitochondria. ATG2 might also regulate the mitoPISD activity. We found that PS levels did not differ between scramble and WIPI4-knockdown conditions (Extended Data Fig. 9h,i). Considering that PISD efficiently converts mitochondrial PS to PE[24], we were concerned that we may not be able to observe any impact on mitochondrial PS after WIPI4 depletion/ATG2 overexpression. WIPI4 depletion in mitoPISD-knockdown cells did not result in elevated PS levels (Extended Data Fig. 9h). However, the ATG2 channel is believed to be passive and bidirectional depending on its cargo concentrations on donor-recipient membranes[25]. PS-to-PE conversion by mitochondrial PISD provides a sink in the acceptor membrane. Thus, if ATG2 increases PS transport into mitochondria in WIPI4-depleted cells, this may only be sustainable in the presence of PISD activity—otherwise ER-to-mitochondrial PS transport will cease as soon as the mitochondrial PS concentrations increase. Therefore, the above experiments could not discriminate between altered PS transport from the ER or enhanced PISD activity as causes for the increased mitochondrial PE after WIPI4 knockdown.

Accordingly, we designed an experiment to compare the amount of newly synthesized PE in intact cells and in a scenario where PS is provided to mitochondria independent of ER-to-mitochondrial transport. If the mitochondria still obtain PS independently of ER-to-mitochondrial transport, then mitochondrial PE synthesis should only be affected by the activity of PISD. In intact cells PS-loaded liposomes need to fuse with the plasma membrane to enable loading of PS into intracellular compartments, which is dependent on endocytosis and classical lipid transport pathways[26]. To allow direct PS loading of intracellular compartments bypassing these routes, we used Seahorse, a plasma membrane-specific permeabilizer that makes pores on the plasma membrane without disrupting intracellular membranes[27]. When cells are permeabilized by Seahorse before they are loaded with PS liposomes, the PS liposomes should be able to enter the cytoplasm independent of endocytosis and reach the mitochondria directly. Phosphatidylserine with a nitrobenzoxadiazole (NBD) fluorophore conjugated to its tail, 18:1-12:0 NBD-PS, can be converted to 18:1-12:0 NBD-PE, which enables tracking of newly synthesized PE. We imaged and quantified mitochondrial PE synthesized within 1.5 h after loading equal amounts of 18:1-12:0 NBD-PS liposomes to living cells (Extended Data Fig. 10a). In intact cells, siWIPI4 resulted in more mitochondrial PE compared with cells treated with scramble siRNA; however, this was not seen in ATG2A/B-KO cells (Fig. 7f). In Seahorse-treated cells (Fig. 7g), siWIPI4 failed to increase the PE synthesis in ATG2A/B wild-type cells. Knockdown of mitoPISD still stopped the synthesis of mitochondrial PE, suggesting that mitoPISD is still active in Seahorse-treated cells.

These data suggest that ATG2-dependent transport of PS to mitochondria, probably from the ER, results in increased mitochondrial PE synthesis after WIPI4 depletion in intact cells but not when the PS can

**Fig. 6 | Mitochondrial PE levels are increased by ATG2 and depend on its lipid transfer function. a–d**, HeLa cells transfected with control siRNA or siWIPI4 were subjected to cell fractionation. Equal amounts of protein samples were then used for PE, PC and PS measurements (**a**, *n* = 4; **b–d**, *n* = 3). **e**, Unsaturated fatty acid (UFA) levels in the mitochondrial fractions and total lysates of HeLa cells transfected with siWIPI4 or scramble siRNA (*n* = 3). HeLa cells transiently transfected with pSpCas9(BB)-2A-GFP with sgRNA targeting WIPI4 exon 3 or its control construct were cultured for 48 h before mitochondrial purification and lipid extraction (after calibration to protein concentrations). **f,g**, Levels of PE in mitochondria (**f**) and whole-cell lysates (**g**) of ATG2A/B-KO HeLa cells transfected with GFP, GFP-ATG2A-WT or GFP-ATG2A-LTD (*n* = 3). **h**, LDH release from ATG2A/B-KO HeLa cells transfected with GFP, GFP-ATG2A-WT or GFP-ATG2A-LTD (*n* = 7). **i**, Representative bright-field (left) and fluorescence (right) images of the phenotypic abnormalities found in clutches injected with 200 pg GFP empty vector or GFP-ATG2A-WT constructs compared with their uninjected siblings at 24 h post fertilization (h.p.f.). Injection of GFP-tagged constructs resulted in green-fluorescent fish with a mosaic pattern. Higher levels of toxicity

were observed for the clutches injected with GFP-ATG2A-WT compared with the GFP-injected clutches and their uninjected siblings, as indicated by more dead embryos (eggs containing black dense material) or abnormal phenotypes (yellow arrowheads). The higher-magnification images (right) showcase the range of morphological defects observed. Scale bars, 2 mm (main images showing clutches) and 1 mm (higher-magnification images showing individual embryos). **j**, Day 15 i[3] neurons were pre-treated with DMSO (control), 10 μM Z-VAD-fmk (Z-VAD), 20 μM necrostatin (Nec-1), 500 nM liproxstatin-1 (Lip-1), 5 μM Fer-1, 10 μM MitoTEMPO (MT), 10 nM MitoQH$_2$ (QH$_2$) or 5 μM inPISD for 12 h, followed by scramble shRNA or shWIPI4 lentivirus treatment in combination with the same drugs for 24 h. LDH release was then assayed as a measure of cytotoxicity (*n* = 3). **a–g**, Data are the mean ± s.d.; two-tailed paired Student's *t*-test. **h,j**, Data are the normalized mean ± s.d.; two-tailed one-sample Student's *t*-test for comparisons with scramble DMSO and two-tailed paired Student's *t*-test for comparisons with other samples. **a–h,j**, *n*, number of biologically independent experiments. \*P < 0.05, \*\*P < 0.01 and \*\*\*P < 0.001. Source numerical data are provided.

access mitochondria directly following treatment with Seahorse. This model was consistent with our observations that silencing of oxysterol binding protein like 5 and 8 (OSBPL5/8; also known as ORP5/8), PS transfer proteins, which also function at ER–mitochondria contacts[28], rescued the increase of cell death and mitochondrial PE levels in siWIPI4-treated cells (Fig. 7h,i and Extended Data Fig. 10b,c). PISD inhibition also protected against other ferroptosis pathways induced by RSL3 and erastin (Extended Data Fig. 10d), suggesting that PISD might be a substrate supplier that initially fuels mitochondrial lipid peroxidation and thereby increases susceptibility to ferroptosis.

Although these data suggest that the effects are dependent on PS transport to mitochondria, we cannot exclude the possibility that this may enhance PISD activity by allostery. The resultant increased PE levels in the mitochondria after WIPI4 depletion (Fig. 6a) are compatible with the increased ferroptosis, as PE is a major substrate of lipid peroxidation driving this process.

## Discussion

While we were working on this project, other studies suggested that *WDR45* depletion causes ferroptosis[29–31]. However, these reports,

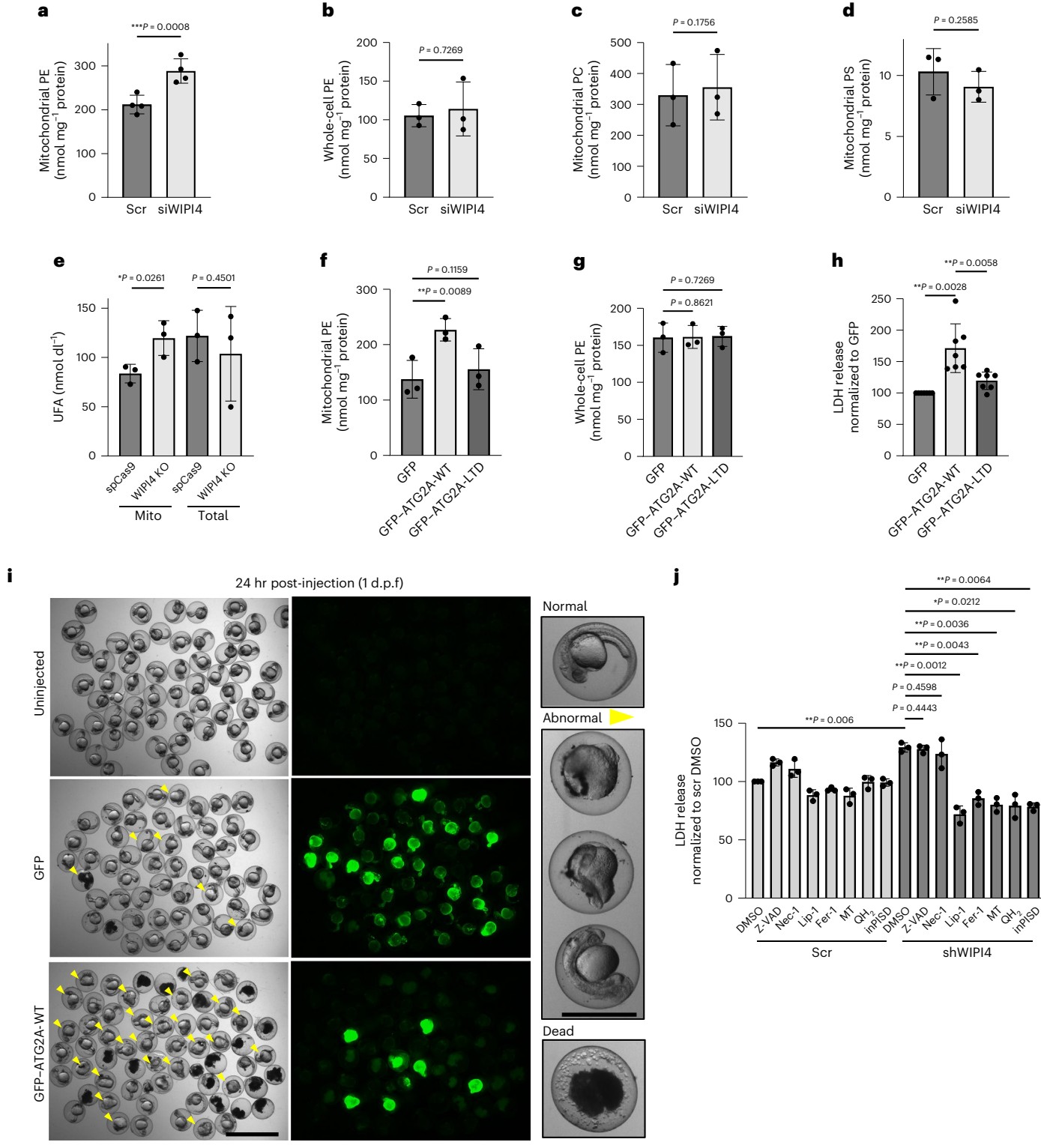

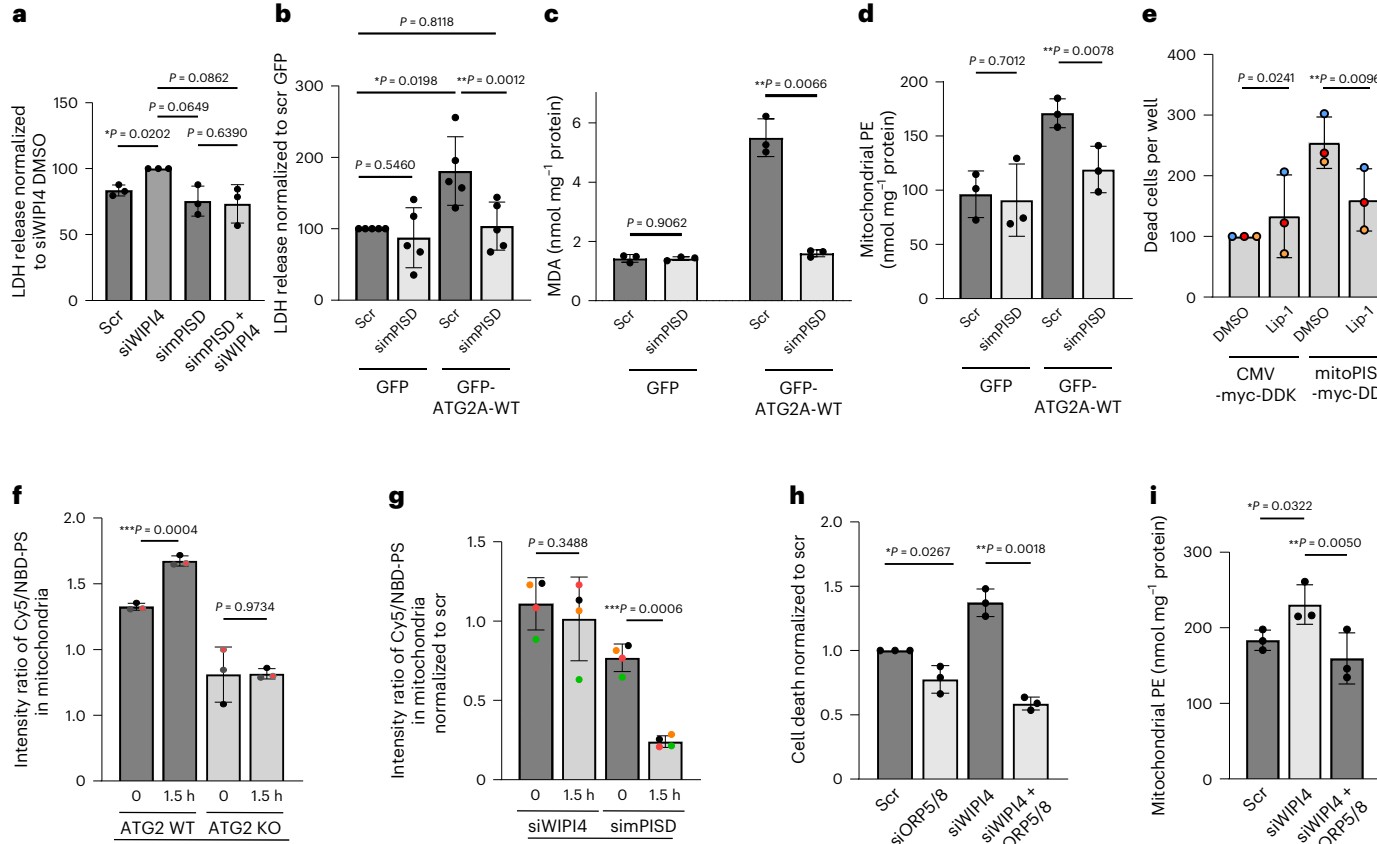

**Fig. 7 | ATG2 transports PS into mitochondria, which is then locally converted to PE. a**, HeLa cells were transfected with scramble siRNA, siWIPI4 or siRNA targeting mitochondrial PISD-2 (simPISD) for 48 h and the levels of LDH release were measured (*n* = 3). **b**, ATG2A/B-KO HeLa cells were transfected with scramble siRNA or simPISD for 48 h before transfection with GFP or GFP–ATG2A constructs for another 24 h. Conditioned media were used for the LDH assay (*n* = 3). **c,d**, Cells from **b** were lysed and assayed for MDA (**c**) and PE (**d**). Two-tailed paired Student's *t*-test (*n* = 3). **e**, Cell death of HeLa cells transfected with CMV-myc-DDK or mitoPISD-myc-DDK (*n* = 3). The cells were cultured for 24 h before treatment with DMSO or 100 nM Lip-1 in media containing IncuCyte red dye. The samples were imaged 48 h post transfection and dead cells were counted (number of red objects). Data are the average number of dead cells per well normalized to the CMV-myc-DDK-transfected/DMSO-treated control. **f**, Mitochondrial conversion of 18:1-12:0 NBD-PS to PE within 1.5 h measured by microscopy (*n* = 3).

**g**, HeLa cells were first treated with Seahorse for 15 min and then treated, imaged and assayed as in **f** (*n* = 4). **h**, Cell death of SH-SY5Y cells transfected with 50 nM siWIPI4 and/or siRNA to ORP5/8 (siORP5/8; *n* = 3). Cells were stained with CellTox green dye 48 h post transfection. Cell death was determined as the ratio of the green area (dead cells) to the phase area (total cells). **i**, Levels of PE in the mitochondrial fractions of HeLa cells transfected with the indicated siRNAs (*n* = 3). Cell fractionation was performed 48 h post transfection and mitochondrial samples with same protein abundance were used for the PE assay. **a,b,e,h**, Data are the normalized mean ± s.d. Two-tailed one-sample Student's *t*-test for comparisons to the siWIPI4 (**a**) or controls (**b,e,h**) and two-tailed paired Student's *t*-test for comparisons between other samples. **f,g,i**, Data are the mean ± s.d. Two-tailed paired Student's *t*-test. **a–i**, *n*, number of biologically independent experiments. **P* < 0.05, ***P* < 0.01 and ****P* < 0.001. Source numerical data are provided.

which were cell-culture based, suggested that the ferroptosis was due to defective autophagy-dependent mechanisms. Our cell culture and in vivo data instead argue that ferroptosis resulting from WIPI4 depletion is autophagy independent (a possibility that was not considered previously).

Thus, we have identified an autophagy-independent role for WIPI4 as a buffer against ferroptosis. The mechanism we have described suggests a pathway causing this form of cell death that probably operates in BPAN, which is dependent on ATG2-mediated PS transport resulting in more mitochondrial PE synthesis. In normal conditions WIPI4 stabilizes the carboxy (C) terminus of ATG2 onto autophagosomes by direct interaction of the two proteins and allows lipid transfer from the ER to autophagosomes. In the absence of WIPI4, localization of the C terminus of ATG2 onto mitochondria is increased, which leads to increased transfer of PS from the ER to mitochondria and higher levels of conversion to PE by mitochondrial PISD. The increase in mitochondrial PE levels promotes ferroptosis, probably because polyunsaturated PE is the main substrate of ferroptosis. When mitochondrial PISD is inhibited,

the ER–mitochondria concentration gradient of PS disappears, which slows the ER–mitochondrial PS transfer and ferroptosis (Extended Data Fig. 10e). These data underscore the importance of mitochondrial PE generation in ferroptosis[32,33]. Although this is a major driver in WIPI4 deficiency, the extent to which this is rate limiting in ferroptosis caused by other stimuli still needs detailed assessment.

Importantly, as PISD inhibition is pharmacologically tractable, this may open up therapeutic possibilities. This may be relevant not only to BPAN but also to conditions like Alzheimer's disease, Parkinson's disease and Huntington's disease[34], all of which have been associated with ferroptosis.

## Online content

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

## Methods

Our research complies with all relevant ethical regulations and guidelines. All zebrafish procedures were performed in accordance with the UK Animals (Scientific Procedures) Act with appropriate Home Office Project and Personal animal licences and with local Ethics Committee approval. The studies were performed in accordance with PREPARE and ARRIVE guidelines. Ethical approval was obtained from the University of Cambridge Animal Welfare and Ethical Review Body (AWERB) in accordance with the UK Animals (Scientific Procedures) Act under project licence P173861E7−Protocol 7 for Fig. 2a (survival assay) and Protocol 9 for all other zebrafish data.

### Reagents

**Cell lines.** HeLa cells (CCL-2; American Type Culture Collection, CVCL-0030) were cultured in DMEM medium (Merck, D6546) supplemented with 10% fetal bovine serum (Merck, F7524), 2 mM L-glutamine (Merck, G7513), and 100 U ml⁻¹ penicillin and 0.1 mg ml⁻¹ streptomycin (Merck, P0781). SH-SY5Y (ECACC, 85120602) cells were cultured in DMEM-F12 medium (Merck, D6421) supplemented with MEM non-essential amino acids (Thermo Fisher, 11140050), 10% fetal bovine serum, 2 mM L-glutamine and 100 U ml⁻¹ penicillin and 0.1 mg ml⁻¹ streptomycin. The HeLa CRISPR−Cas9 ATG2A/B-double-KO cell line (a gift from N. Mizushima[35]) was cultured in the same medium as the HeLa cells. The ATG16L1-KO HeLa cells and its wild-type control cell line were made in house. The Beclin 1-KO cell line and its wild-type control were a gift from W. Wei (Peking University, Beijing). All cell lines were maintained at 37 °C and 5% CO₂, and were regularly tested for mycoplasma.

**Antibodies.** The following primary antibodies were used for western blots (working concentration, 1:1,000): rabbit anti-WDR45 (WIPI4; 19194-1-AP) from Proteintech; mouse anti-α-tubulin (9026) and rabbit anti-actin (A2066) from Sigma Aldrich; rabbit anti-ATG2A (MBL Life Science, PD041); rabbit anti-GFP (ab6556), mouse anti-NDUFA9 (ab14713) and rabbit anti-Sec23a (ab137583), rabbit anti-TOMM20 (ab186735), mouse anti-GM130 (ab52649), mouse anti-LAMP1 (ab24170), rabbit anti-KDEL (ab2898) and mouse anti-β III tubulin [2G10] (ab78078) from Abcam; rabbit anti-LC3 (Proteintech, 14600-1-AP); mouse anti-TOMM40 (Santa Cruz Biotechnology, sc-365467); rabbit anti-TMEM41b (Novus Biologicals, NBP1-81552); rabbit anti-calreticulin (12238), rabbit anti-cleaved caspase 3 (9661), rabbit anti-ATG16L1 (8089), rabbit anti-ATG7 (2631) and rabbit anti-MAP2 antibody (8707S) from Cell Signaling Technologies; and rabbit anti-PISD (Atlas Antibodies, HPA031091).

The following primary antibodies were used for immunofluorescence (working concentration, 1:100): mouse anti-TOMM20 (Santa Cruz, sc-17764), mouse anti-calnexin (Abcam, ab112995), rabbit anti-hNIX (Cell Signaling Technologies, 12396), mouse monoclonal antibody to FLAG M2 (Sigma, F1804-200UG), mouse anti-ORP8 (Santa Cruz, sc-134409), rabbit anti-ORP5 (Atlas Antibodies, HPA038712-100UL), rabbit anti-IP3 receptor 1 (D53A5) (Cell Signaling Technologies, 8568) and mouse monoclonal [20B12AF2] anti-VDAC1/Porin + VDAC3 (Abcam, ab14734).

**Drugs and probes.** *Drugs.* The following drugs were used: Z-VAD-fmk (Promega, G7231), necrostatin-1 (Sigma, N9037), liproxstatin-1 (Sigma, SML1414), Fer-1 (Sigma, SML0583), *N*-acetyl-L-cysteine (Sigma, A7250), MitoTEMPO (Sigma, SML0737), deferoxamine mesylate salt (Merck, D9533), staurosporine (Sigma, S5921), erastin (Sigma, E7781), 1S,3R-RSL3 (Sigma, SML2234), iFSP1 (Sigma, SML2749), arachidonic acid (Sigma, 10931), FIN56 (Sigma, SML1740), sorafenib (Santa Cruz, sc-220125), inPISD inhibitor (or STK770988; Vitasmlab.biz), mitoquinol (Cayman Chemicals, 89950), CCCP (Sigma, C2759) and ULK1 inhibitor SBI-0206965 (Sigma, SML1540-5MG).

*Probes.* The following probes were used: BODIPY FL C₁₂ (Invitrogen, D3822), BODIPY₅₈₁/₅₉₁ C11 (Invitrogen, D3861), MitoPerOx (Abcam, ab146820) and nonyl acridine orange (Thermo Fisher Scientific, A1372).

**Constructs and siRNA oligonucleotides.** We used empty pEGFP (Clontech), pEGFP-C1-hATG2A (Addgene, 36456), GFP-WIPI4 (ref. 36) and Lact-C2-GFP (Addgene, 22852) plasmids. PLKO.1 transfer plasmids expressing shRNAs targeting human WIPI4 (RHS4533-EG11152) and mouse WIPI4 (RMM4534-EG54636) were purchased from the Horizon TRC Lentiviral shRNA library. Human ATG2A was cloned onto an EF1A lentiviral vector adapted from pMK1253 by removing the Tau.K18 (P301L/V337M) insert. The primers used are listed in Supplementary Table 2. Mitochondrial-only PISD-myc-DDK was made from PISD-myc-DDK isoform d (Origene, RC222868) by mutating three leucine residues, required for lipid droplet localization, to positively charged residues (I80K, I89K, I90R).

CRISPR−Cas9 WIPI4-KO constructs were generated as previously described[37]. Briefly, pairs of guide RNAs (sgRNAs) nicking *WDR45* were designed (Supplementary Table 4) using the CRISPR design tool available at http://www.genome-engineering.org/crispr/. Guide RNAs with overhangs were annealed and cloned into a BbsI-digested pSpCas9n(BB)-2A-GFP (PX458) vector (Addgene, 48138). HeLa cells were transfected with pairs of sgRNAs and cultured for two days before use for cell death assays. Only transient KO of *WDR45* was performed in this study and a pair of sgRNAs targeting exon 3 of the *WDR45* gene was used.

The scramble siRNA (ON-TARGETplus non-targeting control; D-001810), and siRNAs targeting human WIPI4 (J-019758-10 for general usage and J-019758-12 as oligonucleotide 2) and ORP5 (J-009274-10), ORP8 (J-009508-06) were predesigned. Mitochondrial PISD-specific siRNA oligonucleotides (Oligo 1, 5′-GAGAAGCUGGGAUUGGAGAUU-3′; and Oligo 2, 5′-GCACCGAUGUCUGGGAUUACU-3′) were designed against the coding region specific to mitochondrial isoforms of PISD (transcript variants NM_001326411−NM_001326421)[23]. The siRNAs targeting human NIX (5′-CCAUAGCUCUCAGUCAGAAUU-3′), ATG2A/B, WIPI2, ATG16L1 and ATG7/10 were reported previously[6,38]. The oligonucleotides were obtained from Dharmacon-Thermo Scientific.

### Techniques

**Transfection.** For the overexpression experiments, 1 μg complimentary DNA construct per well of a six-well plate was transfected with TransIT-2020 (Mirus, MIR5400). For the knockdown experiments, cells were transfected with either a single or double round of 20–100 nM siRNAs using Lipofectamine RNAiMAX (Invitrogen, 13778). In the single transfections, the cells were incubated with siRNA for 72 h or DNA constructs for (the last) 24 h. For the double transfections, the cells were transfected with 50 or 100 nM siRNA, followed by another transfection with 50 nM siRNA after 24–48 h.

**Mutagenesis.** GFP−WIPI4 was used as a template to mutate amino acids of the ATG2A binding site (LOOP3)[9] to alanine residues; the siRNA (J-019578-12) recognises the sequence without altered amino acid residues. The pEGFP-C1-hATG2A (Addgene, 36456) plasmid was used as a template to generate the mitochondria-localization-deficient mutant (GFP−ATG2A-ΔMLD) according to previously published work[18] by truncation, WIPI4 interaction deficient (GFP−ATG2A-YFS) mutant by mutating the three relevant amino acids to alanine residues[9] and the lipid transfer-deficient mutant (GFP−ATG2A-LTD) according to published work[18] by mutating the nine indicated amino acid residues to aspartates. Mutagenesis was performed using a QuickChange multi site-directed mutagenesis kit (Agilent Technologies, 200515) or Q5 site-directed mutagenesis kit (NEB, E0554) and the primers used are listed in Supplementary Table 2.

**Virus packaging for overexpression and knocking down genes in neurons.** WIPI4-targeting shRNA or GFP−hATG2A lentivirus were packaged using the third-generation packaging system. HEK293T cells seeded in a 10 cm dish coated with poly-D-lysine hydrobromide (Merck, P6407) were transfected with 3 μg transfer plasmids, 2.5 μg psPax2

and 1.6 µg PMD2.G following the manufacturer's datasheet for Lipofectamine LTX transfection reagent with plus reagent (Thermo Fisher, 15338100). Growth medium (4.5 ml) without antibiotics was added to the top of the transfection mixture and changed to viral harvest medium (30% FBS + 10 U potassium penicillin and 100 µg streptomycin sulfate per 1 ml growth medium) the next day. Virus was harvested at 40 and 64 h post transfection. The collection was filtered using low-protein-binding Durapore (0.45 µm) filter (Merck, SLHVR33RS) and then centrifuged at 114,000g using SW40Ti rotor for 90 min at 4 °C. After disposing the supernatant, the virus pellet was resuspended in 150 µl medium and the aliquots were stored at −80 °C.

**Culture and virus infection of neurons.** The maintenance and differentiation of i[3] neurons were conducted following published protocols[39]. Wild-type iPSC cells were seeded in plates coated with Matrigel (Corning, 354277) containing induction medium supplemented with 2.5 µM of the Rho-associated protein kinase (ROCK) inhibitor Y-27632 (Tocris Bioscience, 1254). The medium was replaced with fresh induction medium containing doxycycline, but without Y-27632, for two consecutive days. The desired number of Day 3 neurons were seeded in cortical culture medium[39] onto plates coated with poly-L-ornithine (Sigma, P3655) diluted in borate buffer (Thermo Fisher, 28341). Half of the medium was replaced every 3–5 days (96-well plate; $1 \times 10^4$ cells per well). The isolation and culture of mouse primary cortical neurons was described previously[7]. Lentivirus was added to the i[3] neurons 15 days after induction and to mouse primary neurons 5 days after in vitro culture. The medium was aspirated and replaced with fresh culture medium the day after infection. Experiments were performed 24–72 h after infection.

**Cell cytotoxicity assay.** Cell cytotoxicity was measured using an LDH Assay Kit (Abcam, ab65393) according to the manufacturer's instructions. Briefly, cells were plated on a six-well plate. After the drug treatments, knockdown and/or overexpression, the medium with which cells were incubated for 24 h was used for the assay. The medium was exposed to LDH reaction mix for 30 min and absorbance was measured at both 450 and 650 nm (reference wavelength). Cell cytotoxicity was calculated according to the equation: (test sample absorbance − low control absorbance) ÷ (high control absorbance − low control absorbance). The high/positive control used was cells lysed with lysis buffer and the low/negative control was untreated cells.

**Cell viability assay.** Cell viability was assessed using AquaBluer (MultiTarget Pharmaceuticals LLC, 6015) according to the manufacturer's instructions. Briefly, cells were cultured in a 96-well plate for 24 h before the assay following the indicated treatments. The cells were washed twice with PBS and then incubated for 4 h in medium containing AquaBluer. The fluorescence intensities were measured at an excitation wavelength of 540 nm and emission of 590 nm using a TECAN plate reader. Cell viability was calculated based on the formula: (average fluorescence value of the treated cells) ÷ (average fluorescence value of the untreated cells). Results were presented as the percentage of viable cells with respect to the control group.

**IncuCyte assay.** The IncuCyte S3 incubated live imaging system was also used for monitoring cell death. At least 30 images were taken for each sample and the images were analysed using the IncuCyte 2020 software. For cells that did not express GFP-tagged protein, the CellTox green cytotoxicity assay (Promega, G8743) was used to stain dead cells and the ratio of green area to total area (phase) was used as the measurement of cell death. For cells that expressed GFP-tagged proteins, IncuCyte Cytotox red dye was used for counting dead cells (Sartorius, 4632) and the number of dead cells (counted as red dots) per well was used as the measurement of cell death. In Extended Data Fig. 5c, the number of dead cells was normalized to the number of transfected cells

(count of GFP-positive objects) to calibrate to the expression levels of ATG2A WT and ATG2A-mLIR.

**Western blot analysis.** Cells were washed and lysed with RIPA buffer (50 mM Tris–HCl pH 7.4, 150 mM NaCl, 1% Triton X-100, 0.5% sodium deoxycholatemonohydrate and 0.1% SDS supplemented with protease (cOmplete, mini, EDTA-free Protease I; Merck) and phosphatase (Sigma Aldrich, P5726 and P0044) inhibitors cocktails). The lysates were incubated on ice for 15 min and centrifuged at 16,100g and 4 °C for 10 min to remove debris. The protein concentration was determined using a BCA protein assay kit (Thermo Fisher Scientific, 23225). Blots were probed overnight with the indicated primary antibodies and then with DyLight Fluors-conjugated (Invitrogen) secondary antibodies for 1 h before detection on an infra-red imaging system (LICOR Odyssey system). Densitometric analysis of the immunoblots was performed using IMAGE STUDIO Lite software.

**Immunoprecipitation assay.** Cells plated in 100 mm dishes were washed twice with PBS and lysed with cold lysis buffer (20 mM Tris–HCl pH 7.4, 100 mM NaCl, 0.5 mM EDTA, 0.5% NP40, and protease and phosphatase inhibitors cocktails). The lysates were incubated on ice for 15 min and isolated by centrifugation at 16,100g and 4 °C for 10 min to remove debris. The supernatants were moved to new tubes. A fraction (5%) of the sample was stored to be used as the input control. The remaining lysate was incubated overnight with primary antibodies or a control IgG antibody (Cell Signaling, 2729S) at 4 °C with gentle agitation. Subsequently, the lysate was mixed with Dynabeads protein A (Life Technologies) and incubated at 4 °C for 2 h. The beads were washed five times with lysis buffer and the immunoprecipitated proteins were eluted and denatured by boiling with 2×Laemmli buffer containing β-mercaptoethanol for 10 min at 100 °C. Proteins were detected by SDS–PAGE. EGFP-tagged proteins were pulled down using GFP-TRAP beads (ChromoTek) according to the manufacturer's protocol.

**Membrane pull down with ATG2A.** We identified the lipids associated with endogenous ATG2A that were oxidized using a combination of a pull down and lipid peroxidation assay using BODIPY$_{581/591}$C11. Cells were cultured in a 100 mm dish and stained with 2.5 µM BODIPY$_{581/591}$C11 diluted in Hanks' Balanced Salt Solution medium (Invitrogen) for 15 min. The cells were washed twice with PBS, collected by gently scraping in 1 ml cold PBS and centrifuged at 1,000g and 4 °C for 5 min. The pellets were resuspended in isotonic buffer (10 mM Tris–HCl pH 7.4, 250 mM sucrose and 1 mM EDTA supplemented with protease and phosphatase inhibitors cocktails) without detergents and gently homogenized by passing them 12 times through a 26-gauge needle on ice. The lysates were again centrifuged at 1,000g and 4 °C for 5 min and subsequently incubated overnight with 5 µl anti-ATG2A in 500 µl isotonic buffer at 4 °C with gentle agitation. The lysates were mixed with Dynabeads protein A (Thermo Fisher, 10001D) and incubated at 4 °C for 2 h. The beads were washed five times with PBS, transferred into a chambered coverglass (Invitrogen, 155382) and imaged using a LSM880 Zeiss confocal microscope. Volocity software (PerkinElmer) was used for quantification and analysis of the images. In parallel, proteins on the beads were eluted with 2×Laemmli buffer for 10 min at 100 °C and subjected to western blot analysis to detect proteins immunoprecipitated with the ATG2A.

**MDA measurement.** MDA concentrations were assessed using a Lipid peroxidation assay kit (Abcam, ab118970) according to the manufacturer's instructions. One confluent 60–100 mm dish of cells or 30 zebrafish at 5–10 d.p.f. were required for each sample. Cells or fish tissue were lysed with MDA lysis buffer supplemented with butylated hydroxytoluene (provided in the kit) stop solution. The cell lysates were gently homogenized by passing them 12 times through a 27-gauge needle on ice. Before fish homogenization in MDA lysis

buffer (provided in the kit), 5 d.p.f. larvae were deyolked by pipetting up and down in calcium-free Ringer's solution (5 M NaCl, 1 M KCl and 1 M NaHCO₃ in double-distilled water) to eliminate lipids in the yolk sac. The deyolked fish were collected after gentle centrifugation and supernatant removal. Deyolking is not required at 10 d.p.f. when lipids from the yolk are largely depleted. Homogenization of fish larvae in MDA lysis buffer supplemented with butylated hydroxytoluene was performed by sonication in the water-bath of a diagenode BIORUP-TOR (three cycles of 20 s each). The cell or fish homogenates were centrifuged at 13,000$g$ for 10 min at room temperature, and the supernatants were collected for the MDA assay. A TECAN Spark multimode microplate reader was used for measurements at the excitation and emission wavelengths of 532 and 553 nm, respectively. The values were referenced against a standard curve and calculated as the MDA concentration (nmol μg$^{-1}$ protein).

**Cell fractionation and mitochondria purification.** Cells were cultured in a 100 mm dish for cell fractionation and two 140 mm dishes for mitochondrial purification. The cells were washed twice with PBS, centrifuged at 500$g$ for 5 min and homogenized by passing them 12 times through a 26-gauge needle or 20 times through a Dounce homogenizer on ice in isotonic buffer (10 mM Tris–HCl pH 7.4, 250 mM sucrose and 1 mM EDTA containing protease and phosphatase inhibitor cocktails) for cell fractionation or (10 mM HEPES pH 7.4, 70 mM sucrose, 210 mM mannitol and 1 mM EDTA containing protease and phosphatase inhibitor cocktails) for mitochondrial purification. The cell homogenates were centrifuged at 1,500$g$ and 4 °C for 10 min to remove unbroken cells and nuclei. The post-nuclear supernatant was subjected to two sequential centrifugations at 10,500$g$ and 4 °C for 10 min with one wash of the pellets with isotonic buffer in between. The pellets consist of purified mitochondria and the supernatant is the post-mitochondrial fraction.

**Phospholipid measurement.** Intracellular or mitochondrial PE, PC and PS were assessed using PE, PC and PS assay kits (Abcam, ab241005; Sigma-Aldrich, MAK049; and Abcam, ab273295, respectively) according to the manufacturer's instructions. Briefly, for the PE and PC assays, cells from a 100 mm dish were used for phospholipids in total cell lysate and two 140 mm dishes were used for mitochondrial phospholipids. The cells were washed twice with PBS and collected into a tube for total cell lysates, whereas mitochondria were purified following the method described above. The pellets were resuspended in a 5% (vol/vol) solution of peroxide-free Triton X-100 in water and the samples were calibrated according to the protein concentration. The samples were heated to 80 °C and cooled to room temperature; this cycle was repeated another two times to fully extract lipids. The lysates were centrifuged at 10,000$g$ and 4 °C for 10 min to remove insoluble structures. For the PS assay, lipids were extracted using a Lipid extraction kit (chloroform free; Abcam, ab211044) according to the manufacturer's instructions following mitochondrial purification. The lipid film was resuspended in 100 μl of 1% Triton X-100. The fluorescence intensity of lipids stained with the corresponding dyes was measured using a TECAN instrument at 538 nm excitation and 587 nm emission. The levels of PE, PC and PS were calculated according to the equations in the protocols.

**Measurement of unsaturated fatty acids.** The levels of unsaturated fatty acids were measured using a Lipid assay kit (unsaturated fatty acids; Abcam, ab242305). The same amount of HeLa cells was used for mitochondria extraction and lipid extraction as described earlier. Each lipid sample was resuspended in 60 μl DMSO. We incubated 5 μl of each resuspended sample (recalibrated according to the protein concentration) or standards with 150 μl of 18 M sulfuric acid in a 1.5 ml Eppendorf tube at 90 °C for 10 min and followed the manufacturer's instructions for the subsequent steps.

**Immunofluorescence microscopy of cultured cells.** HeLa cells cultured on coverslips were fixed in 4% paraformaldehyde for 5 min. Fluorescence was detected using a LSM880 Zeiss confocal microscope with a 63× oil-immersion lens. Images were acquired using the ZEN Black 2.6 Carl Zeiss Microscopy software. Volocity software (PerkinElmer) was used for co-localization analysis (Pearson's correlation coefficient). Image J (Fiji) was used to quantify the intensity and area of the fluorescent signals. The cells in Extended Data Fig. 5e were stained live with MitoTracker deep red FM (Thermo Scientific, M22426) for 20 min before fixation, and anti-ATG2A (MBL, PD041) after fixation, and mounted with ProLong gold antifade mountant (Thermo Fisher, P10144). Confocal images in other figures were imaged with cells mounted with Invitrogen ProLong gold antifade mountant with DAPI DNA stain (Thermo Fisher, P36941). The time period of RFP–LC3 and GFP–ATG2A co-localization was quantified from movies imaged live using a LSM780 Zeiss confocal microscope. HeLa cells stably expressing RFP–LC3 (made in house) were transfected with GFP–ATG2A construct and cultured for 24 h before live imaging. The time interval between frames was 3.87 s. The movies started when a GFP and RFP double-positive event was identified and ended when GFP–ATG2 left RFP–LC3. There were a maximum of 36 frames in the movies, making the maximum observed time period 140 s. There might be events where GFP–ATG2 and RFP–LC3 co-localize for longer than 140 s but the events we observed were generally shorter than that.

**Super-resolution microscopy.** HeLa cells seeded on high-precision size 1.5 coverslips (Carl Zeiss Ltd) were transfected with BFP–Sec61β and pSpCas9(BB)-2A-GFP-scr/pSpCas9(BB)-2A-GFP-Exon3 sgRNA construct and cultured for 36 h. The cells were then stained with antibodies to ATG2A (MBL, PD041) and TOM20-F10 (Insight Biotech, sc-17764), and mounted with Invitrogen ProLong gold antifade mountant (Thermo Fisher, P10144). Super-resolution structured illumination microscopy was performed using an Elyra PS1 instrument (Carl Zeiss Ltd). The samples were examined on the microscope using a ×63 1.4 numerical aperture plan-APO Carl Zeiss objective lens and Immersol 518 F (23 °C) immersion oil. Image acquisition was carried out using the ZEN 2012 Elyra edition software in which datasets were collected with five grating phases, five rotations and sufficient $z$ positions spaced 110 nm apart to form a 2 mm-deep volume of raw super-resolution structured illumination microscopy data. Optimal gratings were selected for each wavelength used.

Thresholding was based on the underlying raw data and determined algorithmically using the Imaris software as part of the modelling process. The data were captured with identical acquisition settings in all cases and therefore the range of image intensities was directly comparable between all images.

**Proximity ligation assay.** Rabbit anti-IP3R3 (ITPR3) (Chemicon), mouse anti-VDAC1 (Abcam), PLA probe anti-rabbit PLUS (Sigma, DUO92002–100RXN), anti-mouse MINUS (Sigma, DUO92004-100RXN) and Duolink detection fluorophore red (Sigma, DUO92008; excitation, 594 nm and emission, 624 nm) were used. The experiments were conducted as described in the figure legend. In principle, if the two proteins of interest were located ≤40 nm apart, the connector oligonucleotides hybridized with the PLA probes and after ligation, the signal was amplified by rolling circle amplification.

**Flow cytometry.** SH-SY5Y cells were stained for 30 min in growth medium supplemented with 5 μM BODIPY$_{581/591}$C11. After staining, the medium was collected; the cells were washed with PBS and lifted with trypsin. The cells were collected in PBS and combined with the previous PBS and medium. Next, the cells were sorted on a Becton Dickinson LSR Fortessa flow cytometer. The oxidized and total signals were acquired simultaneously using 488 and 568 nm lasers, and detected with 530/30 and 590/30 filters, respectively. For the measurement of

mitochondria mass, HeLa cells were stained with 2.5 μM nonylacridine orange (Thermo Fisher Scientific, A1372; excitation, 490 nm and emission, 540 nm) at 37 °C for 30 min for labelling mitochondria, and sorted using an Attune NxT analyser. The signals were acquired with a 488 nm laser and detected with a BL1 530/30 nm detector. Median fluorescence intensity analysis of labelled mitochondria was performed by gating on single cells. The data were processed using the FlowJo v10.8 Software.

**Preparation of liposomes.** A 3 mM phospholipid working mixture was prepared using 18:1-12:0 NBD PS (Avanti Polar Lipids, 810195C) and 18:0-16:0 PC (Avanti Polar Lipids, 850465) dissolved in import buffer (300 mM sucrose, 10 mM Tris–HCl pH 7.5, 150 mM KCl and 1 mM dithiothreitol) at a ratio of 75:25% (PC:NBD-PS). The mixture was left at room temperature for 1 h for the liposomes to form. Unilamellar liposomes were formed using an Avanti extruder (30 passes).

**Measurement of de novo mitochondrial PE synthesis.** HeLa cells cultured in live-imaging dishes (VWR International, 734-2905) were stained with MitoTracker Red CMXRos (Thermo Fisher, M7512) diluted 1:2,000 in growth medium for 10 min in the incubator to segment the mitochondrial area for future quantification. MAS Buffer was prepared (220 mM mannitol, 70 mM sucrose, 10 mM $KH_2PO_4$, 5 mM $MgCl_2$ and 2 mM HEPES). The cells were first washed with PBS (containing $MgCl_2$ and $CaCl_2$) and then with MAS Buffer pre-warmed to 37 °C. The NBD-PS/PC liposomes in warm MAS buffer containing 5 μl ml$^{-1}$ Duramycin–Cy5 conjugate (Molecular Targeting Technologies, D-1002) were loaded for 50 min and live imaged within 5 min (Time 0) using a LSM880 Zeiss confocal microscope with a ×63 oil-immersion lens. After imaging at Time 0, the samples were washed three times with PBS buffer and fresh warm MAS buffer was added. After 40 min in the incubator, the cells were stained again with Duramycin–Cy5 for 50 min and imaged (Time 1.5 h). For the samples with Seahorse treatment, the cells were incubated with MAS buffer (with 1 mM EGTA) containing 2 nM Seahorse XF-PMP (Agilent, 102504-100) for 15 min before imaging as described above. ImageJ was used to quantify the grey value of the Cy5 signal in the region of interest, which was double-positive for NBD signal and MitoTracker Red. The grey value of Cy5 was then normalized to the grey value of NBD in the corresponding region of interest to calculate the Cy5 intensity per arbitrary unit of NBD-PS.

**Zebrafish maintenance and crosses.** Adult fish that were between 6 and 18 months old were bred to generate embryos and larvae for the experiments described hereafter. For Fig. 2a (survival assay, Protocol 7), 0–7-week-old zebrafish were used. Zebrafish larvae were used for different experiments at the following ages: 0–10 d.p.f. for Figs. 2b,c, 3c,d and Extended Data Fig. 1m; 0–5 d.p.f. for Fig. 2d; 5 d.p.f. for Extended Data Fig. 1l, and 0–2 d.p.f. for Fig. 6i and Extended Data Fig. 8b.

The zebrafish were maintained and cultured under standard conditions under a 14-h light and 10-h dark cycle. Embryos were collected from natural spawnings, staged according to established criteria[40] and reared in embryo medium (5 mM NaCl, 0.17 mM KCl, 0.33 mM $CaCl_2$, 0.33 mM $Mg_2SO_4$ and 5 mM HEPES pH 7.2) at 28.5 °C in the dark. Embryos from crosses of Tupfel Longfin wild-type zebrafish were used for the CRISPR–Cas9 experiments. The *rho*:EGFP line [Tg2(*rho*:EGFP)cu3] expresses EGFP in the rods of the fish retina[41] and was used to investigate rod-cell death after genetic and pharmacological manipulation with ferroptosis modulators. Offspring from outcrosses of the *rho*:EGFP line (heterozygous) to AB wild-type fish were treated with 0.003% phenylthiourea from 24 h.p.f. to inhibit pigmentation and to allow visualization of the fluorescent photoreceptors and the selection of EGFP-positive fish at 3 d.p.f. Fish collected for experimental analyses were culled by an overdose of 1 mg ml$^{-1}$ 3-amino benzoic acid ethyl ester (MS222) before sample processing.

**CRISPR–Cas9 micro-injections.** Four CRISPR sgRNAs per gene designed by Dharmacon (GE Healthcare Dharmacon, Inc.) were used together to target the zebrafish *wdr45* or *atg7* genes (sequences in Supplementary Table 3). To maximize the knockdown efficiency, 100 ng of each sgRNA was mixed with 4 μl TRACR RNA (Dharmacon Edit-R CRISPR–Cas9 synthetic tracrRNA; Dharmacon, U002005) and 1.6 μl nuclease-free water[42]. The mixtures were incubated at room temperature for 10 min before the addition of 2.64 μl of 2 M KCl and stored at −80 °C as 1.5 μl aliquots. On the day of injections, 5 μg Cas9 nuclease protein NLS (Horizon Discovery, CAS12206) was added to each thawed aliquot and incubated for 5 min at 37 °C, after which 0.2 μl phenol red was added to the samples to visualize the injected droplet.

Embryos were collected immediately after spawning and injected at the one-cell stage. the volumes injected were calibrated to a final amount of 0.32 ng CRIPSPR sgRNAs targeting *atg7* and/or 0.64/0.96 ng for *wdr45*. For the double-knockdown experiments, CRISPR injections were performed sequentially−*wdr45*-targeting CRISPR–Cas9 solution was injected first, followed by *atg7*-CRISPR–Cas9 solution. Uninjected siblings were kept from each clutch in every experiment for comparison.

**Endogenous *wdr45* expression analysis.** To determine the efficiency of the *wdr45* knockdown by CRISPR injection, we compared the expression levels of endogenous *wdr45* of CRISPR-injected fish with their uninjected siblings at 5 d.p.f. Pools of ten fish from seven independent clutches were collected at 5 d.p.f. and stored at −80 °C. Messenger RNA was extracted using an RNeasy-plus mini kit (Qiagen) and a QIAshredder kit column (Qiagen) according to the manufacturer's instructions. A total of 1 μg RNA from each sample was then used to generate cDNA using a High-capacity reverse transcription kit (Applied Biosciences, 4368814) following the standard protocol. A primer pair targeting the last exons of the *wdr45* gene (exons 10 and 11) was used for quantitative PCR (primer sequences in Supplementary Table 2). Quantitative PCR was performed in triplicate and the relative gene expression level was calculated after normalization to *rbs11* internal controls using the $2^{-\Delta\Delta CT}$ method with logarithmic transformation for statistical analysis.

**Drug treatment of zebrafish.** GFP-positive offspring from crosses of the *rho*:EFGP line, previously treated with phenylthiourea from 1 d.p.f. and screened for EGFP expression at 3 d.p.f. as described earlier, were treated with Fer-1 at a concentration of 10 μM from 5 to 10 d.p.f. The drug was replenished daily until collection of fish for processing.

**Cell death in *rho*:EGFP zebrafish—fixation, embedding and cryosectioning.** Quantification of rod photoreceptor degeneration was analysed as previously described[41]. EGFP-positive offspring from *rho*:EGFP crosses to wild-type fish were culled at 10 d.p.f. by the addition of an excess of MS222 and fixed overnight in 4% paraformaldehyde in PBS at 4 °C. After washes in PBS, the zebrafish were equilibrated overnight in 30% sucrose in PBS at 4 °C, embedded in Scigen Tissue Plus optimal cutting temperature compound and frozen on dry ice before being stored at −20 °C. Transverse cryosections (10 μm) were cut through the central retina using a LEICA CM3050 S cryostat and collected on Thermo Scientific Menzel Gläser, SuperFrost Plus slides. Imaging and analysis of images were performed as described[41].

**Survival study.** Wild-type TL fish injected with 0.96 ng *wdr45*-targeting CRISPR–Cas9 were raised in parallel to their uninjected siblings up to the age of 7 weeks to evaluate changes in survival. After injection, the fish were maintained in the dark in an incubator at 28.5 °C until 5 d.p.f. and then transferred to the aquarium facility (60 fish per group). The numbers in each group were counted on a weekly basis. Survival was analysed using the log-rank (Mantel–Cox) method to compare the two groups.

**ATG2A construct micro-injections and abnormality quantification.** Embryos from crosses of wild-type TL fish were injected with pEGFP, pEGFP-C1-hATG2A, pEGFP-C1-hATG2A-YFS, pEGFP-C1-hATG2A-ΔMLD

or pEGFP-C1-hATG2A-LTD as described earlier. The DNA constructs were diluted to 100 or 200 ng μl$^{-1}$ in Danieau's solution (17.4 mM NaCl, 210 μM KCl, 120 μM MgSO$_4$, 180 μM Ca(NO$_3$)$_2$ and 1.5 mM HEPES pH 7.6) and injected in different volumes to result in final DNA amounts ranging from 60 to 450 pg, into embryos at the one-cell stage. EGFP expression was visualized after 24 h using a LEICA M205 FA fluorescence microscope and abnormal fish were quantified in each group at 24 and 48 h.p.f. Any fish with aberrant morphology and developmental defects compared with their uninjected siblings were scored as abnormal. The number of fish in each group depended on the clutch size, with a minimum of 12 fish per condition.

**Live imaging of zebrafish embryos.** Live imaging of embryos injected with ATG2 constructs was performed using a LEICA M205 FA microscope equipped with a LEICA DFC7000T camera and the Leica Application Suite X (LAS X) software. Bright-field and EGFP-fluorescence images were taken using a LEICA 10450028 lens and the EGFP filter of an X-Cite 200DC fluorescence illuminator lamp. Larvae were anaesthetized using MS222 at 48 h.p.f. before imaging. Anaesthetic was not required at 24 h.p.f. as motility was limited by the presence of the chorion.

### Statistics and reproducibility
**General information on statistical methods.** Tissue culture data are presented as the (normalized) mean ± s.d., except for Extended Data Fig. 4e where data are the median ± 95% confidence interval. Tissue culture data were analysed using a two-tailed Student's $t$-test, except for Extended Data Fig. 5d where a $\chi^2$ test was used. All in vivo experiments and tests were randomly assigned but no randomization was performed for the cell culture experiments. For the western blots, protein levels were normalized to the indicated internal control proteins.

**General information on reproducibility.** For the western blotting, immunomicroscopy, flowcytometry, lipid assays, cytotoxicity and viability assays, quantification and statistics were derived from $n$ = 3 independent experiments, unless otherwise specified in the legends. Sample sizes were chosen on the basis of extensive experience with the assays we have performed. No data were excluded from the analyses.

Zebrafish data are presented as the mean ± s.e.m. or s.d., as indicated on each graph, and were analysed using a two-tailed Student's $t$-test or log-rank (Mantel–Cox) test.

All statistical analyses were performed using GraphPad Prism 9 or Microsoft Excel; *$P < 0.05$, **$P < 0.01$, ***$P < 0.001$ and NS, not significant.

### Reporting summary
Further information on research design is available in the Nature Portfolio Reporting Summary linked to this article.

### Data availability
All data supporting the findings of this study are available from the corresponding author on reasonable request. Source data are provided with this paper.

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

### Acknowledgements

We thank D. McEwan for ATG2A constructs, M. Gratian and C. Puri for assistance with structured illumination microscopy. Funding for this study was obtained from the UK Dementia Research Institute (funded by the MRC, Alzheimer's Research UK and the Alzheimer's Society), Great Ormond Street Hospital Children's Charity and The National Institute for Health Research Cambridge Biomedical Research Centre at Addenbrooke's Hospital. The views expressed are those of the author(s) and not necessarily those of the NHS, the NIHR or the Department of Health and Social Care. M.F. received funding from Takeda Science Foundation. We thank C. Puri for help with the super-resolution microscopy.

### Author contributions

A.L. and L.S. contributed equally. D.C.R. conceived of and oversaw the project. Y.Z., M.F., A.L. and L.S. performed experiments and analysed data. Y.Z., M.F. and A.L. wrote the manuscript. D.C.R. and A.F. edited the manuscript. D.C.R. and M.F. provided resources and funding for the investigation.

### Competing interests
D.C.R. is a consultant for Drishti Discoveries, PAQ Therapeutics, MindRank AI, Retro Biosciences and Nido Biosciences. All other authors declare no competing interests.

### Additional information
**Extended data** is available for this paper at https://doi.org/10.1038/s41556-024-01373-3.

**Correspondence and requests for materials** should be addressed to David C. Rubinsztein.

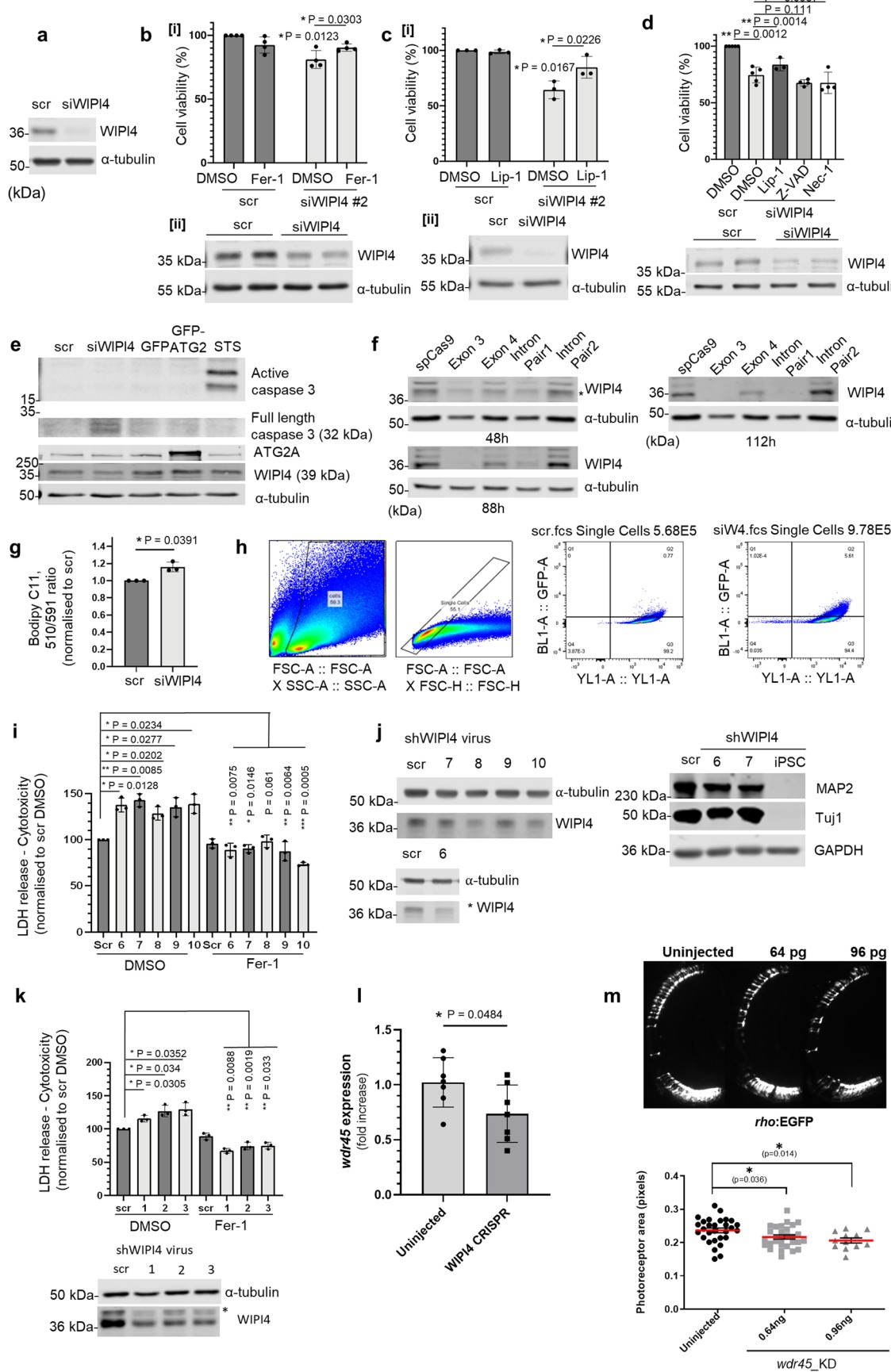

**Extended Data Fig. 1 | See next page for caption.**

**Extended Data Fig. 1 | WIPI4 loss of function induces ferroptosis. a**. Blots of SH-SY5Y cells transfected with control or WIPI4 siRNAs for 72 hours (this western blots were run 4 times, once for each biological repeats). **b** and **c**. SH-SY5Y (b, i) or HeLa cells (c, i) were transfected with control or WIPI4 siRNA Oligo 2. Cell viability was examined in the cells prior to treatment with DMSO or 1 μM Fer-1 (**b**) or 100 nM Lip-1 (**c**) (n = 3 independent experiments). Blots below show the knockdown efficiency (n = 3). **d**. Cell viability (n = 5 independent experiments) measured using AquaBluer in control and WIPI4 knockdown HeLa cells treated with DMSO (control), 1 μM Z-VAD-fmk, 10 μM necrostatin-1, 100 nM liproxstatin-1 (Lip-1) and 1 μM ferrostatin-1 (Fer-1) for 48 hours. Cells were grown for another 24 hours before immunoblotting. **e**. HeLa cells transfected with control, WIPI4 siRNA, GFP, or GFP-ATG2A were immunoblotted. Cells treated with 1 μM staurosporine for 4 hours were positive control for cleaved caspase 3 (this experiment was done twice). **f**. HeLa cells transfected with pSpCas9(BB)-2A-GFP with gRNA targeting WIPI4 Exon 3 or its control construct were grown for indicated time before lysis for knockout efficiency detection. **g**. Lipid peroxidation evaluated by Bodipy$_{581/591}$C11 staining of SH-SY5Y cells. The ratio of peroxidised (510 emission) to total (591 emission) signal was calculated as a derivative in Flowjo (n = 3 independent experiments). **h**. The sequential gating strategy for flow cytometry experiments in **g**. SH-SY5Y cells positive of Bodipy$_{581/591}$C11 staining was defined by thresholding against unstained SH-SY5Y cells. Cells with greater red (YL1-A) fluorescence than that of unstained cells (YL1-A + ) were selected as positive of Bodipy581/591C11 staining (Q2 and Q3). Cells in Q2 are positive of both reduced and oxidized Bodipy$_{581/591}$C11 signal and are quantified for oxidized (510) and total (591) signals. The ratio of GFP-A median vs YL1-A median is 1984/8723 = 0.2274 for scr and 2144/7826 = 0.274 for siWIPI4

sample in this experiment. The numbers of events are 568000 for scr and 978000 for siWIPI4. **i**. Cytotoxicity of i$^3$ neurons treated by scramble or 5 deconvoluted shRNA oligos for 24 hours (n = 3 independent experiments). **j**. i$^3$ neurons infected with lentivirus packaged with anti-human WIPI4 shRNAs from **i**. The right panel: i$^3$ neurons infected with shRNA6 and shRNA7 lentiviruses, undifferentiated iPSC cells. Microtubule-Associated Protein 2 (MAP2) is a neuron specific protein. Tuj1 antibody reacts with the β-III isoform of tubulin, which is expressed primarily in neurons. Both MAP2 and Tuj1 are used to distinguish neurons from other cell types. Western blots were run 3 times, once for each biological repeat. **k**. Primary neurons infected with lentiviruses packaged with anti-mouse WIPI4 shRNAs were subjected to LDH assay before lysis and blotting (n = 3 independent experiments). **l**. The injection of *wdr45*-targetting CRISPRs caused a reduction in the expression of zebrafish wdr45 measured by qPCR at 5 d.p.f. Graph represents the relative expression of wdr45 normalized to the levels of rbs11 in fish injected with *wdr45*-CRISPRs compared to their uninjected siblings (N = 7 independent experiments 10 fish per group; unpaired t-test after log-transformation). **m**. *wdr45*-KD by the injection of 64 or 96 pg of CRISPRs caused a reduction in the population of fluorescently-labelled rods in rho:EGFP transgenic fish at 10 d.p.f. The upper panel shows representative images of EGFP-labelled rods in cryosections across the central retina of *rho*:EGFP zebrafish at 10 d.p.f. Scale bar = 10 μm. The lower panel shows the quantification of the photoreceptor area. N = 12 eyes per group. Data and error bars in b-k are normalized. Mean ± SD, two-tailed one-sample t-test to control and two-tailed paired t-test between other samples. l and m: Mean ± SD, unpaired t-test. Source numerical data and unprocessed blots are provided.

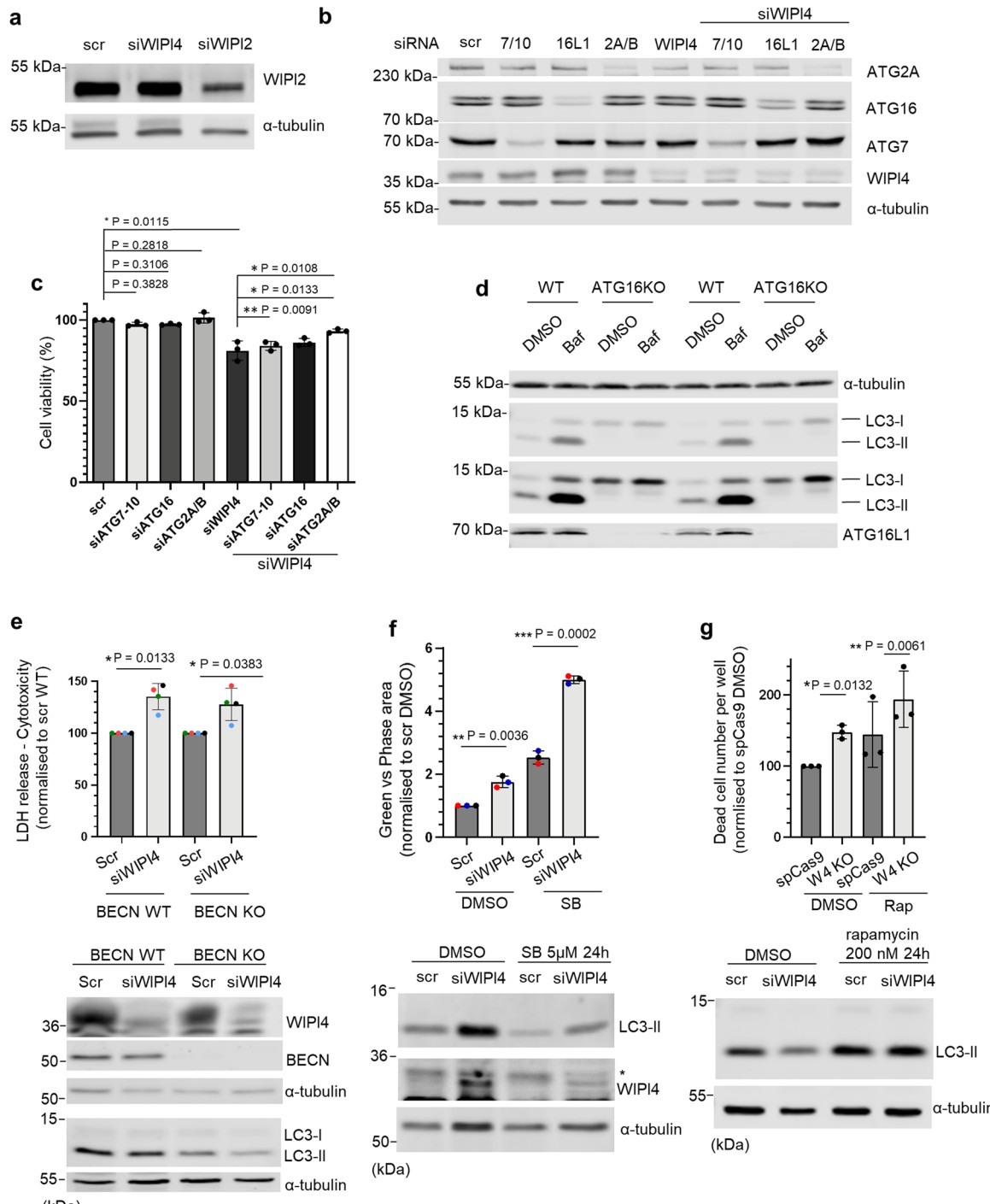

**Extended Data Fig. 2 | See next page for caption.**

**Extended Data Fig. 2 | WIPI4-depletion induced ferroptosis is autophagy-independent. a**. Blots of samples in Fig. 3a showing the knockdown efficiency of WIPI4 and WIPI2 (n = 3). **b**. Blots showing the knockdown efficiency of ATG7, WIPI4, ATG7, ATG16, ATG2A (n = 3). Two-tailed one sample t-test between scramble and siRNA knockdown of ATG2 genes and WDR45, two-tailed paired t-test between siWIPI4 and siWIPI4+ siATG7/10 double knockdown, siWIPI4 and siWIPI4 + siATG16 double knockdown, aiWIPI4 and siATG2A/B double knockdown. **c**. Cytotoxicity (n = 4) of SH-SY5Y cells with siRNA KD of ATG7/10, ATG16L1 or ATG2A/B, alone or in combination with WIPI4 knockdown. **d**. Blots showing that there is no LC3-II due to defective lipid conjugation in ATG16L1 knockout HeLa cell line we used and this is an autophagy-null cell line. Cells were treated with DMSO or 250 nM bafilomycin A1 for 4 hours (n = 3). **e**. Cytotoxicity measured by LDH assay of Beclin-1 knockout HeLa and its control cells treated as indicated (n = 4). On Day 0, cells were transfected with WIPI4 or scramble siRNA and grown for 48 hours before LDH assay. The western blots show the WIPI4 and Beclin-1 levels in the cells treated as described above. Note that the data points from each individual biological replicate are shown in different colours. Two-tailed one-sample t-test between scramble and siWIPI4 of WT cells, and same with scramble and siWIPI4 of Beclin 1 knockout cells. **f**. Cell death levels of HeLa cells measured with staining with CellTox Green Dye from Promega (n = 3). Cells were transfected with 50 nM WIPI4 or scramble siRNA and then treated with 5 μM ULK1 inhibitor SBI-0206965 for 24 hours before staining. Images were taken with the IncucyteS3 live imaging system and dead cells were quantified as the ratio of green (dead cells) area vs. phase (total cells) area using the Incucyte 2020 software. Western blots show the WIPI4 and LC3-II levels in the cells treated as described above (western blots were done for each biological replicates to detect knockdown efficiency). **g**. Cell death levels of HeLa cells treated as indicated (n = 3). On Day 0, HeLa cells were transfected with pSpCas9(BB)-2A-GFP construct containing gRNA targeting WIPI4 gene Exon 3 (KO) or its control construct (spCas9). Cells were grown for 24 hours before treatment with 200 nM rapamycin in media containing Incucyte Cytotox Red Dye from Sartorius. Images were taken between 48-72 hours after transfection. Dead cells were counted as red objects. WIPI4 and LC3-II levels are in blots. Data and error bars in c, e-g are normalized Mean ± SD. Two-tailed one sample t-test to control and two-tailed paired t-test between other samples. N represents the number of biologically independent experiments. Source numerical data and unprocessed blots are provided.

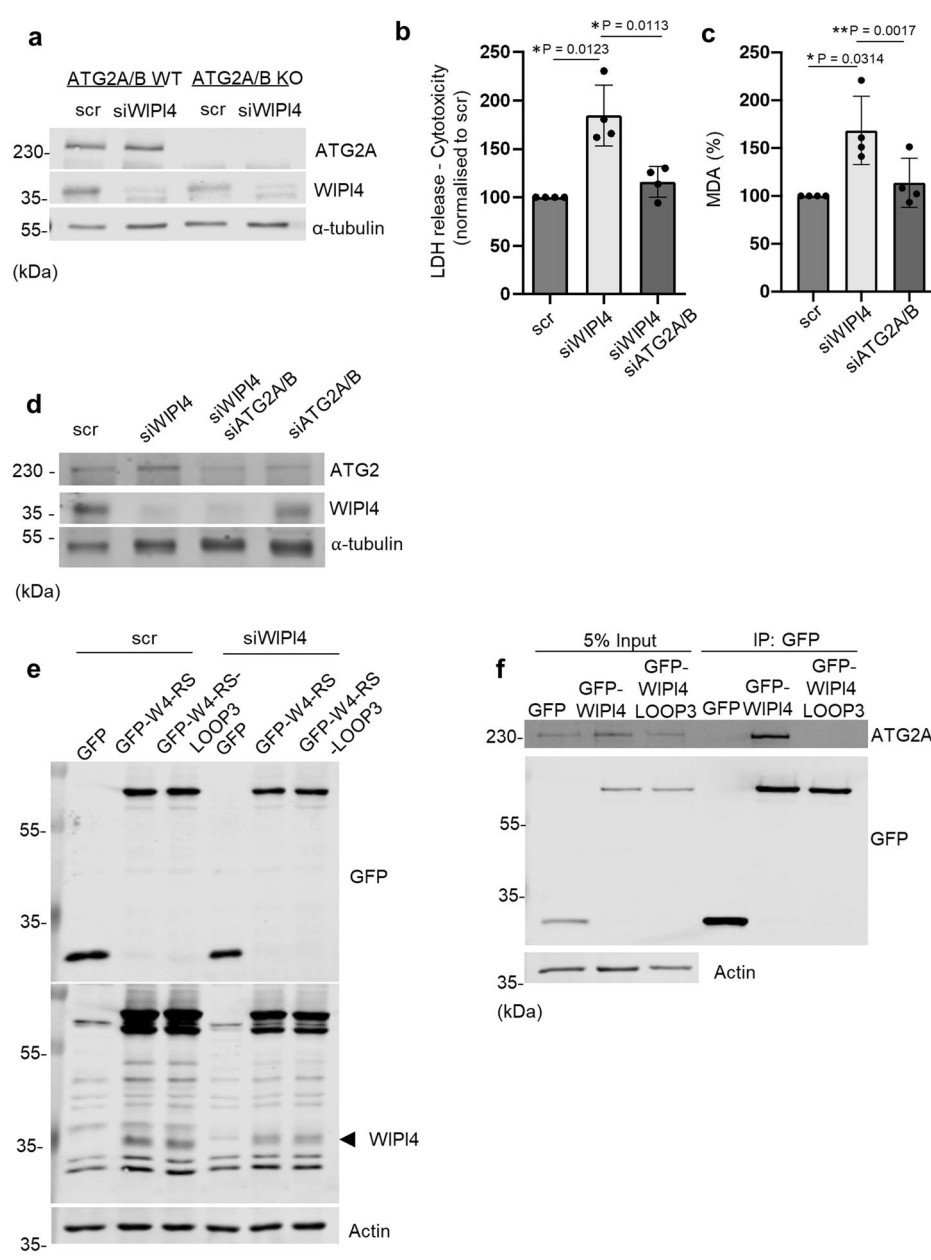

**Extended Data Fig. 3 | WIPI4-depletion induced ferroptosis is ATG2-dependent. a.** Blots of samples in Fig. 3e stained with indicated antibodies (representative blots of 6 biological replicates). **b, c.** SH-SY5Y cells were transfected with scramble, WIPI4, or WIPI4 in combination with ATG2A/B siRNAs for 72 hours and tested for cytotoxicity with LDH assay (b, n = 4), MDA levels (c, n = 4) and cell viability with AquaBluer (d, n = 4). Normalized mean ± SD. Two-tailed one sample t-test to control and two-tailed paired t-test between other samples. ns; not significant, *P < 0.05, **P < 0.01, ***P < 0.001. **d.** SH-SY5Y cells in b-c were lysed and subjected to western blot and immunoblotted with indicated antibodies (n = 3). **e.** Control or WIPI4 knockdown HeLa cells transfected with GFP, GFP-WIPI4-siRNA resistant mutant, or GFP-WIPI4-LOOP3-siRNA-resistant mutant were blotted with GFP and WIPI4 antibodies to check the recovery of WIPI4 levels (this experiment was done 3 times). **f.** HeLa cells transfected with GFP, GFP-WIPI4 WT, or GFP-WIPI4-LOOP3 were immunoprecipitated by GFP-TRAP and blotted with ATG2A antibody (n = 3). N represents the number of biologically independent experiments. Source numerical data and unprocessed blots are provided (this experiment was done twice).

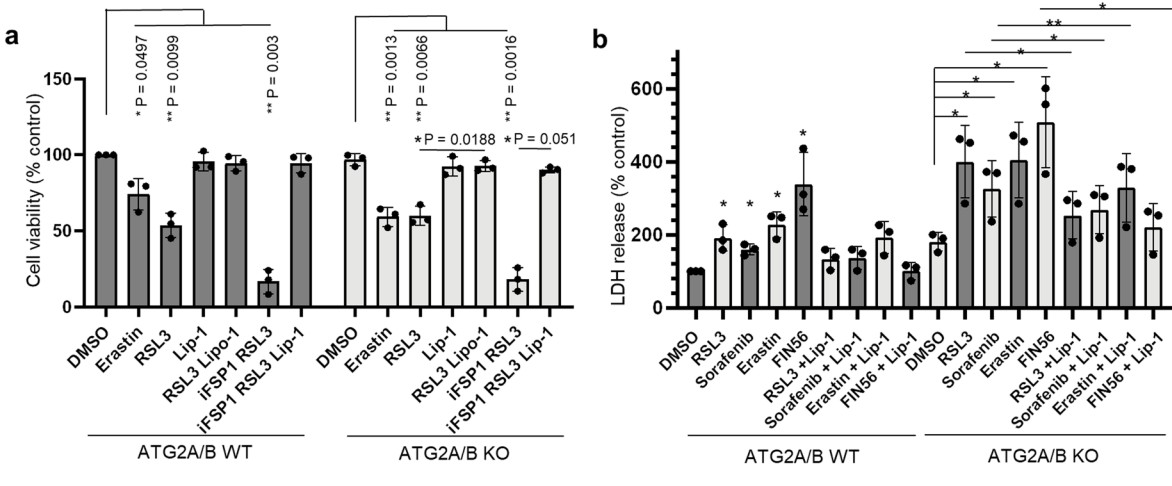

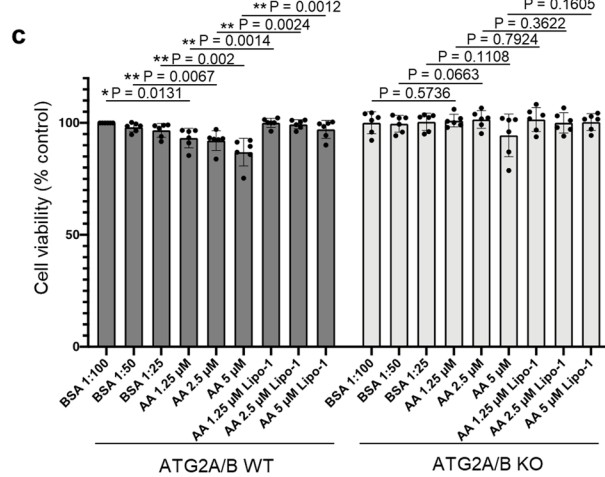

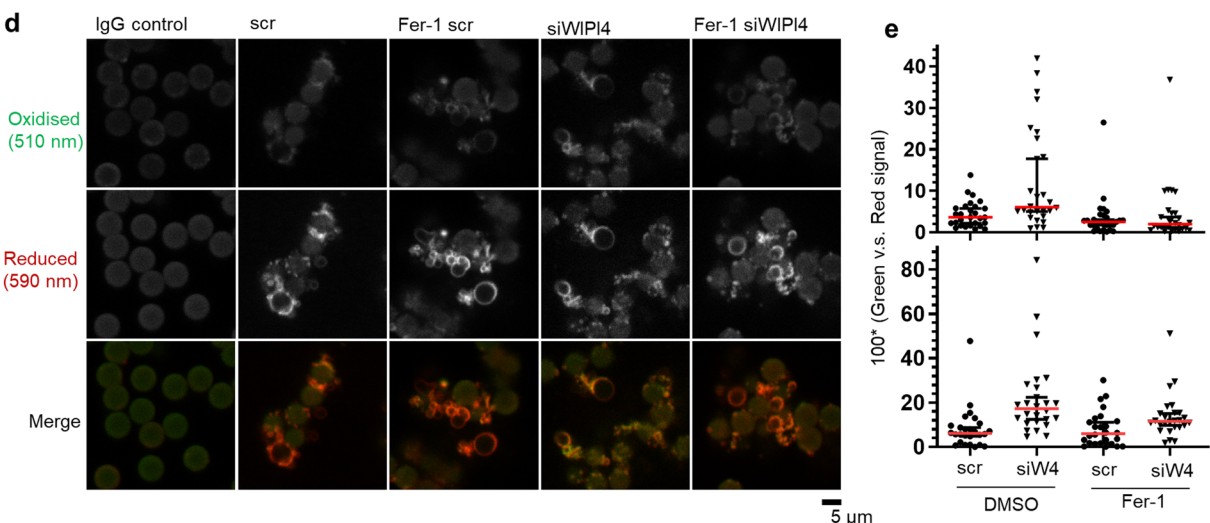

**Extended Data Fig. 4 | See next page for caption.**

**Extended Data Fig. 4 | WIPI4–ATG2 axis is a novel ferroptosis pathway.**
**a**. Cell viability was measured by Aquabluer staining in ATG2A/B WT and KO HeLa cells treated with DMSO, 10 μM erastin, 3 μM RSL3, 1 μM iFSP1, 100 nm Lip-1, or indicated combinations of these drugs (n = 3). **b**. Cell cytotoxicity was examined by LDH assay in ATG2A/B WT and KO HeLa cells treated with 3 μM RSL3, 10 μm sorafenib, 10 μM erastin, 5 μM FIN56 with or without 100 nm Lip-1 for 24 hours (n = 3). P values could be found in the numerical source data. **c**. Cell viability measured by Aquabluer assay in ATG2A/B WT and KO HeLa cells treated with 1.25 μM, 2.5 μM, or 5 μM arachidonic acid or same concentration of BSA, with or without 100 nM Lip-1 (n = 5) for 24 hours. **d**. HeLa cells transfected with control or WIPI4 siRNAs for 72 hours were treated with 1 μM Fer-1 for 24 hours. The cells were incubated with Bodipy$_{581/591}$C11 for 15 minutes before

cell lysis and were then subjected to needle homogenization without detergent prior to immunoprecipitation with endogenous ATG2A. Indicated fluorescent strength on the objects attached to magnetic beads was observed with confocal microscopy. This experiment was done twice with HeLa and SH-SY5Y cells. **e**. Quantification of representative experiments measuring the shift of Bodipy C11 signal on ATG2A pulled-down beads in SH-SY5Y cells (upper, n = 1) and HeLa cells (lower, n = 1). Datapoints represent the ratio of two fluorescent levels (510 nm, oxidized / 590 nm, reduced) of each taken field. Median with 95% CI. Data and error bars in **a-c** are normalized mean ± SD. Two-tailed one sample t-test to control and two-tailed paired t-test between other samples. N represents the number of biologically independent experiments. Source numerical data are available in source data.

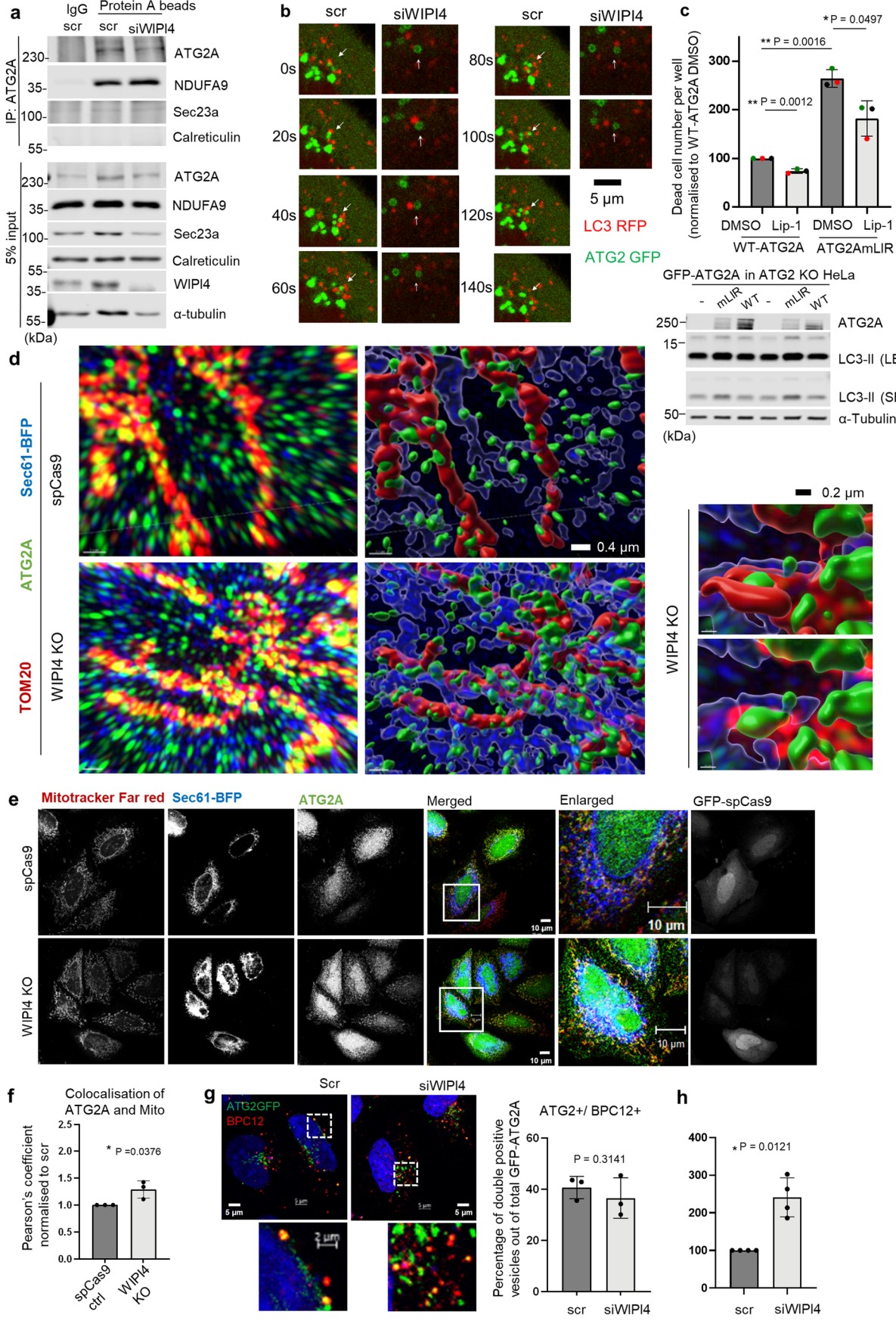

**Extended Data Fig. 5 | See next page for caption.**

**Extended Data Fig. 5 | ATG2A mislocalises more onto mitochondria in WIPI4 depleted conditions. a**. After immunoprecipitation with endogenous ATG2A (ED Fig. 4e), proteins were eluted from the magnetic beads with Laemmli buffer. Sec23a is the marker for ribosome-free transitional face of the endoplasmic reticulum (ER) and associated vesicles. Calreticulin is a marker for ER resident proteins (n = 2). **b**. Representative images from videos of RFP-LC3 stable HeLa cell line 24 hours after transfection with GFP-ATG2A. The images are from videos taken in one representative of 3 biologically independent experiments. **c**. HeLa cells transfected with pHAGE-GFP-ATG2AmLIR and pHAGE-GFP-ATG2A were treated with DMSO or Lip-1 for 24 hours in media containing Incucyte Cytotox Red Dye (n = 3). Images were taken 30 hours after first transfection. Dead cells were counted as red objects and the number of transfected cells were counted as green objects. Datapoints represent the ratio of red object count vs. the green object count normalized to the transfection efficiency of each sample. At least 30 images were taken per sample per experiment. Protein levels of overexpressed wildtype ATG2A and ATG2AmLIR are shown in the blots. **d**. Representative super-resolution structured illumination microscopy images of endogenous ATG2A co-stained with mitochondria marker TOMM20 and ER protein Sec61-BFP.

Left hand panels are unprocessed and other panels are rendered using Imaris software. Scale bar in the left 4 panels = 0.4 μm, scale bar in the right hand panels = 0.2 μm. This shows increased ATG2 on TOMM20 (mitochondria) in WIPI4 knockout cells. In control cells, 471/2531 ATG2A structures colocalised with mitochondria, while in WIPI4 knockout cells, 407/1438 ATG2A structures colocalised with mitochondria (p = 1.52216E-12 by chi-squared test). 5 cells were scored per condition. **e**. Representative confocal images of endogenous ATG2A co-stained with Mitotracker Far-red and ER protein Sec61-BFP. **f**. Quantification of Pearson's coefficient of ATG2A and Mitotracker Far-red from confocal images (n = 3, >= 20 images per experiment). Scale bar = 10 μm. **g**. Live imaging of HeLa cells transfected with GFP-ATG2A and stained with BodipyC12 (n = 3). Scale bar = 5 μm. Two-tailed paired t-test. **h**. The ATG2A levels in Fig. 4b were compared based on the proportion in mitochondrial fraction to post-mitochondrial fraction (n = 4). Two-tailed one sample *t*-test to control and two-tailed paired *t*-test between other samples. Graphs c, f, h present normalized mean ± SD. Graph in g presents mean ± SD. N represents biologically independent experiments. Source numerical data and unprocessed blots are provided.

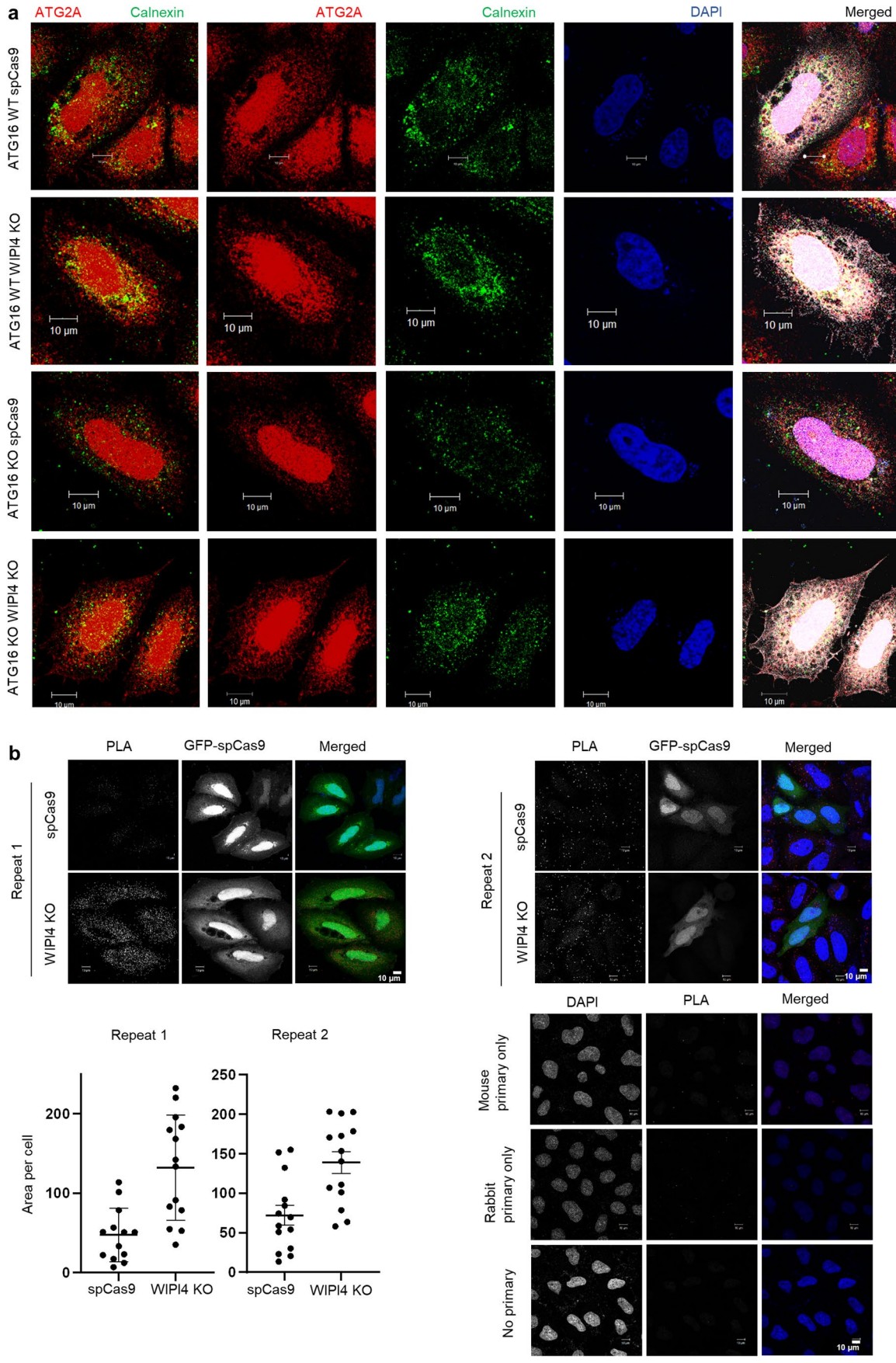

**Extended Data Fig. 6 | See next page for caption.**

**Extended Data Fig. 6 | ATG2A mislocalises more onto ER–mitochondria contact sites in WIPI4 depleted conditions. a**. Confocal images of endogenous ATG2A co-stained with MAM marker calnexin in wildtype and ATG16 knockout HeLa cells depleted of WIPI4. These images are from one representative experiment out of 3 biological repeats. **b**. N = 2 biologically independent experiments of proximity ligation assay for VDAC1 and IP3R1 proteins (>= 15 images were taken per sample per experiment). After fixation and blocking, HeLa cells were incubated overnight with rabbit anti-IP3R3 1:100 and mouse anti-VDAC1 1:100 primary antibodies. After PBS wash, PLA probes anti-rabbit PLUS and anti-mouse MINUS were added for 1 hour in the antibody diluent buffer and then stained with Duolink fluorophore red (excitation/emission = 594/624). As a negative control, one or both of the primary antibodies was/were omitted, and the PLA probe background staining was imaged. The background staining was limited for both probes. The average area of PLA signal in cells positive of GFP-spCas9 was quantified using ImageJ. Each datapoint represents the average area (in arbitrary units) of PLA signal per GFP-spCas9 positive cell in the given image. Source numerical data are provided.

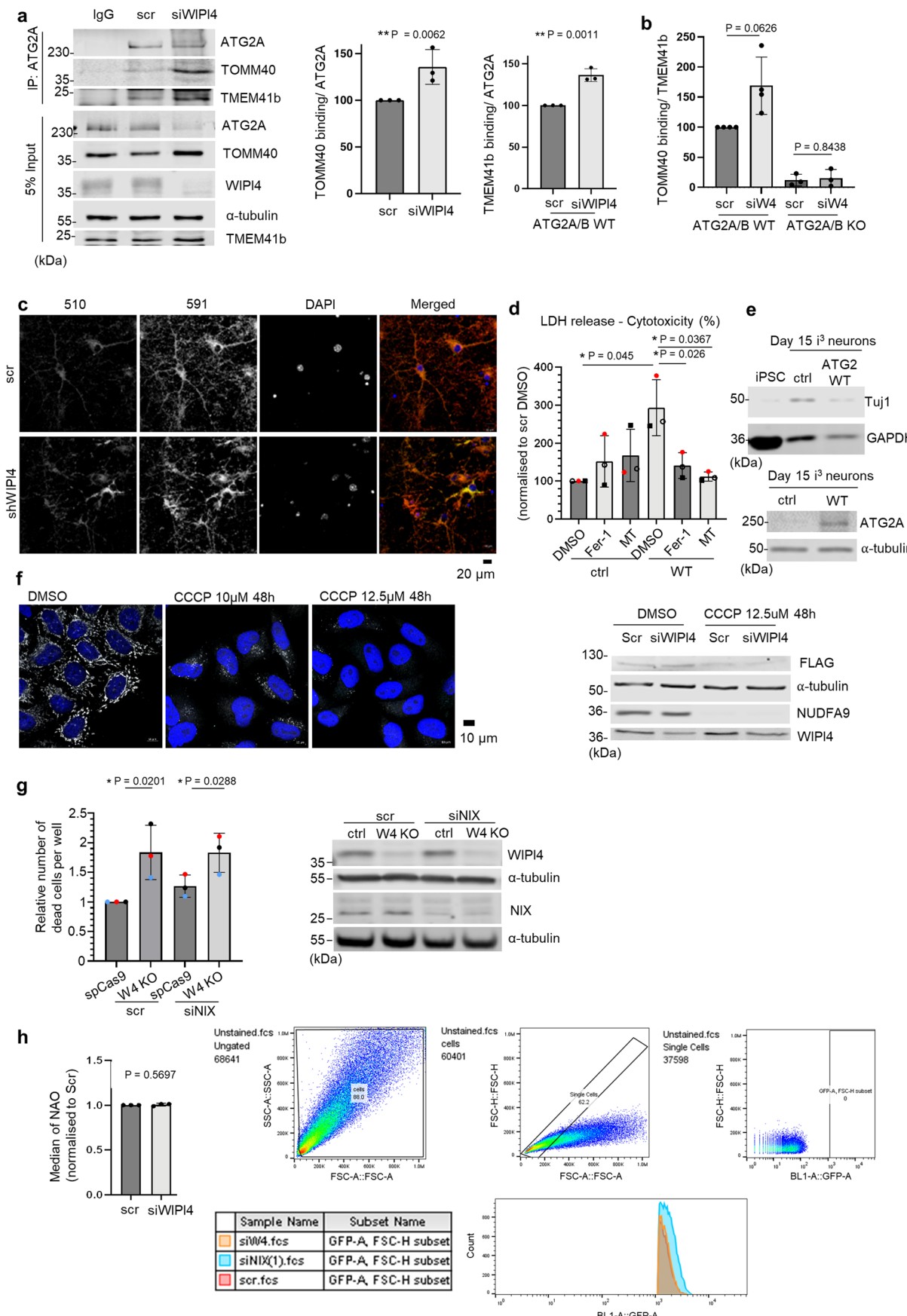

**Extended Data Fig. 7 | See next page for caption.**

**Extended Data Fig. 7 | ATG2A mislocalization induced mitochondrial lipid peroxidation. a**. HeLa cells transfected with control or WIPI4 siRNAs were immunoprecipitated with endogenous ATG2A antibody and blotted for TOMM40 and TMEM41b (n = 3). Data points represents TOMM40 or TMEM41B levels normalized to ATG2A. **b**. ATG2A/B WT or KO HeLa cells transfected with control or WIPI4 siRNAs were immunoprecipitated with endogenous TMEM41b antibody and blotted for ATG2A and TOMM40 (WT cells: n = 4; ATG2A/B KO cells: n = 3). Data points are TOMM40 levels normalized to TMEM41b. **c**. Representative confocal images showing the MitoBodipy$_{581/591}$C11 staining of primary neurons infected with scramble or anti-mouse WDR45 shRNA lentiviruses. These images are from one representative experiment out of 3 biological repeats. **d**. Day 13 i$^3$ neurons were infected with lentiviruses expressing the empty backbone construct or wildtype ATG2A for 24 hours (n = 3). Neurons were then treated with 1 μM Fer-1 or 10 μM Mitotempo (MT) for 24 hours in fresh media before cytotoxicity assay on Day 15. **e**. i$^3$ neurons in d were then harvested and blotted to check the differentiation efficiency and overexpression of ATG2A (n = 3). These blots are from one representative experiment out of 3 biological repeats. **f**. Representative confocal images of PARKIN-FLAG stable HeLa cells from Fig. 5i showing the mitochondrial depletion using MitoTracker Far-red staining. Cells were also lysed and immunoblotted with mitochondria marker NUDFA9 to check the depletion of mitochondria. These images are from one representative experiment out of 3 biological repeats. **g**. Cell death levels measured as dead cell numbers counted as red objects (n = 3). HeLa cells transiently knocked out of WIPI4 were transfected with 50 nM of NIX or scramble siRNA oligos 24 hours later and then grown for another 24 hours before staining with Incucyte Cytotox Red Dye. Images were taken between 48-72 hours after transfection of gRNAs. Western blots show the silencing efficiency of WIPI4 and NIX. Here we have used 1 as an arbitrary number for normalizing the control data to account for variability between biological replicate experiments. The number of dead cells per well in the control ranges from 400 to 1200 and this is a function of the variability in the numbers of cells initially seeded in the wells in the different experiments. **h**. Mitochondrial abundance as measured by flow cytometry with the intensity of nonylacridine orange (NAO) staining (n = 3). HeLa cells were transfected with indicated siRNAs and grown for 24 hours before analysis. The histogram below shows the intensity of green signal of a representative experiment. Anti-NIX siRNA transfected HeLa cells were used as a positive control for increase of mitochondria mass. Data and error bars in this graph are normalized mean ± SD. Two-tailed one sample t-test to control and two-tailed paired t-test between other samples. N represents the number of biologically independent experiments. Source numerical data and unprocessed blots are provided.

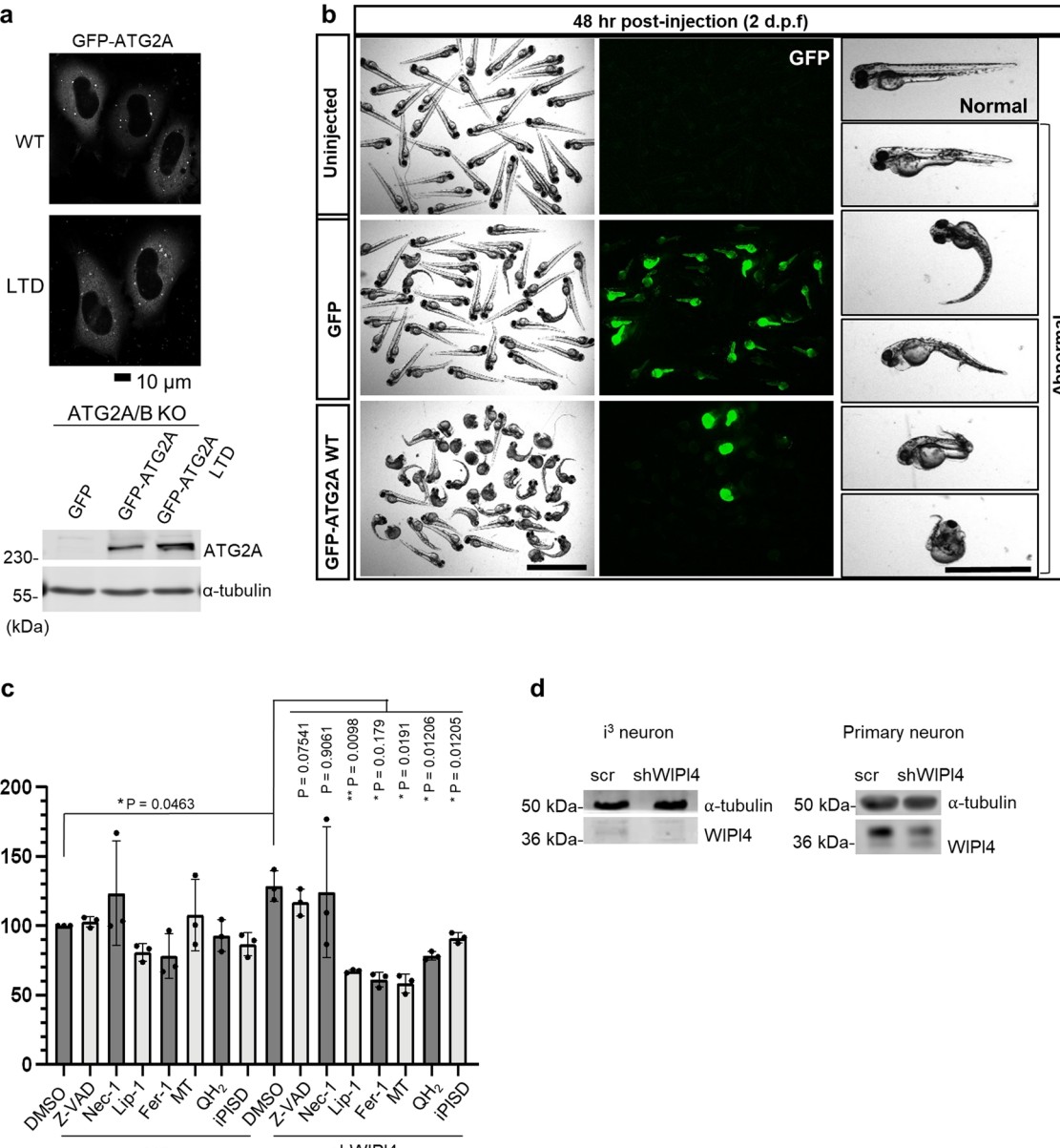

**Extended Data Fig. 8 | ATG2 overexpression induces toxicity through its lipid transfer function. a**. Representative GFP immunofluorescence images and western blots in ATG2A/B KO HeLa transfected with indicated constructs showing that the localization and expression of GFP-ATG2-LTD is similar with GFP-ATG2A-WT. Scale bar: 10 μm. This experiment was done twice. **b**. Representative brightfield and fluorescence images of the phenotypic abnormalities found in clutches injected either with 200 pg GFP empty vector or GFP-ATG2A WT constructs compared to their uninjected siblings at 48 h.p.f. Scale bars: 2 mm for clutches, 1 mm for individual embryos. N = 4 independent experiments. **c**. Primary neurons were pretreated with the following drugs: DMSO (control),

10 μM Z-VAD-fmk, 20 μM necrostatin, 500 nM liproxstatin-1 (Lip-1), 5 μM ferrostatin-1 (Fer-1), 10 μM Mitotempo (MT), 10 nM ubiquinol (QH2), 5 μM PISD inhibitor (iPISD) for 12 hours before 24-hour scramble and smartpool shWIPI4 lentivirus treatment together with the above drugs. Cell cytotoxicity (n = 3 biologically independent experiments) was measured by LDH release assay. Data are presented as normalized mean ± SD. ns., not significant, *P < 0.05, **P < 0.005. Two-tailed one-sample t-test when compared with scr DMSO and two-tailed paired t-test with shWIPI4 DMSO. **d**. The knockdown efficiency of WIPI4 (n = 3 independent experiments) in Fig. 6j and Extended Data Fig. 8c. Source numerical data and unprocessed blots are provided.

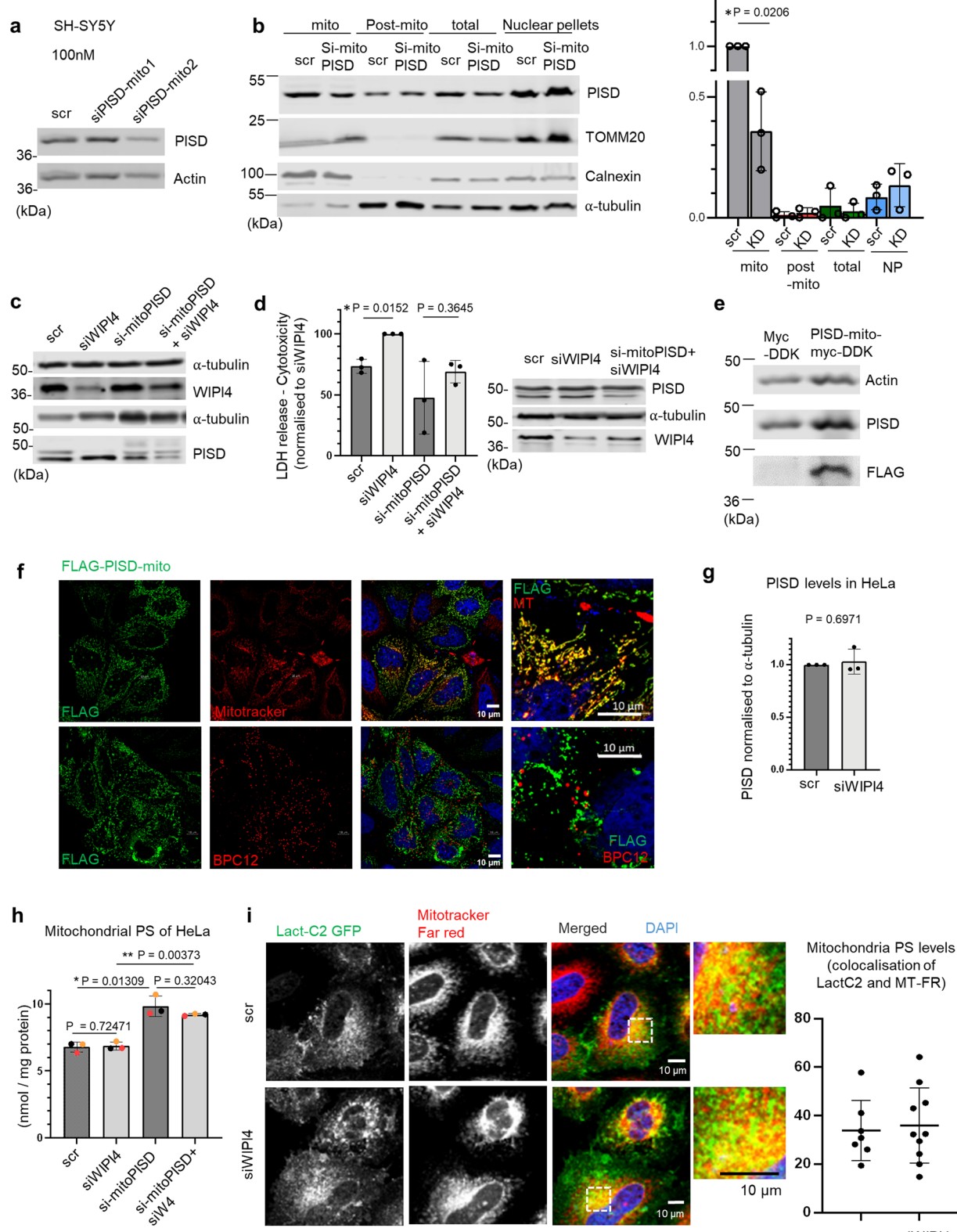

**Extended Data Fig. 9 | See next page for caption.**

**Extended Data Fig. 9 | ATG2 transports PS onto mitochondria, which is then locally converted to PE. a.** The blots show the knockdown efficiency with two siRNA oligos targeting mitochondria specific form of PISD (mito-PISD) in SH-SY5Y cells (n = 3). Oligo 2 was used for the following experiments because it's of better knockdown efficiency. **b.** The silencing efficiency of anti-mitoPISD siRNA in mitochondria fractions, post-mitochondria fractions, total lysates and nuclear pellets (NP) of HeLa cell lysates. HeLa were transfected with 100 nM of scramble or anti-mitoPISD (#2) siRNA oligos and grown for 48 hours before fractionation. TOMM20, the marker of the mitochondrial fraction; calnexin, the MAM enriched fraction. Datapoints represent the PISD protein levels normalized to the loading (α-tubulin) in each fraction (n = 3). **c.** The blots show the knockdown efficiency of WIPI4 and PISD in Fig. 7a (representative blots of 3 biological replicates). **d.** SH-SY5Y cells were transfected with scramble, WIPI4 or si-mitoPISD-2 siRNAs and grown for 48 hours before LDH (n = 3) was measured. Mean (normalized to siWIPI4) ± SD. Blots on the right show the knockdown efficiency of WIPI4 and PISD (representative blots of 3 biological replicates). **e.** Expression levels of mitoPISD-Myc-DDK measured by staining of anti-FLAG and anti-PISD antibodies

(n = 3). HeLa cells transfected with CMV-myc-DDK or mitoPISD-myc-DDK were grown for 24 hours before lysis. **f.** Confocal images showing the colocalisation of mitoPISD-Myc-DDK with Mitotracker Red and BPC12 (neutral lipid/ lipid droplet probe). HeLa cells transfected with CMV-myc-DDK or mitoPISD-myc-DDK were grown for 24 hours before fixation and immunostaining. These images are from one representative experiment of 3 biological repeats. **g.** PISD levels relative to α-tubulin in HeLa cells lysates measured by western blotting (n = 3). **h.** PS levels of mitochondria purified from HeLa cells 48 hours after transfection of anti-WIPI4 and anti-mito-PISD-2 siRNAs (n = 3). **i.** Representative live-imaging experiment showing the colocalization of transiently overexpressed Lactadherin-C2 GFP and Mitotracker Far-red in HeLa cells (n = 1). Datapoints represent the percentage of Lactadherin C2 GFP area positive for Mitotracker Far-red as a function of total Lactadherin-C2 GFP area. Data and error bars in b, d & g: normalized mean ± SD. Two-tailed one sample t-test to scramble and two-tailed paired t-test between other samples; in h: Mean ± SD. Two-tailed paired t-test. Source numerical data and unprocessed blots are available in source data. N = number of biologically independent experiments.

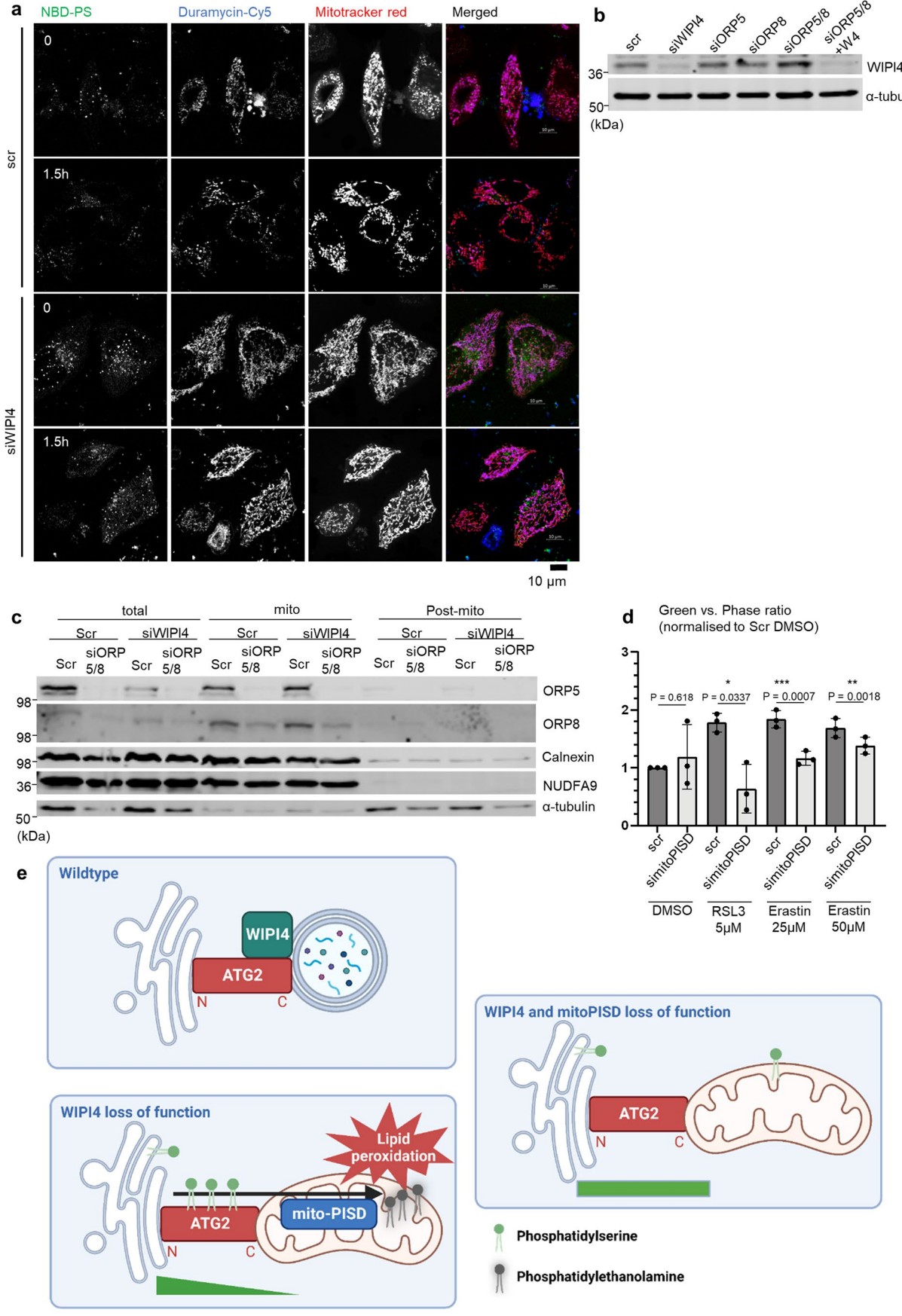

**Extended Data Fig. 10 | See next page for caption.**

**Extended Data Fig. 10 | PS transport onto mitochondria is required for WIPI4–ATG2-dependent ferroptosis. a**. Confocal images of the de novo mitochondrial PE-synthesis measurement experiment with ATG2A/B wildtype cells. Scale bar = 10 μm. Time 0: images taken immediately after 50 minutes' loading of NBD-PS liposomes. Time 1.5 h, images taken 1.5 hours later, including 40 minutes incubation in 37 °C and 50 minutes of restaining of Duramycin-Cy5. These images are from one representative experiment out of 3 biological repeats. **b**. Representative blots showing the silencing efficiency of WIPI4 in samples (n = 3 independent experiments). **c**. Representative blots showing the silencing efficiency of ORP5, ORP8 and enrichment of MAM marker calnexin (n = 3 independent experiments). **d**. SH-SY5Y cells were transfected with 100 nM of scramble or anti-mitoPISD (#2) siRNA oligos. Cells were treated with drugs of indicated concentration in media containing CellTox Green Dye from 48 hours post first transfection. Images were taken at 60 hours post transfection. The

levels of cell death were quantified as the ratio of green (dead cells) area vs. phase (total cells) area using the Incucyte 2020 software. Normalized mean ± SD (n = 3 biologically independent experiments). Two-tailed one sample t-test to scramble DMSO control and two-tailed paired t-test for other samples. **e**. Diagram. WIPI4 stabilizes the C-terminus of ATG2 onto autophagosomes by direct interaction. Without WIPI4 the C-terminus of ATG2 localizes more onto mitochondria, and this mislocalisation leads to more transfer of PS into mitochondria and more conversion to PE by mitochondrial PISD thereafter. The increase in mitochondrial PE levels promotes ferroptosis, possibly because poly unsaturated PE is the main substrate of ferroptosis. When mitochondrial PISD is depleted, the ER mitochondria concentration gradient of PS disappears, which slows the ER mito PS transfer and ferroptosis. Source numerical data and unprocessed blots are available in source data.

# Reporting Summary

## Statistics

For all statistical analyses, confirm that the following items are present in the figure legend, table legend, main text, or Methods section.

| n/a | Confirmed | |
|---|---|---|
| ☐ | ☒ | The exact sample size (*n*) for each experimental group/condition, given as a discrete number and unit of measurement |
| ☐ | ☒ | A statement on whether measurements were taken from distinct samples or whether the same sample was measured repeatedly |
| ☐ | ☒ | The statistical test(s) used AND whether they are one- or two-sided *Only common tests should be described solely by name; describe more complex techniques in the Methods section.* |
| ☒ | ☐ | A description of all covariates tested |
| ☒ | ☐ | A description of any assumptions or corrections, such as tests of normality and adjustment for multiple comparisons |
| ☐ | ☒ | A full description of the statistical parameters including central tendency (e.g. means) or other basic estimates (e.g. regression coefficient) AND variation (e.g. standard deviation) or associated estimates of uncertainty (e.g. confidence intervals) |
| ☐ | ☒ | For null hypothesis testing, the test statistic (e.g. *F*, *t*, *r*) with confidence intervals, effect sizes, degrees of freedom and *P* value noted *Give P values as exact values whenever suitable.* |
| ☒ | ☐ | For Bayesian analysis, information on the choice of priors and Markov chain Monte Carlo settings |
| ☒ | ☐ | For hierarchical and complex designs, identification of the appropriate level for tests and full reporting of outcomes |
| ☐ | ☒ | Estimates of effect sizes (e.g. Cohen's *d*, Pearson's *r*), indicating how they were calculated |

*Our web collection on statistics for biologists contains articles on many of the points above.*

## Software and code

Policy information about availability of computer code

| | |
|---|---|
| Data collection | Confocal images were collected with ZEN Black 2.6 Carl Zeiss Microscopy. Superresolution images were acquired with Zen 20 12 Elyra edition software. Plate reader Tecan SparkControl V3.0. The IncucyteS3 incubated live imaging system. Images of zebrafish cryosections were taken by the QImaging Retiga 2000 R digital camera using Qcapture software. Live imaging of zebrafish larvae injected with ATG2A constructs were taken using Leica Application Suite X (LAS X) software Attune'" NxT BD FACSDiva'M Software. |
| Data analysis | IMAGE STUDIO Lite LI-COR ver 5.2, Inc and Image J (National Institute of Health, USA) for gel analysis. ZEN imaging software (ZEN Black 2.3 Carl Zeiss Microscopy) and Image J ver 1.54f for microscopic image analysis. Volocity 6.3 Software (PerkinElmer) for Pearson's correlation coefficient (PCC). Microsoft Excel (Excel 2016 Microsoft office) and GraphPad Prism v9 (GraphPad Software) for statistical analysis. Imaris image analysis software Incucyte 2020 software FlowJo vl0.8 SparkControl V3.0 LightCycler® 480 Software (Vl.5.1.62) |

For manuscripts utilizing custom algorithms or software that are central to the research but not yet described in published literature, software must be made available to editors and reviewers. We strongly encourage code deposition in a community repository (e.g. GitHub). See the Nature Portfolio guidelines for submitting code & software for further information.

## Data

Policy information about availability of data

All manuscripts must include a data availability statement. This statement should provide the following information, where applicable:
- Accession codes, unique identifiers, or web links for publicly available datasets
- A description of any restrictions on data availability
- For clinical datasets or third party data, please ensure that the statement adheres to our policy

All data supporting the findings of this study are available from the corresponding author upon reasonable request. Source data are provided with this paper.

## Research involving human participants, their data, or biological material

Policy information about studies with human participants or human data. See also policy information about sex, gender (identity/presentation), and sexual orientation and race, ethnicity and racism.

| | |
|---|---|
| Reporting on sex and gender | not applicable |
| Reporting on race, ethnicity, or other socially relevant groupings | not applicable |
| Population characteristics | not applicable |
| Recruitment | not applicable |
| Ethics oversight | not applicable |

Note that full information on the approval of the study protocol must also be provided in the manuscript.

# Field-specific reporting

Please select the one below that is the best fit for your research. If you are not sure, read the appropriate sections before making your selection.

☒ Life sciences   ☐ Behavioural & social sciences   ☐ Ecological, evolutionary & environmental sciences

For a reference copy of the document with all sections, see nature.com/documents/nr-reporting-summary-flat.pdf

# Life sciences study design

All studies must disclose on these points even when the disclosure is negative.

| | |
|---|---|
| Sample size | Sample sizes were chosen on the basis of extensive experience with the assays we have performed. (e.g. Wrobel et al. Nature Communications 2022) |
| Data exclusions | No data were excluded from the analysis. |
| Replication | Mammalian: Cell death, viability, MDA assays, and western blots, usually 3 biological repeats were done for each experiment. For immunofluorescence experiments, 5-50 images were taken for each biological repeat and 3 biological repeats were done for each experiment. Zebrafish: For the analysis of images of fish retinal degeneration a minimum of 12 eyes per group were quantified. For the phenotypic analysis of fish after ATG2A, a minimum of 4 set of injections for each condition was analysed. For MDA analysis, sample size corresponds to 5 pools of 10 fish each per condition. Expression was analysed in 7 samples (10 fish each) per condition. Survival analysis was done using 60 fish per group. Each replicate was perform at different time. |
| Randomization | All in vivo experiments and tests were randomly assigned, but no randomization was performed for cell culture experiments. |
| Blinding | Immunofluorescence analysis was blinded when possible. Western blot analysis was not blinded as it was not possible as the gel loading order needs to be defined. Investigators were not blinded during the other experiments. |

# Reporting for specific materials, systems and methods

We require information from authors about some types of materials, experimental systems and methods used in many studies. Here, indicate whether each material, system or method listed is relevant to your study. If you are not sure if a list item applies to your research, read the appropriate section before selecting a response.

## Materials & experimental systems

| n/a | Involved in the study |
|---|---|
| ☐ | ☒ Antibodies |
| ☐ | ☒ Eukaryotic cell lines |
| ☒ | ☐ Palaeontology and archaeology |
| ☐ | ☒ Animals and other organisms |
| ☒ | ☐ Clinical data |
| ☒ | ☐ Dual use research of concern |
| ☒ | ☐ Plants |

## Methods

| n/a | Involved in the study |
|---|---|
| ☒ | ☐ ChIP-seq |
| ☐ | ☒ Flow cytometry |
| ☒ | ☐ MRI-based neuroimaging |

# Antibodies

| | |
|---|---|
| Antibodies used | Primary antibodies:<br>Antibodies for western blots:<br>mouse anti-Flag [M2] (#F3165, RRID:AB_262044, WB 1:2000),<br>rabbit anti-WIPl4( Cat# 19194-1-AP) from Proteintech, RRID:AB_2215404<br>mouse anti-α-Tubulin [DM1A] (Cat# T9026) from Sigma-Aldrich, RRID:AB_477593<br>rabbit anti-actin (#A2066) from Sigma Aldrich, RRID:AB_476693<br>rabbit anti-ATG2A (PD041) from MBL Life Science, RRID:AB_2810871<br>rabbit anti-GFP(ab6556) from Abcam, RRID:AB_305564<br>mouse anti-NDUFA9 [20C11B11B11] (ab14713) from Abcam, RRID:AB_301431<br>rabbit anti-Sec23a (ab137583) from Abcam<br>mouse anti-Tom40 [D-2] (sc-365467) from Santa Cruz Biotechnology, RRID:AB_10847086<br>rabbit anti-TMEM41b (NBPl-81552) from Novus Biologicals, RRID:AB_11015584<br>rabbit anti-Calreticulin (#12238) from Cell Signalling, RRID:AB_2688013<br>rabbit anti-cleaved Caspase-3 (#9661) from Cell Signalling, RRID:AB_2341188<br>rabbit anti-ATG16L1 (#8089) from Cell Signalling, RRID:AB_10950320<br>rabbit anti-ATG7 (#2631) from Cell Signalling, RRID:AB_2227783<br>rabbit anti-TOMM20 (ab186735) from Abcam, RRID:AB_2889972<br>mouse anti-GM130 [EP892Y] (ab52649) from Abcam, RRID:AB_880266<br>rabbit anti-LAMP1 (ab24170) from Abcam, RRID:AB_775978<br>rabbit anti-KDEL (ab2898) from Abcam, RRID:AB_303392<br>rabbit anti-PISD (HPA031091), from Atlas antibodies, RRID:AB_10600893<br>mouse anti-beta Ill Tubulin antibody [2G10] (ab78078) from Abcam, RRID:AB_2256751<br>rabbit anti-MAP2 antibody (8707S), from Cell signalling, RRID:AB_10693782<br>rabbit anti-NIX (12396), from cell signalling, RRID:AB_2688036<br>mouse anti-ORP8 [PL-C26] (sc-134409), from Santa Cruz, RRID:AB_2156227<br>rabbit anti-ORP5 (HPA038712), from Atlas Antibodies, RRID:AB_10675949<br>rabbit anti IP3 Receptor 1 (D53A5)(#8568), from Cell signalling, RRID:AB_10890699<br>mouse anti VDAC1 [20B12AF2] (ab14734), from Abcam, RRID:AB_443084<br>Antibodies for immunofluorescence:<br>mouse anti-TOMM20 F-10 (sc-17764) from Santa Cruz, RRID:AB_628381<br>mouse anti-Calnexin [6F12BE10] (ab112995), from Abcam, RRID:AB_10860712<br><br>Secondary Antibodies: anti-mouse (#NA931V, RRID:AB_772210) and anti-rabbit (#NA934V) horseradish peroxidise (HRP)-conjugated secondary antibodies (GE Healthcare); anti-goat horseradish peroxidise (HRP)-conjugated secondary antibody (#611620, RRID:AB_87867, Invitrogen/Life Technologies). For immunoflourescence, goat-anti-mouse Alexa Fluor 488 (#A11029, RRID:AB_2534088, 1:400), 555 (#A21147, RRID:AB_1500897, 1:400) and 594 (#A11032, RRID:AB_2534091, 1:400), goat-anti-rabbit Alexa Fluor 488 (#A32731, RRID:AB_2633280, 1:400) and 555 (#A21428, RRID:AB_141784, 1:400) from ThermoFisher Scientific. |
| Validation | All antibodies used in this study were purchased from commercial vendors who had validated specificity in human cells/ mouse tissues for the specific assays (Western blot, immunoprecipitation and/or immunofluorescence). It is described on data sheets and online.<br>rabbit anti-Sec23a (ab137583) from Abcam was also validated in Lassalle Set al., Oncotarget, 2016. PMC: 27036030 |

# Eukaryotic cell lines

Policy information about cell lines and Sex and Gender in Research

| | |
|---|---|
| Cell line source(s) | Human cervical epithelium HeLa (ATCC; #CCL-2; CVCL_0030), human neuroblastoma SH-SY5Y (ECACC; #94030304), human embryonic kidney cell line HEK293 (ECACC; #85120602)<br>CRISPR/Cas9 ATG2A/B double knockout cell line, from Mizushima's lab<br>The human iPS WT line (KOLF-2) was generated by the Sanger Wellcome Institute Induced Pluripotent Stem Cell Initiative (HipsSci).<br>ATG16L1 knockout HeLa and its control were made in house.<br>Beclin 1 knockout HeLa and its control, from Wensheng Wei's lab in Beijing |
| Authentication | The cell lines were ordered from ATCC, Horizon or Coriell Institute with authentication. |

| Authentication | HeLa authenticatd by ATCC (by Short Tandem Repeat (STR) profiling; FTA barcode:STRA1466)<br>SH-SY5Y authenticatd by LGC (STR profiling, FTA barcode:STRA1440)<br>HEK293 authenticatd by LGC (STR profiling, FTA barcode:STRA1472)<br>CRISPR/Cas9 ATG2A/B double knockout cell line (Tamura, N. et al. EBS Lett, 2017)<br>The information about the human iPS WT line including the Certificate of Analysis can be found on the website (www.hipsci.org).<br>ATG16L1 knockout HeLa and its control (Bento, C. F. et al., Nature communication,2016)<br>Beclin1 knockout and its control (He, R. et al, Autophagy, 2015) |
|---|---|
| Mycoplasma contamination | The cells were regularly tested using EZ-PCR Mycoplasma Test Kit (Biological Industries; cat#20-700-20) and Mycrostripl00 (lnvivoGen- rep-mys-100). Cells used in this study were mycoplasma negative. |
| Commonly misidentified lines<br>(See ICLAC register) | no commonly misidentified cell lines were used in the study. |

# Animals and other research organisms

Policy information about studies involving animals; ARRIVE guidelines recommended for reporting animal research, and Sex and Gender in Research

| Laboratory animals | Transgenic Zebrafish line (from Zebrafish Information Network (ZFIN)): Tg2(rho:EGFP)cu3, RRID: ZFIN_ZDB-ALT-101103-1 Wiltype TL.<br>Adult fish of between 6 months and 18 months old were bred to generate embryos and larvae for the experiments described below.<br>Zebrafish from 0 - 7 weeks old were used for Fig 2a (survival assay, Protocol 7).<br>Zebrafish larvae from 0 - 10 days post-fertilisation (10 d.p.f.) were used for Fig 2 b& c.<br>Zebrafish larvae from 0 -5 d.p.f. were used for Fig 2d.<br>Zebrafish larvae from 0 - 10 d.p.f. were used for Fig 3c& d.<br>Zebrafish larvae at 5 d.p.f. were used for Extended Fig 1l.<br>Zebrafish larvae from 0 - 10 d.p.f. were used for Extended Fig 1m.<br>Zebrafish larvae from 0 - 2 d.p.f. were used for Fig 6i, Extended Fig 8b. |
|---|---|
| Wild animals | No wild animals were used in the study. |
| Reporting on sex | For zebrafish sex cannot be differented at the stages experiments were done. |
| Field-collected samples | No field collected samples were used in the study. |
| Ethics oversight | All zebrafish experiments were performed in accordance with the UK Animals (Scientific Procedures) Act with appropriate Home Office Project and Personal animal licenses and with local Ethics Committee approval. Studies were performed in accordance with PREPARE and ARRIVE guidelines. |

Note that full information on the approval of the study protocol must also be provided in the manuscript.

# Flow Cytometry

## Plots

Confirm that:

☒ The axis labels state the marker and fluorochrome used (e.g. CD4-FITC).

☒ The axis scales are clearly visible. Include numbers along axes only for bottom left plot of group (a 'group' is an analysis of identical markers).

☐ All plots are contour plots with outliers or pseudocolor plots.

☒ A numerical value for number of cells or percentage (with statistics) is provided.

## Methodology

| Sample preparation | The cell line used for BPC11 staining is SH-SY5Y cells detailed in above sections. Cells were stained for 30 minutes in growth media supplemented with 5 µM C11-Bodipy 581/591. All cells in the media and lifted are subject to analysis. The cell line used for NAO staining was HeLa. |
|---|---|
| Instrument | Becton Dickinson LSR Fortessa |
| Software | FLowJo TM Vl0.8 |
| Cell population abundance | At least 50000 cells were analysed for each sample. |
| Gating strategy | For BPC11 staining, first, SH-SY5Y cells were isolated using forward scatter (FSC) and side scatter (SSC) properties. Live cells were distinguished from debris that are of lower FSC and SSC. Then, single cell population was defined by comparing the area of forward scatter (FSC-A) to the height of forward scatter (FSC-H). Cells in which FSC-A is not correlated with FSC-H are likely to be doublets and were therefore excluded from analysis. Last, SH-SY5Y cells positive of BodipyCll staining was defined by thresholding against unstained SH-SY5Y cells. Cells with greater red (YLl-A) fluorescence than that of unstained cells (YLl-A+) were selected as positive of BodipyCll staining (Q2 and Q3). The autofluorescnce of cells was also gated out by thresholding |

against unstained SH-SY5Y cells. Cells in Q2 are positive of both reduced and oxidised BodipyC11 signal and are used for quantitative analysis. For NAO staining, median fluorescence intensity analysis of labelled mitochondria was performed by gating on single cells.

☒ Tick this box to confirm that a figure exemplifying the gating strategy is provided in the Supplementary Information.

