## [Peer Review File · Nature Cell Biology]

Peer Review Information

Journal: Nature Cell Biology

Manuscript Title: Loss of WIPI4 in neurodegeneration causes autophagy-independent ferroptosis

Corresponding author name(s): Professor David Rubinsztein

Editorial Notes:

Reviewer Comments & Decisions:

Decision Letter, initial version:
--

*Please delete the link to your author homepage if you wish to forward this email to co-authors.

Dear Professor Rubinsztein,

Thank you for submitting your manuscript, "WIPI4 depletion associated with Beta-propeller protein-associated neurodegeneration causes autophagy-independent PE-dependent ferroptosis", to Nature Cell Biology. The manuscript has now been seen by 3 referees, who are experts in lipid transfer and membrane contact sites (Referee #1); ferroptosis (Referee #2); and autophagy (Referee #3). As you

will see from their comments (attached below), they found this work of potential interest but have raised substantial concerns, which in our view would need to be addressed with considerable revisions before we can consider publication in Nature Cell Biology.

As per our standard editorial process, we have now discussed the referee reports in detail within the editorial team, including the chief editor, to identify key referee points that should be addressed with priority, as opposed to requests that are beyond the scope of the current study. To guide the scope of the revisions, I have listed these points below. Our standard revision period is six months, and we are committed to providing a fair and constructive peer-review process, so please feel free to contact me if you would like to discuss any of the referee comments further or if you anticipate any delays or issues addressing the reviews.

As you will see, the reviewers (and we agree) found the model for WIPI4 function in controlling ATG2 localization at different contact sites and in autophagy-independent, PE-dependent ferroptosis very interesting. However, they also felt that key aspects of the model needed to be better supported to convince experts in these fields. In particular, we feel it would be essential to dedicate experimental efforts in revision to address the following points:

- Please further test the possibility that the effects of WIPI4 are independent of autophagy following the recommendations of Rev#3 in points #2 through #6
- The reviewers feel that strengthening the link between WIPI4 and the localization of ATG2 as well as further assessing the potential role for other PS transfer proteins are important parts of the revision:
Rev#1 point #2
Rev#2 point #3
Rev#3 point #1
- Rev#1 and Rev#2 feel that further analyses of the link between PE and ferroptosis are needed to maintain the core conclusions and that the nature of the cell death process requires further investigation:
Rev#1, points #3, 6
Rev#2 points #1, 2, 4, 5, 6
- Please also address the referees' suggestions for controls, methodological details, textual edits, and technical points about the existing dataset.
- Finally, please pay close attention to our guidelines on statistical and methodological reporting (listed below) as failure to do so may delay the reconsideration of the revised manuscript. In particular, please provide:
 - a Supplementary Figure including unprocessed images of all gels/blots in the form of a multi-page pdf file. Please ensure that blots/gels are labeled and the sections presented in the figures are clearly indicated.
 - a Supplementary Table including all numerical source data in Excel format, with data for different figures provided as different sheets within a single Excel file. The file should include source data giving rise to graphical representations and statistical descriptions in the paper and for all instances where the figures present representative experiments of multiple independent repeats, the source data of all

repeats should be provided.

We would be happy to consider a revised manuscript that would satisfactorily address these points, unless a similar paper is published elsewhere, or is accepted for publication in Nature Cell Biology in the meantime.

- ensure that it conforms to our format instructions and publication policies (see below and <https://www.nature.com/nature/for-authors>).
- provide a point-by-point rebuttal to the full referee reports verbatim, as provided at the end of this letter.
- provide the completed Reporting Summary (found here <https://www.nature.com/documents/nr-reporting-summary.pdf>). This is essential for reconsideration of the manuscript will be available to editors and referees in the event of peer review. For more information see <http://www.nature.com/authors/policies/availability.html> or contact me.

When submitting the revised version of your manuscript, please pay close attention to our [href="https://www.nature.com/nature-portfolio/editorial-policies/image-integrity">Digital Image Integrity Guidelines](https://www.nature.com/nature-portfolio/editorial-policies/image-integrity). and to the following points below:

Nature Cell Biology is committed to improving transparency in authorship. As part of our efforts in this direction, we are now requesting that all authors identified as 'corresponding author' on published papers create and link their Open Researcher and Contributor Identifier (ORCID) with their account on the Manuscript Tracking System (MTS), prior to acceptance. ORCID helps the scientific community achieve unambiguous attribution of all scholarly contributions. You can create and link your ORCID from the home page of the MTS by clicking on 'Modify my Springer Nature account'. For more information please visit www.springernature.com/orcid.

This journal strongly supports public availability of data. Please place the data used in your paper into a public data repository, or alternatively, present the data as Supplementary Information. If data can only be shared on request, please explain why in your Data Availability Statement, and also in the correspondence with your editor. Please note that for some data types, deposition in a public

repository is mandatory - more information on our data deposition policies and available repositories appears below.

[Redacted]

We hope that you will find our referees' comments and editorial guidance helpful. Please do not hesitate to contact me if there is anything you would like to discuss. Thank you very much for considering Nature Cell Biology for your work,

Best wishes,

Melina

Melina Casadio, PhD
Senior Editor, Nature Cell Biology
ORCID ID: <https://orcid.org/0000-0003-2389-2243>

Reviewers' Comments:

Reviewer #1:

Remarks to the Author:

WIPI4 depletion leads to an induction of lipid peroxidation and ferroptosis. The authors show that this is through an enhanced association of ATG2 with ER-mitochondrial contacts, enhanced PS delivery to mitochondria, and enhanced mitochondrial PE production through mito-PISD. As such, the authors suggest that WIPI4 plays a role in keeping ATG2 away from MAM, providing a potential autophagy-independent mechanism for WIPI4's role in BPAN. This is a very interesting model.

Whilst the overall quality of the data is good and a range of complementary techniques were used, there are several issues in the manuscript that require rectification and clarification. General comments:

1. The link between WIPI4 mutation and ferroptosis in BPAN is not new (e.g. PMID: 34012978). This manuscript confirmed this link (not 'revealed' as in the abstract).
2. Other PS transfer proteins should also be functioning at ER-mitochondria contacts or mitochondria-associated ER membrane (MAM) (e.g. PMID: PMC9375143). The effect of these proteins and the integrity of MAM - critical for PS synthesis and transfer to mitochondria - should be investigated in WIPI4-depleted cells.
3. If PS to PE conversion drives ferroptosis, what are the effects of blocking PS synthesis in the ER by

interfering with PS synthases?

4. WT cells should be included in the analysis shown in Fig. 4E, F, and G. This is particularly important for demonstrating the PS transfer activity of ATG2 from the ER to mitochondria.
5. WT cells should also be included in Fig. 5D, E, and F.
6. Besides total mitochondria PE, the unsaturation of acyl chains should also be examined. Unsaturated double bonds are more critical for ferroptosis.

Other comments:

1. Line 139, does TMEM41b interact with ATG2 so to keep it at MAM? Ref. 11 does not have the data.
2. Fig S1b, what are the 3 asterisks above the 3rd column referring to?
3. Effects seen in S1i is surprising given the lack of knockdown in S1j (minimal difference between shWIPI4 and the scr control, and yet a marked difference in LDH release). Is this blot representative, also which band is the appropriate band?
4. Bands in Fig 3C are too closely cropped (some bands are nearly excluded) and the expected sizes of each protein should be indicated accordingly to clarify those that have been quantified. Further, there is a clear size shift in WIPI4 in the mitochondrial fraction – authors should comment on this.
5. Figure S5b has cropped blots that are merged into one, these need to be separated and more of the gel shown. Why is TMEM41B IPed in the IgG control lane?
6. In figure 5I, the authors state that “there was also an increase of cytotoxicity in cells infected with wildtype ATG2A lentivirus, which could be rescued by co-treatment with Fer-1; but ATG2A-RLA lentivirus failed to induce cytotoxicity”, however, whilst non-significant, there is a striking trend (2-fold) from the ctrl DMSO to the RLA-infected DMSO condition. Stating this as a failure to induce cytotoxicity appears misleading.

Regarding the methodology, there are some issues too:

1. There are no methods and insufficient information regarding how the LC3 and ATG2A colocalization experiments were quantified.
2. More information is required surrounding the state of the cells in Seahorse-treated conditions. It is unclear whether the cells are dead or not and whether normal enzymatic functions of the cell can still occur. Why does this rule out ER-mitochondrial PS transport? Are the mitochondria still functional? If so, an assay to demonstrate this would be of use, if not, how can we assume mitoPISD retains activity?

Other less important issues include:

1. In Fig. S8, Duramycin is misspelled.
2. Make references to extended figures and Supp figures consistent in their naming.

Reviewer #2:

Remarks to the Author:

In this manuscript, the authors present a comprehensive set of data to support a role of WIPI4 and ATG2A/B, canonical autophagy proteins, in the regulation of ferroptosis, as well as the underlying mechanism as they propose: independently of autophagy, loss of WIPI4 allows ATG2 to transfer phospholipid PS from ER to mitochondria, where PS can be enzymatically converted to PE, followed by PE peroxidation and ferroptosis. This mechanism has potential clinical relevance, as loss of function of WIPI4 is associated with certain specific neurodegenerative diseases.

Major experiments were performed using cancer cell lines, complemented with some studies using iPSC-derived neurons, primary neurons, and zebrafish models. Overall, the study is of excellent quality, and the proposed mechanisms, if sufficiently substantiated, will be highly valuable to the field of ferroptosis. On the other hand, evidence for some key mechanistic conclusions are correlative, the major experimental system (elimination/inhibition of WIPI4 to trigger ferroptosis) is less than optimal, and the authors need to pay more attention to certain experimental design and control. Specifically,

1. The study starts with the observation that siRNA against WIPI4 in several cell lines causes cell death, presumably, ferroptosis. Since the observed cell death is quite modest (around 30% as in Fig. 1 and Fig. S1) compared to other commonly used ferroptosis triggers, this reviewer recommend the authors to establish a more reliable and easy-to-control approach for WIPI4-inhibition-triggered ferroptosis, such as Dox-inducible WIPI4 shRNA using both a cancer cell line and iPSC-derived neurons. Such an inducible system allows for more thorough analysis such as time course and dose analysis (Dox concentration will control the degree of WIPI4 elimination), which is needed to lend more convincing support.

2. About the nature of the modest cell death triggered by WIPI4 elimination, although their pharmacological analysis suggests ferroptosis, the confidence is weakened by their result that NAC, a commonly-used ferroptosis inhibitor, failed to prevent the death (Fig. 1A and Fig. S1D). This might be due to that they were using a much lower NAC concentration in their experiments. To validate it is indeed ferroptosis, in an inducible shRNA system (see point-1), suggest (1) in experiments using NAC, increase NAC dose to 0.2 mM and also include a positive control (for example, erastin as ferroptosis trigger) +/- NAC ; and (2) test whether WIPI4 loss-induced cell death can also be inhibited by iron chelation.

3. A major conclusion is that loss of WIPI4 renders more mitochondrial localization of ATG2, yet data supporting this claim should be strengthened. For example, for Fig. 3c, if one normalizes the loading (using NDUFA, PNS frx vs mito frx), ATG2 amount in mito-enriched frx was very low; also, it seems that WIPI4 depletion reduced total/PNS ATG2 - is this really the case? For Fig. S4C image of ATG2A-mito co-localization, can you do a high-resolution imaging as Fig. S4B? The resolution of Fig. S4C is just too low.

4. In Fig. 5 and S7, etc., authors test the involvement of mito PE synthase (mitoPISD) by using pharmacological agents and siRNA specific to mitoPISD. Based on the presented results, it is hard to appreciate whether mitoPISD but not cytosolic PISD was specifically eliminated by siRNA . What exactly is the assay for this claim?

5. Also for mitoPISD, (1) would miPISD overexpression increase mito PE content and trigger ferroptosis? This seems to be the logical conclusion based on the proposed mechanism. (2) Examine whether mitoPISD modulates the sensitivity to ferroptosis induced by erastin or RSL3. This is important considering the reported role of mitochondria in ferroptosis induced by erastin but not RSL3.

6. Although most of their experimental observations are consistent with the conclusion that WIPI4 elimination triggers ferroptosis via increasing mitochondrial PE contents, all the evidence are correlative and there lacks a causative experiment directly confirming that increased mitochondrial PE triggers ferroptosis (PS liposome experiment is for this purpose, but the experiment may cause too many changes to the cell and as such one cannot attribute the outcome only to mito PE). The authors either need to conduct such an experiment (miPISD overexpression?) or tune down their conclusion.

Minor points:

Please carefully proofread the whole manuscript and the citation of both published literature and your own figures in the text. For example, (1) where are your Fig. S1n and S1o as you cited in the text? (2) There is a wealth of important papers about mitochondria and ferroptosis, and a careful and brief discussion on this can put your study into this relevant context.

Reviewer #3:

Remarks to the Author:

Autophagy maintains neuronal homeostasis by intracellular elimination of damaged material in general, including protein aggregates and damaged mitochondrial fragments in particular, termed aggrephagy and mitophagy, respectively. Accordingly, failure of mitophagy is implicated in the etiology of neurodegeneration, e.g. due to mutations in the mitophagy signaling protein Parkin in Parkinson Disease. However, whether autophagy proteins promote mitochondrial well-being in non-autophagic contexts remains unclear. In their submitted manuscript, Zhu et al. study a model for the neurodegeneration disease BPAN that develops upon loss-of-function of WIPI4, a protein implicated in phagophore expansion and maturation, through autophagic targeting of the ER-phagophore tether and lipid transporter ATG2A.

The authors study the role of WIPI4 in BPAN by employing an in vitro model using siRNA-mediated knockdown of WIPI4 in HeLa and neuroblastoma or iPS-derived neurons and a zebrafish in vivo model. These are augmented by genetic rescue using different WIPI4 constructs or artificial gain of ATG2A function through exogenous expression of different ATG2A constructs. Coupled with autophagic inactivation by knockdown of WIPI2 or Atg7, cytotoxicity due to loss of WIPI4 is elegantly demonstrated to result from induction of ferroptosis. Furthermore, mislocalization of ATG2A from the autophagic context - due to loss of interaction with WIPI4 or upon artificial upregulation - leads to abnormal localization of ATG2A to ER-mitochondrial contact sites. This, in turn, is shown to promote autophagy-independent ferroptosis due to excessive ATG2A-mediated transport of PS from ER to mitochondria and consequent conversion to oxidation-prone mitochondrial PE.

The suggested model posits that in wildtype conditions, WIPI4 shields ATG2A from unchecked mitochondrial-associated activity that is unleashed in BPAN patients, leading to ferroptotic toxicity. Importantly, pharmaceutical inhibition of mitochondrial PS decarboxylase is shown to suppress this toxicity - hinting at a potential therapeutic intervention. The study is well-conducted and provides a new angle on the link between autophagy proteins, iron-related mitochondrial well-being and neuronal diseases. Nevertheless, important controls are warranted to establish more rigorously the model in general and the autophagy-independent activity of ATG2A therein in particular:

1. As duly mentioned, mitochondrial association ATG2A reportedly depends on TOM40 (mediated by TOM70), whereas lipid transport from the ER requires the ER-resident scramblase TMEM41B. Loss-of-function of each should therefore be shown to suppress siWIPI4/GFP-ATG2A-mediated cytotoxicity. In contrast, exogenous expression of the LIR-deficient ATG2A (mLIR, PMID: 32009292) and ATG9A-inert ATG2A (Δ 237-431, PMID: 31412244) should be shown to be at least as toxic as WT ATG2A in a manner that is suppressed by ferroptosis inhibitors - in presence or absence of WIPI4.

2. The lack of toxicity upon knockdown of WIPI2 or ATG7 alone and the lack of suppression of siWIPI4-mediate toxicity are taken as evidence for autophagy-independence toxicity (Fig. S2C). However, as these knockdowns are visibly only partial (Fig. S2A, B), at least one complete knockout of a core autophagy gene other than WIPI4 and ATG2A/B is required.
3. A role for mitophagy in siWIPI4/GFP-ATG2A-mediated ferroptosis should be specifically ruled out by downregulation of implicated factors, for example, Parkin/PINK1, Nix, Mfn2/DRP1.
4. WIPI2 and ATG7 act after ATG2A and WIPI4 have already been associated with the nascent phagophore. The upstream autophagy signaling protein FIP200, nucleation factor ATG9A and autophagic PI3-kinase protein ATG14 should also be shown to bear no effect on siWIPI4 and ATG2A-mediated ferroptotic toxicity. This should be correlated with the degree of ER-mito contact localization of GFP-ATG2A - as well as endogenous ATG2A if possible.
5. In case the elimination of upstream proteins (FIP200, ATG9A, ATG14) is found to induce ferroptosis or exacerbate siWIPI4/GFP-ATG2A-mediated toxicity, the authors should tone down the statement on autophagy-independence mechanism, and discuss the possibility that autophagy is required to actively sequester ATG2A away from mitochondria. This should be addressed experimentally by testing a WIPI4/ATG2A-dependent alleviation of ferroptosis (or oxidation of mitochondrial PE) upon induction of autophagy (e.g. by mTOR inhibition). On the other hand, if the authors maintain that WIPI4 shields ATG2A from association with TOM40 (Fig. S5A) in an autophagy-independent manner, the effect on this association of lost FIP200/ATG9A/ATG14 should be ruled out experimentally - while maintaining the effect of WIPI4 depletion.
6. Autophagic and mitophagic activities should be assayed, quantified and discussed for each genetic intervention performed throughout the study - by knockdown and/or exogenous expression. This is critical to conclude an autophagy-independent mechanism.

Methods should be written concisely, but should contain all elements necessary to allow interpretation

and replication of the results. As a guideline, Methods sections typically do not exceed 3,000 words. The Methods should be divided into subsections listing reagents and techniques. When citing previous methods, accurate references should be provided and any alterations should be noted. Information must be provided about: antibody dilutions, company names, catalogue numbers and clone numbers for monoclonal antibodies; sequences of RNAi and cDNA probes/primers or company names and catalogue numbers if reagents are commercial; cell line names, sources and information on cell line identity and authentication. Animal studies and experiments involving human subjects must be reported in detail, identifying the committees approving the protocols. For studies involving human subjects/samples, a statement must be included confirming that informed consent was obtained. Statistical analyses and information on the reproducibility of experimental results should be provided in a section titled "Statistics and Reproducibility".

All Nature Cell Biology manuscripts submitted on or after March 21 2016 must include a Data availability statement as a separate section after Methods but before references, under the heading "Data Availability". For Springer Nature policies on data availability see <http://www.nature.com/authors/policies/availability.html>; for more information on this particular policy see <http://www.nature.com/authors/policies/data/data-availability-statements-data-citations.pdf>. The Data availability statement should include:

- Accession codes for primary datasets (generated during the study under consideration and designated as "primary accessions") and secondary datasets (published datasets reanalysed during the study under consideration, designated as "referenced accessions"). For primary accessions data should be made public to coincide with publication of the manuscript. A list of data types for which submission to community-endorsed public repositories is mandated (including sequence, structure, microarray, deep sequencing data) can be found here <http://www.nature.com/authors/policies/availability.html#data>.
- Unique identifiers (accession codes, DOIs or other unique persistent identifier) and hyperlinks for datasets deposited in an approved repository, but for which data deposition is not mandated (see here for details <http://www.nature.com/sdata/data-policies/repositories>).
- At a minimum, please include a statement confirming that all relevant data are available from the authors, and/or are included with the manuscript (e.g. as source data or supplementary information), listing which data are included (e.g. by figure panels and data types) and mentioning any restrictions on availability.
- If a dataset has a Digital Object Identifier (DOI) as its unique identifier, we strongly encourage including this in the Reference list and citing the dataset in the Methods.

We recommend that you upload the step-by-step protocols used in this manuscript to the Protocol Exchange. More details can found at www.nature.com/protocolexchange/about.

FIGURES – Colour figure publication costs \$600 for the first, and \$300 for each subsequent colour

figure. All panels of a multi-panel figure must be logically connected and arranged as they would appear in the final version. Unnecessary figures and figure panels should be avoided (e.g. data presented in small tables could be stated briefly in the text instead).

All imaging data should be accompanied by scale bars, which should be defined in the legend. Cropped images of gels/blots are acceptable, but need to be accompanied by size markers, and to retain visible background signal within the linear range (i.e. should not be saturated). The boundaries of panels with low background have to be demarked with black lines. Splicing of panels should only be considered if unavoidable, and must be clearly marked on the figure, and noted in the legend with a statement on whether the samples were obtained and processed simultaneously. Quantitative comparisons between samples on different gels/blots are discouraged; if this is unavoidable, it should only be performed for samples derived from the same experiment with gels/blots were processed in parallel, which needs to be stated in the legend.

Regardless of format, all figures must be vector graphic compatible files, not supplied in a flattened raster/bitmap graphics format, but should be fully editable, allowing us to highlight/copy/paste all text and move individual parts of the figures (i.e. arrows, lines, x and y axes, graphs, tick marks, scale

bars etc.). The only parts of the figure that should be in pixel raster/bitmap format are photographic images or 3D rendered graphics/complex technical illustrations.

The total number of Supplementary Figures (not including the “unprocessed scans” Supplementary Figure) should not exceed the number of main display items (figures and/or tables (see our Guide to Authors and March 2012 editorial <http://www.nature.com/ncb/authors/submit/index.html#suppinfo>; <http://www.nature.com/ncb/journal/v14/n3/index.html#ed>). No restrictions apply to Supplementary Tables or Videos, but we advise authors to be selective in including supplemental data.

Each Supplementary Figure should be provided as a single page and as an individual file in one of our accepted figure formats and should be presented according to our figure guidelines (see above). Supplementary Tables should be provided as individual Excel files. Supplementary Videos should be

provided as .avi or .mov files up to 50 MB in size. Supplementary Figures, Tables and Videos must be accompanied by a separate Word document including titles and legends.

GUIDELINES FOR EXPERIMENTAL AND STATISTICAL REPORTING

REPORTING REQUIREMENTS – We are trying to improve the quality of methods and statistics reporting in our papers. To that end, we are now asking authors to complete a reporting summary that collects information on experimental design and reagents. The Reporting Summary can be found here <https://www.nature.com/documents/nr-reporting-summary.pdf> If you would like to reference the guidance text as you complete the template, please access these flattened versions at <http://www.nature.com/authors/policies/availability.html>.

Author Rebuttal to Initial comments

PART 1: REVIEWERS' COMMENTS THAT EDITOR DEEMED ESSENTIAL TO ADDRESS:

Rev#3 in points #2 through #6

Point 2. The lack of toxicity upon knockdown of WIPI2 or ATG7 alone and the lack of suppression of siWIPI4-mediate toxicity are taken as evidence for autophagy-independence toxicity (Fig. S2C). However, as these knockdowns are visibly only partial (Fig. S2A, B), at least one complete knockout of a core autophagy gene other than WIPI4 and ATG2A/B is required.

Response to Point 2: We have tested this in ATG16L1 knockout complete null-autophagy background (**Figure 2B and S2D**).

Left: **Figure 2b.** The graph represents the cytotoxicity (n=3) in WIPI4 siRNA and scramble siRNA treated ATG16L1 knockout HeLa and its control cells. Cell cytotoxicity was measured at 72 hours post transfection by LDH release and data are presented as normalised mean values \pm SEM. One sample *t*-test to wildtype scramble and two-tailed paired *t*-test between samples. *P<0.05.

Right: **Figure S2d.** Western blots showing that there is no LC3-II due to defective membrane conjugation in ATG16L1 knockout HeLa cell line we used and this is an autophagy-null cell line. Cells were treated with DMSO or 250 nM bafilomycin A1 for 4 hours.

Point 3. A role for mitophagy in siWIPI4/GFP-ATG2A-mediated ferroptosis should be specifically ruled out by downregulation of implicated factors, for example, Parkin/PINK1, Nix, Mfn2/DRP1.

Response to Point 3: We tested this with NIX knockdown as a representative mitophagy adaptor. We chose NIX because MFN2 plays a significant role in the connection of the ER and mitochondria. MFN2 in mitochondria is assembled into homo- or heterodimeric complexes with MFN2 in the ER when the mitochondrial fusion process is initiated (PMID: 19052620). When the expression of MFN2 is suppressed, the structure and function of the mitochondria are destroyed. However, when MFN2 is overexpressed, the interaction of the ER and mitochondria is enhanced (PMID: 29309902). Parkin alters MAMs integrity by affecting the ubiquitination of MFN2 (PMID: 30219582).

NIX knockdown did not rescue the cell death induced by WIPI4 depletion (**Figure S7g**).

Mitochondria mass is not changed by WIPI4 knockdown as measured by nonylacridine orange (NAO) staining (**Figure S7h**). NAO rapidly enters and accumulates in mitochondria regardless of the mitochondrial transmembrane potential with 50% of fluorescent signal from a 30 minute incubation being present after just 1 minute and is not altered by oxidative and nitrosative stress (PMID: 29806036).

Figure S7g. The graph represents the cell death levels measured as dead cell numbers. On Day 0, HeLa cells were transfected with pSpCas9(BB)-2A-GFP construct containing sgRNA targeting WIPI4 gene Exon 3 (KO) or the non-targeting sgRNA (nt-43) (ctrl). HeLa cells were transfected with 50 nM of NIX or scramble siRNA oligos 24 hours later and then grown for another 24 hours before staining with Incucyte® Cytotox Red Dye from Sartorius (Cat.# 4632). Images were taken between 48-72 hours with IncucyteS3 live imaging system after first transfection and dead cells were counted as red objects using the Incucyte 2020 software. Data are presented as normalised mean values \pm SEM (n=3). One sample t-test to control and two-tailed paired t-test between other samples. ns., not significant, *P<0.05, **P<0.005, ***P<0.001 throughout this document and the paper. Western blots on the right show the silencing efficiency of WIPI4 and NIX. **S7h.** Mitochondria abundance as measured by flow cytometry for the intensity of nonylacridine orange (NAO) staining signal (n=3). HeLa cells were transfected with WIPI4 or scramble siRNA and grown for 24 hours before stained with 2.5 μ M NAO and subjected to flow cytometry to measure mitochondria mass.

Point 4. WIPI2 and ATG7 act after ATG2A and WIPI4 have already been associated with the nascent phagophore. The upstream autophagy signaling protein FIP200, nucleation factor ATG9A and autophagic PI3-kinase protein ATG14 should also be shown to bear no effect on siWIPI4 and ATG2A-mediated ferroptotic toxicity. This should be correlated with the degree of ER-mito contact localization of GFP-ATG2A - as well as endogenous ATG2A if possible.

Response to Point 4: Beclin-1 knockout (**Figure S2e**) and ULK1 inhibition (FIP200 acts with ULK1) (**Figure S2f**) cannot rescue WIPI4 siRNA induced cell death. Both of these target steps very upstream in autophagosome biogenesis, prior to WIPI2, ATG7, WIPI4 and ATG2A.

Figure S2e. The graph represents the cytotoxicity measured by LDH assay of Beclin-1 knockout HeLa and its control cells treated as indicated (n=3). On Day 0, cells were transfected with WIPI4 or scramble siRNA and grown for 24 hours before LDH assay. Data are presented as normalised mean values ± SEM. One sample t-test to control and two-tailed paired t-test between scramble and siWIPI4 samples in Beclin1 knockout cells. The western blots show the WIPI4 and Beclin-1 levels in the cells treated as described above. **Figure S2f.** The graph represents the cell death levels of HeLa cells treated as indicated (n=3). Cells were transfected with 50 nM WIPI4 or scramble siRNA and then treated with 5 µM ULK1 inhibitor SBI-0206965 for 24 hours before staining with CellTox™ Green Dye from Promega (Cat.# G8741). Images were taken with the IncucyteS3 live imaging system and dead cells were quantified as green (dead cells) area vs. phase (total cells) area using the Incucyte 2020 software. Data are presented as normalised mean values ± SEM, one-sample t-test to the scramble in wildtype cells and paired t-test between SBI-0206965 treated scramble and siWIPI4 cells. Western blots show the WIPI4 and LC3-II levels in the cells treated as described above.

Point 5. In case the elimination of upstream proteins (FIP200, ATG9A, ATG14) is found to induce ferroptosis or exacerbate siWIPI4/GFP-ATG2A-mediated toxicity, the authors should tone down the statement on autophagy-independence mechanism, and discuss the possibility that autophagy is required to actively sequester ATG2A away from mitochondria. This should be addressed experimentally by testing a WIPI4/ATG2A-dependent alleviation of ferroptosis (or oxidation of mitochondrial PE) upon induction of autophagy (e.g. by mTOR inhibition). On the other hand, if the authors maintain that WIPI4 shields ATG2A from association with TOMM40 (Fig. S5A) in an autophagy-independent manner, the effect on this association of lost FIP200/ATG9A/ATG14 should be ruled out experimentally - while maintaining the effect of WIPI4 depletion.

Response to Point 5: Although upstream autophagy regulators Beclin-1 and ULK1 loss of function did not cancel the cell death induced by WIPI4 depletion as shown above, we still tested whether induction of autophagy would alleviate WIPI4/ATG2A-dependent ferroptosis for curiosity. As shown in Figure S2g, Rapamycin treatment did not inhibit WIPI4 knockout induced ferroptosis.

Figure S2g. The graph represents the cell death levels of HeLa cells treated as indicated (n=3). On Day 0, HeLa cells were transfected with pSpCas9(BB)-2A-GFP construct containing sgRNA targeting WIPI4 gene Exon 3 (KO) or its control construct (spCas9). Cells were grown for 24 hours before treatment with 200 nM Rapamycin in media containing Incucyte® Cytotox Red Dye from Sartorius. Images were taken between 48-72 hours after transfection and dead cells were counted as red objects using the Incucyte 2020 software. Data are presented as normalised mean values \pm SEM. One sample t-test to control and two-tailed paired t-test between other samples. Western blots show the WIPI4 and LC3-II levels in the cells treated as described above.

Point 6. Autophagic and mitophagic activities should be assayed, quantified and discussed for each genetic intervention performed throughout the study - by knockdown and/or exogenous expression. This is critical to conclude an autophagy-independent mechanism.

Response to Point 6: LC3-II levels were measured in the above experiments and in the experiment of ATG2mLIR overexpression (**Figure S5c**). Please note that ATG16L1 knockout (**Figure 2b, S2d, 3b**) is autophagy-null and by definition mitophagy-null.

Strengthening the link between WIPI4 and the localization of ATG2 as well as further assessing the potential role for other PS transfer proteins:

Rev#1 point #2

Rev#2 point #3

Rev#3 point #1

Rev#1 point #2: Other PS transfer proteins should also be functioning at ER-mitochondria contacts or mitochondria-associated ER membrane (MAM) (e.g. PMID: PMC9375143). The effect of these proteins and the integrity of MAM - critical for PS synthesis and transfer to mitochondria - should be investigated in WIPI4-depleted cells.

Response to Reviewer #1 point #2. Thank you for your advice. We found silencing other PS transfer proteins (ORP5/8) rescued the cell death (**Figure 5h**) and increase of mito-PE (**Figure 5i**) induced by WIPI4 depletion. The western blots in **Figure S10b&c** show the silencing efficiency of WIPI4 and ORP5/8 proteins. This further verified that the abnormal PS transport onto mitochondria underlies WIPI4-ATG2 dependent ferroptosis. The integrity of MAM is not significantly changed as measured by PLA assay in **Figure S6b**.

Figure 5h. The graph represents the cell death levels of HeLa cells transfected with 50 nM of each siRNA (n=3). Cells were stained with CellTox™ Green Dye from Promega (Cat.# G8741) 48 hours post transfection. The levels of cell death were quantified as the ratio of green (dead cells) area vs. phase (total cells) area using the Incucyte 2020 software. Data presented are normalised mean values ± SEM. One sample t-test to control and two-tailed paired t-test between other samples.

Figure 5i. PE levels in mitochondria fraction of HeLa cells transfected with indicated siRNAs (n=3). Cells were subjected to cell fractionation 48 hours post transfection and mitochondrial samples of the same amount of protein were used for the phosphatidylethanolamine assay. Data in 5h and 5i are presented as normalised mean values ± SEM. One sample t-test to control and two-tailed paired t-test between other samples. ns; not significant, *P<0.05, **P<0.01.

Figure S10b&c Western blots showing the silencing efficiency of WIPI4, ORP5, ORP8 and enrichment of MAM marker calnexin.

Figure S6b. Proximity ligation assay for VDAC1 and IP3R1 proteins (n=2). After fixation and blocking, HeLa cells were incubated overnight with rabbit anti-IP3R3 (ITPR3) 1:100 (Chemicon) and mouse anti-VDAC1 1:100 (Abcam) primary antibodies. After PBS wash, the proximity ligation assay (PLA) probes anti-rabbit PLUS anti-mouse MINUS were added for 1 hour in the antibody diluent buffer. The Duolink detection fluorophore red excitation/emission = 594/624 nm was used. The area of PLA signal in each cell positive of GFP-spCas9 was analysed using ImageJ. As a negative control, one or both of the primary antibodies was/were omitted, and the PLA probe background staining was imaged. The background staining was limited for both probes. The data presented are the area of PLA signal in cells positive of GFP-spCas9 in arbitrary units (unpaired t-test). There was no significant decrease of PLA signal in WIPI4 knockout cells in these two biological repeats.

Rev#2 Point 3. A major conclusion is that loss of WIPI4 renders more mitochondrial localization of ATG2, yet data supporting this claim should be strengthened. For example, for Fig. 3c, if one normalizes the loading (using NDUFA, PNS frx vs mito frx), ATG2 amount in mito-enriched frx was very low; also, it seems that WIPI4 depletion reduced total/PNS ATG2 - is this really the case? For Fig. S4C image of ATG2A-mito co-localization, can you do a high-resolution imaging as Fig. S4B? The resolution of Fig. S4C is just too low.

Response to Rev#2 Point 3.

ATG2 localises on multiple other cellular compartments, including ER, autophagosomes and lipid droplets. During enrichment of mitochondria from PNS, a lot of cytoplasmic proteins including non-mitochondria binding ATG2A were removed. Therefore, the amount of ATG2 protein in mitochondria-enriched fraction is definitely less than that in crude PNS, and normalising to NUDFA9 only makes the difference bigger because NUDFA9 is enriched, while non-mitochondria-binding ATG2A is removed in the mitochondria-enriched fraction. But in the 'purified' mitochondria fraction, ATG2A is more enriched in siWIPI4 cells than scramble cells.

When normalising the ATG2A levels to the loading (α -tubulin) using the blot from **Figure 3b**, we found that ATG2A was not reduced in PNS of siWIPI4 cells (1.229597) compared to scr cells (1.006758). The NUDFA9 levels are not significantly different in scr and siWIPI4 total/PNS when referring to more blots (e.g. **Figure S5a**).

To improve the resolution of ATG2-mito localisation, we have now added super-resolution microscopy data (**Figure S5d**) and additional confocal data (**Figure S5e &f, S6a**). **Figure S5d** shows representative super-resolution images of endogenous ATG2A co-stained with mitochondria marker TOMM20 and ER protein Sec61-BFP. In control cells, 471/2531 ATG2A structures colocalised with mitochondria, while in WIPI4 knockout cells, 407/1438 ATG2A structures colocalised with mitochondria ($p < 0.0001$ chi-squared test). 5 cells were scored for each condition. **Figure S5e** shows confocal images of endogenous ATG2A co-stained with Mitotracker Far-red and ER protein Sec61-BFP. **Figure S5f** shows that the Pearson's coefficient of ATG2A and Mitotracker Far-red is increased in WIPI4 Exon 3 knockout HeLa cells ($n=3$).

Figure S6a shows confocal images of endogenous ATG2A co-stained with the MAM marker calnexin. Quantification of pearson's coefficient using images from these experiments ($n=3$) shows that the colocalization of ATG2A and calnexin was increased in wildtype and ATG16L1 knockout HeLa cells depleted of WIPI4 (**Figure 3c**).

Rev#3 point #1. As duly mentioned, mitochondrial association ATG2A reportedly depends on TOMM40 (mediated by TOM70), whereas lipid transport from the ER requires the ER-resident scramblase TMEM41B. Loss-of-function of each should therefore be shown to suppress siWIPI4/GFP-ATG2A-mediated cytotoxicity. In contrast, exogenous expression of the LIR-deficient ATG2A (mLIR, PMID: 32009292) and ATG9A-inert ATG2A (Δ 237-431, PMID: 31412244) should be shown to be at least as toxic as WT ATG2A in a manner that is suppressed by ferroptosis inhibitors - in presence or absence of WIPI4.

Response to Rev#3 point #1:

Loss-of-function of TOMM40 suppressed siWIPI4/GFP-ATG2A-mediated cytotoxicity as shown in **Figure 3e**. We tried TMEM41b knockdown and CRISPR knockout but both increased cell death. Therefore we could not further test whether TMEM41B depletion cancels siWIPI4/GFP-ATG2A-mediated cytotoxicity (see data below).

Left: **Figure 3e.** The graph represents the cytotoxicity (n=3) measured by LDH release in HeLa cells transfected with indicated siRNAs. Cell cytotoxicity was measured at 72 hours post transfection and data are presented as normalised mean values \pm SEM. One sample t-test to scramble and two-tailed paired t-test between other samples. The western blots show the silencing efficiency of WIPI4 and TOMM40.

Right: The graph represents the cell death levels of HeLa cells treated with indicated siRNA oligos (n=3). HeLa cells were transfected with 50 nM WIPI4 and 50 nM TMEM41B siRNA on Day 0 and 50 nM TMEM41B siRNA oligos on Day 1. Cells were grown for another 24 hours before staining with Incucyte® Cytotox Red Dye from Satorious. Images were taken 72 hours after first transfection and dead cells were counted as red objects using the Incucyte 2020 software. Data are presented as normalised mean values \pm SEM. One sample t-test to control and two-tailed paired t-test between samples. Western blots show the WIPI4 and LC3-II levels in the cells treated as described above.

We don't think ATG9-inert ATG2 would be ideal for testing the effect of ATG9 on WIPI4-ATG2 dependent ferroptosis due to the role ATG9 plays in plasma membrane integrity. ATG9A counters physiological plasma membrane (PM) perturbations associated with pyroptotic and necroptotic programmed cell death pathways. During pyroptosis, gasdermin pores form on the PM after proteolytic processing of gasdermin-D (GSDMD), which entails the liberation of the amino-terminal fragment (GSDMD-NT) and its subsequent oligomerization into a pore-like structure at the PM (PMID: 27281216). During necroptosis, mixed lineage kinase domain like (MLKL) generates pores at the PM (PMID: 33124082). In response to Ca^{2+} influx due to PM damage, IQGAP1, a known Ca^{2+} responder, recruits ATG9A to the damaged PM, and together they organize the ESCRT machinery for PM repair (PMID: 34257406). It is unknown whether artificial overexpression of ATG9-inert ATG2 would complicate the plasma membrane repair and the cell death assays available generally depend on the measurement of the PM permeabilisation.

ATG2mLIR: (Figure S5c)

Figure S5c. HeLa cells transfected with pHAGE-GFP-ATG2AmLIR and pHAGE-GFP-ATG2A were treated with DMSO or Lip-1 for 24 hours in media containing Incucyte® Cytotox Red Dye from Sartorius (n=3). Images were taken 30 hours after first transfection. Dead cells were counted as red objects and the number of transfected cells were counted as green objects using the Incucyte 2020 software. Data are presented as the ratio of red objects count vs. the green objects count to normalise to the transfection efficiency of each sample. At least 30 images were taken for each sample. The average dead cell ratios of each sample were normalised to that of WT-ATG2A DMSO. Western blots on the right show the protein levels of overexpressed wildtype ATG2A and ATG2AmLIR.

We overexpressed the LIR-deficient ATG2A and its wildtype control in HeLa cells using the constructs kindly gifted by David McEwan Lab which were produced for their paper you mentioned (PMID: 32009). mLIR-ATG2A induced more cell death compared to its wildtype control as predicted by our model and this was rescued by ferroptosis inhibitor Lip-1 (n=3) as shown in **Figure S5c**.

Rev#1 and Rev#2 feel that further analyses of the link between PE and ferroptosis are needed to maintain the core conclusions and that the nature of the cell death process requires further investigation:

Rev#1, points #3, 6

Rev#2 points #1, 2, 4, 5, 6

Rev#1 point #3 If PS to PE conversion drives ferroptosis, what are the effects of blocking PS synthesis in the ER by interfering with PS synthases?

Response to Rev#1 points #3: Thanks for your advice. We tried to knockdown both PS synthases but this induced cell death. Therefore we could not further test whether PTDSS1/2 depletion cancels siWIP14/GFP-ATG2A-mediated cytotoxicity.

The graph represents the cell death levels of HeLa cells transfected with 50 nM of each siRNA (n=3). Cells were stained with CellTox™ Green Dye from Promega (Cat.# G8741) 48 hours post transfection. The levels of cell death were quantified as the ratio of green (dead cells) area vs. phase (total cells) area using the Incucyte 2020 software. Data presented are normalised mean values \pm SEM. One sample t-test to scramble.

Rev#1 point #6 Besides total mitochondria PE, the unsaturation of acyl chains should also be examined. Unsaturated double bonds are more critical for ferroptosis.

Response to Rev#1 points #6. The levels of unsaturated fatty acids measured using Lipid Assay Kit (ab242305) were increased by transient WIPI4 knockout in mitochondria fraction but not in total lysates (n=3) as shown in **Figure 4e**.

Figure 4e. Unsaturated fatty acids in mitochondria fraction and total lysates of anti-WIPI4 and scramble siRNA transfected HeLa cells (n=3). HeLa cells transiently transfected with pSpCas9(BB)-2A-GFP with sgRNA targeting WIPI4 Exon 3 or its control construct were grown for 48 hours before mitochondria purification and lipid extraction (after calibration to protein concentrations). Each mitochondria or total cell lysate lipid sample was resuspended with 60 μ l DMSO and 5 μ l was used for each technical repeat for the measurement of unsaturated fatty acids. Data represented are mean values (nmol unsaturated fatty acids per deciliter lysates) \pm SEM. Two-tailed paired *t*-test.

Rev#2 point #1 The study starts with the observation that siRNA against WIPI4 in several cell lines causes cell death, presumably, ferroptosis. Since the observed cell death is quite modest (around 30% as in Fig. 1 and Fig. S1) compared to other commonly used ferroptosis triggers, this reviewer recommend the authors to establish a more reliable and easy-to-control approach for WIPI4-inhibition-triggered ferroptosis, such as Dox-inducible WIPI4 shRNA using both a cancer cell line and iPSC-derived neurons. Such an inducible system allows for more thorough analysis such as time course and dose analysis (Dox concentration will control the degree of WIPI4 elimination), which is needed to lend more convincing support.

Rev#2 Point #2. About the nature of the modest cell death triggered by WIPI4 elimination, although their pharmacological analysis suggests ferroptosis, the confidence is weakened by their result that NAC, a commonly-used ferroptosis inhibitor, failed to prevent the death (Fig. 1A and Fig. S1D). This might be due to that they were using a much lower NAC concentration in their experiments. To validate it is indeed ferroptosis, in an inducible shRNA system (see point-1), suggest (1) in experiments using NAC, increase NAC dose to 0.2 mM and also include a positive control (for example, erastin as ferroptosis trigger) +/- NAC ; and (2) test whether WIPI4 loss-induced cell death can also be inhibited by iron chelation.

Response to Rev#2 point #1 & 2

Left: Figure S1f. HeLa cells transiently transfected with pSpCas9(BB)-2A-GFP with sgRNA targeting WIPI4 Exon 3 or its control construct were grown for indicated hours before lysis for western blot analysis to detect the knockout efficiency of WIPI4 protein.

Figure 1c. Cell death quantified as dead cell numbers in HeLa cells transiently transfected with above constructs. Cells were grown for 24 hours before treatment with n-acetyl cysteine (NAC) or deferoxamine (DFO) at the of indicated concentrations in the media containing Incucyte Red dye (from Sartorius). Images were taken at 72 hours post transfection. Dead cells were counted as red objects using the Incucyte 2020 software. Normalised mean \pm SEM. One sample *t*-test to control and two-tailed paired *t*-test between other samples. **P*<0.05.

Thank you for your advice. We designed sgRNA oligos targeting different parts of WDR45 and cloned them into pSpCas9(BB)-2A-GFP construct. We found that the sgRNA targeting Exon 3 was most efficient and WIPI4 was depleted at 88 hours post transfection of the construct (**Figure S1f**). We then used this construct for transient knockout of WIPI4 in HeLa cells and monitored cell death using Incucyte. As shown in **Figure 1c**, transient CRISPR knockout of WIPI4 induces doubles the cell death as measured by dead cell numbers. And 0.2 mM NAC, 1 mM NAC and 100 μ M DFO could rescue this increase of cell death.

Rev#2 point #4 In Fig. 5 and S7, etc., authors test the involvement of mito PE synthase (mitoPISD) by using pharmacological agents and siRNA specific to mitoPISD. Based on the presented results, it is hard to appreciate whether mitoPISD but not cytosolic PISD was specifically eliminated by siRNA. What exactly is the assay for this claim?

Response to Rev#2 point #4:

Figure S9b. Western blots show the silencing efficiency of anti-mitoPISD siRNA in mitochondria fractions, post-mitochondria fractions, total lysates and nuclear pellets of HeLa cell lysates. HeLa were transfected with 100 nM of scramble or anti-mitoPISD (#2) siRNA oligos and grown for 48 hours before fractionation. TOMM20 is used as the marker of the mitochondria enriched fractions and calnexin is the marker of the MAM enriched fraction. The graph represents the PISD protein levels normalised to the loading (α -tubulin) in each fraction (n=3). Data are presented as normalised mean values \pm SEM. One sample t-test to scramble and two-tailed paired t-test between other samples.

According to Santosh Kumar et al. (PMID: 33593792), there is another isoform of PISD which also locates on lipid droplets (transcript variants NM_0014338 and NM_00178022, including isoform d). We designed the simitoPISD siRNA oligo (GCACCGAUGUCUGGGAUUACU) using the sequence specific to the mitochondria isoform of PISD (transcript variants NM_001326411 to 001326421). Fractionation experiments show that this oligo only decreased PISD levels in the mito fraction but not other fractions (**Figure S9b**).

Rev#2 point #5 Also for mitoPISD, (1) would miPISD overexpression increase mito PE content and trigger ferroptosis? This seems to be the logical conclusion based on the proposed mechanism. (2) Examine whether mitoPISD modulates the sensitivity to ferroptosis induced by erastin or RSL3. This is important considering the reported role of mitochondria in ferroptosis induced by erastin but not RSL3.

Response to Rev#2 point #5:

(1) The overexpression of mitochondria specific PISD induced more cell death in HeLa cells and this was rescued by Lip-1 (**Figure 5e**). The characterisation of mitoPISD construct: We purchased the PISD-myc-DDK construct from Origene (CAT#: RC222868) which is isoform d that localises on both mitochondria and lipid droplets. According to Santosh Kumar et al. (PMID: 33593792), mutation of three Leu residues of this isoform to positively charged residues (Leu80Lys, Lue89Lys, Leu90Arg) results in a protein that does not localize to the lipid droplets, however, mitochondrion targeting is unaffected. We made mitoPISD following their strategy and validated the mitochondrial localisation by both western blotting (**Figure S9e**) and immunofluorescence experiments (**Figure S9f**). As shown in **Figure S9f**, the mitoPISD we made, which is stained with FLAG antibody, colocalised with Mitotracker but not with the lipid droplet marker BPC12.

Figure 5e. Cell death quantified as dead cell numbers in HeLa cells transfected with CMV-myc-DDK or mitoPISD-myc-DDK (n=3). Cells were grown for 24 hours before treatment with DMSO or 100 nM Lip-1 in media containing Incucyte Red dye. Images were taken at 48 hours post transfection. Dead cells were counted as red objects using the Incucyte 2020 software. Normalised mean \pm SEM. One sample *t*-test to control and two-tailed paired *t*-test between samples. *P<0.05.

Figure S9e. Western blots showing the expression levels mitoPISD-Myc-DDK as stained by anti-FLAG and anti-PISD antibodies. HeLa cells transfected with CMV-myc-DDK or mitoPISD-myc-DDK were grown for 24 hours before lysis and western blotting.

Figure S9f. Confocal images showing the colocalisation of mitoPISD-Myc-DDK with Mitotracker red and BPC12. HeLa cells transfected with CMV-myc-DDK or mitoPISD-myc-DDK were grown for 24 hours before fixation and immunostaining with indicated antibodies.

(2) We found that simitoPISD rescued both erastin and RSL3 induced ferroptosis (**Figure S10d, Figure S9a**) in SH-SY5Y cells. PISD might be a substrate supplier that initially fuels mitochondrial lipid peroxidation and thereby increases susceptibility to ferroptosis. PE converted from PS by mitoPISD might be translocated and used in other organelles and membrane structures too. Therefore the reduction of PE in cells reduces cellular susceptibility to ferroptosis even when it is induced by RSL3.

Figure S10d. SH-SY5Y cells were transfected with 100 nM of scramble or anti-mitoPISD (#2) siRNA oligos. Cells were treated with drugs of indicated concentration in media containing CellTox™ Green Dye from 48 hours post first transfection. Images were taken at 60 hours post transfection. The levels of cell death were quantified as the ratio of green (dead cells) area vs. phase (total cells) area using the Incucyte 2020 software. Data presented are normalised mean \pm SEM (n=3). One sample t-test to scramble DMSO control and two-tailed paired t-test between other samples.

Figure S9a. Representative blots showing the silencing efficiency of 100 nM anti-mitoPISD siRNA Oligo #1 and Oligo #2 in SH-SY5Y cells .

Rev#2 point #6 Although most of their experimental observations are consistent with the conclusion that WIPI4 elimination triggers ferroptosis via increasing mitochondrial PE contents, all the evidence are correlative and there lacks a causative experiment directly confirming that increased mitochondrial PE triggers ferroptosis (PS liposome experiment is for this purpose, but the experiment may cause too many changes to the cell and as such one cannot attribute the outcome only to mito PE). The authors either need to conduct such an experiment (miPISD overexpression?) or tune down their conclusion.

Response to Rev#2 point #6: Now we have shown that mitoPISD overexpression induced more ferroptosis. We may not need to tune down conclusions about mitochondrial PE contents as the direct cause of WIPI4-ATG2 induced ferroptosis.

PART 2: RESPONSE TO ALL REVIEWERS' COMMENTS

REVIEWER 1

Remarks to the Author:

WIPI4 depletion leads to an induction of lipid peroxidation and ferroptosis. The authors show that this is through an enhanced association of ATG2 with ER-mitochondrial contacts, enhanced PS delivery to mitochondria, and enhanced mitochondrial PE production through mito-PISD. As such, the authors suggest that WIPI4 plays a role in keeping ATG2 away from MAM, providing a potential autophagy-independent mechanism for WIPI4's role in BPAN. This is a very interesting model.

Whilst the overall quality of the data is good and a range of complementary techniques were used, there are several issues in the manuscript that require rectification and clarification. General comments:

1. The link between WIPI4 mutation and ferroptosis in BPAN is not new (e.g. PMID: 34012978). This manuscript confirmed this link (not 'revealed' as in the abstract).

Response: We have changed 'revealed' to 'observed'.

2. Other PS transfer proteins should also be functioning at ER-mitochondria contacts or mitochondria-associated ER membrane (MAM) (e.g. PMID: PMC9375143). The effect of these proteins and the integrity of MAM - critical for PS synthesis and transfer to mitochondria - should be investigated in WIPI4-depleted cells.

Addressed in Part 1 Page 6.

3. If PS to PE conversion drives ferroptosis, what are the effects of blocking PS synthesis in the ER by interfering with PS synthases?

Addressed in Part 1 Page 11.

4. WT cells should be included in the analysis shown in Fig. 4E, F, and G. This is particularly important for demonstrating the PS transfer activity of ATG2 from the ER to mitochondria.

Response: ATG2A/B KO cells reconstituted with ATG2A GFP are compared to ATG2A/B null cells - we believe these are better controls as they allow us to test the ATG2-dependency of these phenomena, which was the objective of these experiments. As an extra control we have added new data in these panels with the ATG2A-LTD mutant that is lipid-transfer-deficient (and behave like the ATG2-null cells with these assays – this also demonstrates that lipid transfer activity of ATG2 is required for the ferroptosis. Extensive data are shown throughout the paper with WT cells and WIPI4-depleted cells for many readouts in Fig 2.

5. WT cells should also be included in Fig. 5D, E, and F.

Response: Same as above.

6. Besides total mitochondria PE, the unsaturation of acyl chains should also be examined. Unsaturated double bonds are more critical for ferroptosis.

Addressed in Part 1 Page 12.

Other comments:

1. Line 139, does TMEM41b interact with ATG2 so to keep it at MAM? Ref. 11 does not have the data.

Response: The N-terminal of ATG2 interacts with TMEM41B to localise it onto ER. Sorry that we missed one reference (PMID: 33850023). Now this reference is added to the main text.

2. Fig S1b, what are the 3 asterisks above the 3rd column referring to?

Response: They indicate the p-value. It is now one asterisk when we do a one-sample t-test. One asterisk means $P < 0.05$ and there is significant difference of viability between scr and siWIPI4 DMSO treated samples by one-sample t-test. This is now added to the figure legend.

3. Effects seen in S1i is surprising given the lack of knockdown in S1j (minimal difference between shWIPI4 and the scr control, and yet a marked difference in LDH release). Is this blot representative, also which band is the appropriate band?

Response: We have changed to more representative blots (Figure S1j). The lower band (labelled by an asterisk mark) is the band for WIPI4. It shows that even incomplete knockdown of WIPI4 causes ferroptosis.

4. Bands in Fig 3C are too closely cropped (some bands are nearly excluded) and the expected sizes of each protein should be indicated accordingly to clarify those that have been quantified. Further, there is a clear size shift in WIPI4 in the mitochondrial fraction – authors should comment on this.

Response: We have fixed the cropping to show all bands and added the molecular weights of proteins in this gel. The different mobility of WIPI4 in the mito fraction may be due to the different buffers used in fractionation. The mitochondria were collected as pellets and resuspended in PBS, while the PNS and post-mito fractions were both collected in isotonic buffer (10mM Tris-Cl, pH7.4, 250 mM sucrose, 1mM EDTA supplemented with protease and phosphatase inhibitors). We have added text to the figure to explain this phenomenon.

All full-length blots are in the supplementary figure that was requested for the revision.

5. Figure S5b has cropped blots that are merged into one, these need to be separated and more of the gel shown. Why is TMEM41B IPed in the IgG control lane?

Response: We now show more of the gel (current Figure 3d) and it is easier to see that this blot is not merged with other blots. The band in the IgG control lane is a non-specific band because it is of different size from the bands of TMEM41b in other lanes.

All full-length blots are in the supplementary figure that was requested for the revision.

6. In Figure 5I, the authors state that “there was also an increase of cytotoxicity in cells infected with wildtype ATG2A lentivirus, which could be rescued by co-treatment with Fer-1; but ATG2A-RLA lentivirus failed to induce cytotoxicity”, however, whilst non-significant, there is a striking trend (2-fold) from the ctrl DMSO to the RLA-infected DMSO condition. Stating this as a failure to induce cytotoxicity appears misleading.

Response: We now deleted data of ATG2A-RLA from the paper – this mutant is testing the same deficiency as ATG2A-MLD, where our data are more complete, and thus we felt that it was sensible to remove the ATG2A-RLA data.

Comments regarding the methodology

1. There are no methods and insufficient information regarding how the LC3 and ATG2A colocalization experiments were quantified.

Response: We now added relevant information in ‘Immunofluorescence Microscopy for cultured cells’ of the methods.

‘The time period of LC3-RFP and ATG2A-GFP colocalisation was quantified from movies imaged live with Confocal 780. HeLa cells stably expressing RFP-LC3 were transfected with GFP-ATG2A constructs and grown for 24 hours before live imaging. The time interval between frames was 3.87 seconds. The imaging of movies started as soon as a GFP/RFP double positive event was identified and ended when GFP-ATG2 leaves RFP-LC3. The maximum number of frames in the movies is 36 so the maximum observed time length is 140 seconds.

There might be events that GFP-ATG2 and RFP-LC3 colocalise longer than 140s but the events we observed are generally shorter than that.'

2. More information is required surrounding the state of the cells in seahorse-treated conditions. It is unclear whether the cells are dead or not and whether normal enzymatic functions of the cell can still occur. Why does this rule out ER-mitochondrial PS transport? Are the mitochondria still functional? If so, an assay to demonstrate this would be of use, if not, how can we assume mitoPISD retains activity?

Response: The Seahorse reagent is routinely used for mitochondrial functional assays of a range of activities using the Seahorse analyser. Thus mitochondrial activities and ATPase activities are preserved with this reagent. Also note that we have included a control to show that the synthesis of PE was cancelled by mitoPISD siRNA knockdown (Figure 5g). When the plasma membrane is permeabilised with Seahorse, saturating levels of PS liposomes could reach the mitochondria and this makes the PS transported from ER unnecessary for mitochondrial PE synthesis, and cancels the difference of PE synthesis caused by WIPI4 loss of function. Therefore, it is the dysregulation of ER-mitochondrial PS transport that accounts for the difference of mitochondrial PE synthesis in WIPI4 depletion of ATG2 overexpression conditions.

Other less important issues include:

1. In Fig. S8, Duramycin is misspelled.

Response: Thank you. We now have corrected it.

2. Make references to extended figures and Supp figures consistent in their naming.

Response: We have done this.

REVIEWER 2

Remarks to the Author:

In this manuscript, the authors present a comprehensive set of data to support a role of WIPI4 and ATG2A/B, canonical autophagy proteins, in the regulation of ferroptosis, as well as the underlying mechanism as they propose: independently of autophagy, loss of WIPI4 allows ATG2 to transfer phospholipid PS from ER to mitochondria, where PS can be enzymatically converted to PE, followed by PE peroxidation and ferroptosis. This mechanism has potential clinical relevance, as loss of function of WIPI4 is associated with certain specific neurodegenerative diseases.

Major experiments were performed using cancer cell lines, complemented with some studies using iPSC-derived neurons, primary neurons, and zebrafish models. Overall, the study is of excellent quality, and the proposed mechanisms, if sufficiently substantiated, will be highly valuable to the field of ferroptosis. On the other hand, evidence for some key mechanistic conclusions are correlative, the major experimental system (elimination/inhibition of WIPI4 to trigger ferroptosis) is less than optimal, and the authors need to pay more attention to certain experimental design and control. Specifically,

1. The study starts with the observation that siRNA against WIPI4 in several cell lines causes cell death, presumably, ferroptosis. Since the observed cell death is quite modest (around 30% as in Fig. 1 and Fig. S1) compared to other commonly used ferroptosis triggers, this reviewer recommend the authors to establish a more reliable and easy-to-control approach for WIPI4-inhibition-triggered ferroptosis, such as Dox-inducible WIPI4 shRNA using both a cancer cell line and iPSC-derived neurons. Such an inducible system allows for more thorough analysis such as time course and dose analysis (Dox concentration will control the degree of WIPI4 elimination), which is needed to lend more convincing support.

Response: Addressed in Part 1 Page 13.

2. About the nature of the modest cell death triggered by WIPI4 elimination, although their pharmacological analysis suggests ferroptosis, the confidence is weakened by their result that NAC, a commonly-used ferroptosis inhibitor, failed to prevent the death (Fig. 1A and Fig. S1D). This might be due to that they were using a much lower NAC concentration in their experiments. To validate it is indeed ferroptosis, in an inducible shRNA system (see point-1), suggest (1) in experiments using NAC, increase NAC dose to 0.2 mM and also include a positive control (for example, erastin as ferroptosis trigger) +/- NAC ; and (2) test whether WIPI4 loss-induced cell death can also be inhibited by iron chelation.

Response: Thank you. This was also addressed in Part 1 Page 13.

3. A major conclusion is that loss of WIPI4 renders more mitochondrial localization of ATG2, yet data supporting this claim should be strengthened. For example, for Fig. 3c, if one normalizes the loading (using NDUFA, PNS frx vs mito frx), ATG2 amount in mito-enriched frx was very low; also, it seems that WIPI4 depletion reduced total/PNS ATG2 - is this really the case? For Fig. S4C image of ATG2A-mito co-localization, can you do a high-resolution imaging as Fig. S4B? The resolution of Fig. S4C is just too low.

Response: Addressed in Part 1 Page 8.

4. In Fig. 5 and S7, etc., authors test the involvement of mito PE synthase (mitoPISD) by using pharmacological agents and siRNA specific to mitoPISD. Based on the presented results, it is hard to appreciate whether mitoPISD but not cytosolic PISD was specifically eliminated by siRNA. What exactly is the assay for this claim?

Response: Addressed in Part 1 Page 14.

5. Also for mitoPISD, (1) would miPISD overexpression increase mito PE content and trigger ferroptosis? This seems to be the logical conclusion based on the proposed mechanism. (2) Examine whether mitoPISD modulates the sensitivity to ferroptosis induced by erastin or RSL3. This is important considering the reported role of mitochondria in ferroptosis induced by erastin but not RSL3.

Response: Addressed in Part 1 Page 15 and 16.

6. Although most of their experimental observations are consistent with the conclusion that WIPI4 elimination triggers ferroptosis via increasing mitochondrial PE contents, all the evidence are correlative and there lacks a causative experiment directly confirming that increased mitochondrial PE triggers ferroptosis (PS liposome experiment is for this purpose, but the experiment may cause too many changes to the cell and as such one cannot attribute the outcome only to mito PE). The authors either need to conduct such an experiment (miPISD overexpression?) or tune down their conclusion.

Response: Addressed in Part 1 Page 17.

Minor points:

Please carefully proofread the whole manuscript and the citation of both published literature and your own figures in the text. For example, (1) where are your Fig. S1n and S1o as you cited in the text? (2) There is a wealth of important papers about mitochondria and ferroptosis, and a careful and brief discussion on this can put your study into this relevant context.

Response: We have added a brief discussion about mitochondria and ferroptosis at Line 295. Sorry that we don't have enough space to do a detailed review about this topic due to the formatting limitations.

REVIEWER 3

Remarks to the Author:

Autophagy maintains neuronal homeostasis by intracellular elimination of damaged material in general, including protein aggregates and damaged mitochondrial fragments in particular, termed aggrephagy and mitophagy, respectively. Accordingly, failure of mitophagy is implicated in the etiology of neurodegeneration, e.g. due to mutations in the mitophagy signaling protein Parkin in Parkinson Disease. However, whether autophagy proteins promote mitochondrial well-being in non-autophagic contexts remains unclear. In their submitted manuscript, Zhu et al. study a model for the neurodegeneration disease BPAN that develops upon loss-of-function of WIPI4, a protein implicated in phagophore expansion and maturation, through autophagic targeting of the ER-phagophore tether and lipid transporter ATG2A.

The authors study the role of WIPI4 in BPAN by employing an in vitro model using siRNA-mediated knockdown of WIPI4 in HeLa and neuroblastoma or iPS-derived neurons and a zebrafish in vivo model. These are augmented by genetic rescue using different WIPI4 constructs or artificial gain of ATG2A function through exogenous expression of different ATG2A constructs. Coupled with autophagic inactivation by knockdown of WIPI2 or Atg7, cytotoxicity due to loss of WIPI4 is elegantly demonstrated to result from induction of ferroptosis. Furthermore, mislocalization of ATG2A from the autophagic context - due to loss of interaction with WIPI4 or upon artificial upregulation - leads to abnormal localization of ATG2A to ER-mitochondrial contact sites. This, in turn, is shown to promote autophagy-independent ferroptosis due to excessive ATG2A-mediated transport of PS from ER to mitochondria and consequent conversion to oxidation-prone mitochondrial PE.

The suggested model posits that in wildtype conditions, WIPI4 shields ATG2A from unchecked mitochondrial-associated activity that is unleashed in BPAN patients, leading to ferroptotic toxicity. Importantly, pharmaceutical inhibition of mitochondrial PS decarboxylase is shown to suppress this toxicity - hinting at a potential therapeutic intervention. The study is well-conducted and provides a new angle on the link between autophagy proteins, iron-related mitochondrial well-being and neuronal diseases. Nevertheless, important controls are warranted to establish more rigorously the model in general and the autophagy-independent activity of ATG2A therein in particular:

1. As duly mentioned, mitochondrial association ATG2A reportedly depends on TOM40 (mediated by TOM70), whereas lipid transport from the ER requires the ER-resident scramblase TMEM41B. Loss-of-function of each should therefore be shown to suppress siWIPI4/GFP-ATG2A-mediated cytotoxicity. In contrast, exogenous expression of the LIR-deficient ATG2A (mLIR, PMID: 32009292) and ATG9A-inert ATG2A (Δ 237-431, PMID: 31412244) should be shown to be at least as toxic as WT ATG2A in a manner that is suppressed by ferroptosis inhibitors - in presence or absence of WIPI4.

Response: Addressed in Part 1 Page 10.

2. The lack of toxicity upon knockdown of WIPI2 or ATG7 alone and the lack of suppression of siWIPI4-mediated toxicity are taken as evidence for autophagy-independence toxicity (Fig.

S2C). However, as these knockdowns are visibly only partial (Fig. S2A, B), at least one complete knockout of a core autophagy gene other than WIPI4 and ATG2A/B is required.

Response: Addressed in Part 1 Page 1.

3. A role for mitophagy in siWIPI4/GFP-ATG2A-mediated ferroptosis should be specifically ruled out by downregulation of implicated factors, for example, Parkin/PINK1, Nix, Mfn2/DRP1.

Response: Addressed in Part 1 Page 2.

4. WIPI2 and ATG7 act after ATG2A and WIPI4 have already been associated with the nascent phagophore. The upstream autophagy signaling protein FIP200, nucleation factor ATG9A and autophagic PI3-kinase protein ATG14 should also be shown to bear no effect on siWIPI4 and ATG2A-mediated ferroptotic toxicity. This should be correlated with the degree of ER-mito contact localization of GFP-ATG2A - as well as endogenous ATG2A if possible.

Response: Addressed in Part 1 Page 3.

5. In case the elimination of upstream proteins (FIP200, ATG9A, ATG14) is found to induce ferroptosis or exacerbate siWIPI4/GFP-ATG2A-mediated toxicity, the authors should tone down the statement on autophagy-independence mechanism, and discuss the possibility that autophagy is required to actively sequester ATG2A away from mitochondria. This should be addressed experimentally by testing a WIPI4/ATG2A-dependent alleviation of ferroptosis (or oxidation of mitochondrial PE) upon induction of autophagy (e.g. by mTOR inhibition). On the other hand, if the authors maintain that WIPI4 shields ATG2A from association with TOM40 (Fig. S5A) in an autophagy-independent manner, the effect on this association of lost FIP200/ATG9A/ATG14 should be ruled out experimentally - while maintaining the effect of WIPI4 depletion.

Response: Addressed in Part 1 Page 4.

6. Autophagic and mitophagic activities should be assayed, quantified and discussed for each genetic intervention performed throughout the study - by knockdown and/or exogenous expression. This is critical to conclude an autophagy-independent mechanism.

Response: Addressed in Part 1 Page 5.

Pre-decision discussion of reviewer comments with the authors

Dear Dr. Rubinsztein,

We now have received all reviews from the original revs on your revision "WIPI4 depletion associated with Beta-propeller protein-associated neurodegeneration causes autophagy-independent PE-dependent ferroptosis" -- we have been discussing the reviewers' comments editorially and apologize for the delay in sending our decision to you.

I am writing because two reviewers shared some questions about the revision data and persistent concerns that we found important. I am pasting their comments below. At this stage, we were hoping to ask if you would be willing to please send us responses to the comments pasted below.

****To be clear, we are not asking for a revised manuscript/new experiments/new data now. We are interested in hearing your thoughts on these points within 1-2 business days if possible - including clarifications regarding data presentation/analyses, or if you may have data that could address any of the points, or if you think you would be able to provide minimal experiments along these lines in a reasonable time frame if needed.****

This would be extremely informative to us as we continue the editorial process. We might then discuss your response with the reviewers again before reaching our decision editorially.

Please let me know if you have any questions and I look forward to hearing your thoughts on the points below. Thank you so much for your time and consideration,

Best wishes,
Melina

--

Melina Casadio, PhD
Senior Editor, NCB

REVIEWER #2

The authors have addressed most questions from the reviewers. Some minor issues about new data provided in the revision:

1. Fig. S7g: Dead cell number (Y axis) only 1-2 per well under all treatments? Is this mislabeling?
2. Fig. S2e: in BECN-KO cells, although authors claim a statistically significant increase of cell death caused by siWIPI4, visually there is no appreciable difference whatsoever caused by siWIPI4 (bar #3 vs bar #4). As this is an important claim, reproducible data of better quality is needed.

REVIEWER #3

The authors generally addressed my concerns successfully and in its present form, the study meets NCB merit.

The authors should check the data presented in figure 3D of the revised manuscript as TMEM41b shows six lanes while the other only has five.

Author responses pre-decision

1. Fig. S7g: Dead cell number (Y axis) only 1-2 per well under all treatments? Is this mislabeling?

Here we have used 1 as an arbitrary number for normalising the control data to account for variability between biological replicate experiments. The number of dead cells per well in the control ranges from 400 to 1200 and this is a function of the variability in the numbers of cells initially seeded in the wells in the different experiments. We are happy to clarify this in the figure legend.

2. Fig. S2e: in BECN-KO cells, although authors claim a statistically significant increase of cell death caused by siWIPI4, visually there is no appreciable difference whatsoever caused by siWIPI4 (bar #3 vs bar #4). As this is an important claim, reproducible data of better quality is needed.

Here again we have used 1 as an arbitrary number for normalising the control data to account for variability between biological replicate experiments. The data for lanes 3 and 4 are the data from each experiment relative to the normalisation in lane 1. The colours of the dots allow one to compare the data from the three individual experiments and it is clear that these are reproducibly higher in lane 4 vs lane 3. We used a paired t test to compare the data in lanes 3 and 4 and this is significant. We are happy to clarify this in the figure legend.

Raw data are shown in excel spreadsheet supplementary Table 5.

The authors should check the data presented in Figure 3d of the revised manuscript as TMEM41b shows six lanes while the other only has five.

We are grateful to the reviewer for picking this up. We made an error and included some additional lanes for TMEM41b.

The corrected panel is below.

Decision Letter, first revision:

Our ref: NCB-LE50815A

13th December 2023

Dear Dr. Rubinsztein,

Thank you for submitting your revised manuscript "WIPI4 depletion associated with Beta-propeller protein-associated neurodegeneration causes autophagy-independent PE-dependent ferroptosis" (NCB-LE50815A). It has now been seen by the original referees and their comments are below. The reviewers find that the paper has improved in revision, and we really appreciate your time getting back to us on the final reviewer points.

We have discussed your responses further with Rev#2, who offered more feedback below. Overall, we will leave it to you to decide how to address these remaining questions (e.g., with data presentation edits or text if you feel this is appropriate), but we will not require more experimentation.

Therefore, we'll be happy in principle to publish the manuscript in Nature Cell Biology, pending minor revisions to satisfy the referees' final requests and to comply with our editorial and formatting guidelines.

We will now begin to perform detailed checks on your paper and will send you a checklist detailing our editorial and formatting requirements in about 1-2 weeks. Please do not upload the final materials and make any revisions until you receive this additional information from us.

Thank you again for your efforts in revision and for your interest in Nature Cell Biology. Please do not hesitate to contact me if you have any questions.

Sincerely,

Melina

Melina Casadio, PhD
Senior Editor, Nature Cell Biology
ORCID ID: <https://orcid.org/0000-0003-2389-2243>

Reviewer #1 (Remarks to the Author):

The authors have addressed my concerns.

Reviewer #2 (Remarks to the Author):

The authors have addressed most questions from the reviewers. Some minor issues about new data provided in the revision:

1. Fig. S7g: Dead cell number (Y axis) only 1-2 per well under all treatments? Is this mislabeling?
2. Fig. S2e: in BECN-KO cells, although authors claim a statistically significant increase of cell death caused by siWIPI4, visually there is no appreciable difference whatsoever caused by siWIPI4 (bar #3 vs bar #4). As this is an important claim, reproducible data of better quality is needed.

ADDITIONAL POINTS FROM REV#2 ABOUT THE AUTHORS' RESPONSES TO THE ABOVE:

1. Understood. Suggestion: since Y axis in Fig. S7g is arbitrary number instead of absolute number as used for most other plots in the paper, then it should be labeled as "...ratio" instead of "... number" to avoid confusion. The preferred way is to use absolute cell number instead of arbitrary number for the Y-axis in Fig. S7g, so that the labeling of this plot is consistent with other similar experiments presented in the paper.
2. Fig. S2e: this reviewer understood that the authors claim here a statistically significant difference, based on statistical calculation. However, as WIPI4 is the centerpiece of the paper, it is strange that they show here a "statistically significant" difference caused by siWIPI4 but that cannot even be easily appreciated visually (or did I get it wrong?). With that said, it is up to the authors to decide if they want to use current data or obtain new data that is reproducible and with more profound difference caused by siWIPI4. Using the current very modest data may make readers wonder how much relevance their WIPI4 mechanism has.

Reviewer #3 (Remarks to the Author):

The authors generally addressed my concerns successfully and in its present form, the study meets NCB merit.

The authors should check the data presented in figure 3D of the revised manuscript as TMEM41b shows six lanes while the other only has five.

Author Rebuttal, first revision:

Reviewer #2 (Remarks to the Author):

The authors have addressed most questions from the reviewers. Some minor issues about new data provided in the revision:

1. Fig. S7g: Dead cell number (Y axis) only 1-2 per well under all treatments? Is this mislabeling?
2. Fig. S2e: in BECN-KO cells, although authors claim a statistically significant increase of cell

death caused by siWIPI4, visually there is no appreciable difference whatsoever caused by siWIPI4 (bar #3 vs bar #4). As this is an important claim, reproducible data of better quality is needed.

ADDITIONAL POINTS FROM REV#2 ABOUT THE AUTHORS' RESPONSES TO THE ABOVE:

1. Understood. Suggestion: since Y axis in Fig. S7g is arbitrary number instead of absolute number as used for most other plots in the paper, then it should be labeled as "...ratio" instead of "... number" to avoid confusion. The preferred way is to use absolute cell number instead of arbitrary number for the Y-axis in Fig. S7g, so that the labeling of this plot is consistent with other similar experiments presented in the paper.

2. Fig. S2e: this reviewer understood that the authors claim here a statistically significant difference, based on statistical calculation. However, as WIPI4 is the centerpiece of the paper, it is strange that they show here a "statistically significant" difference caused by siWIPI4 but that cannot even be easily appreciated visually (or did I get it wrong?). With that said, it is up to the authors to decide if they want to use current data or obtain new data that is reproducible and with more profound difference caused by siWIPI4. Using the current very modest data may make readers wonder how much relevance their WIPI4 mechanism has.

RESPONSES:

Extended data Fig 7g – We have now amended the figure legend and the y axis label to indicate that we are showing the relative numbers of dead cells per well, in line with what the reviewer suggested. This should avoid confusion.

Fig S2e – Ultimately, we felt that it was easiest just to perform another biological repeat. We have also represented the data using a one-sample t test – now it is very clear that the effects of WIPI4 depletion in the BECN-KO cells are quantitatively similar to those in the wild-type cells (and the effects are clearly statistically significant).

Reviewer #3 (Remarks to the Author):

The authors generally addressed my concerns successfully and in its present form, the study meets NCB merit.

The authors should check the data presented in figure 3D of the revised manuscript as TMEM41b shows six lanes while the other only has five.

We are grateful to the reviewer for picking this up. We made an error and included some additional lanes for TMEM41b – this has been corrected.

Decision Letter, second revision:

Our ref: NCB-LE50815A

21st December 2023

Dear Dr. Rubinsztein,

Thank you for your patience as we've prepared the guidelines for final submission of your Nature Cell Biology manuscript, "WIPI4 depletion associated with Beta-propeller protein-associated neurodegeneration causes autophagy-independent PE-dependent ferroptosis" (NCB-LE50815A). Please carefully follow the step-by-step instructions provided in the attached file, and add a response in each row of the table to indicate the changes that you have made. Please also check and comment on any additional marked-up edits we have proposed within the text. Ensuring that each point is addressed will help to ensure that your revised manuscript can be swiftly handed over to our production team.

In recognition of the time and expertise our reviewers provide to Nature Cell Biology's editorial process, we would like to formally acknowledge their contribution to the external peer review of your manuscript entitled "WIPI4 depletion associated with Beta-propeller protein-associated neurodegeneration causes autophagy-independent PE-dependent ferroptosis". For those reviewers who give their assent, we will be publishing their names alongside the published article.

Nature Cell Biology offers a Transparent Peer Review option for new original research manuscripts submitted after December 1st, 2019. As part of this initiative, we encourage our authors to support

increased transparency into the peer review process by agreeing to have the reviewer comments, author rebuttal letters, and editorial decision letters published as a Supplementary item. When you submit your final files please clearly state in your cover letter whether or not you would like to participate in this initiative. Please note that failure to state your preference will result in delays in accepting your manuscript for publication.

Cover suggestions

COVER ARTWORK: We welcome submissions of artwork for consideration for our cover. For more information, please see our guide for cover artwork.

Nature Cell Biology has now transitioned to a unified Rights Collection system which will allow our Author Services team to quickly and easily collect the rights and permissions required to publish your work. Approximately 10 days after your paper is formally accepted, you will receive an email in providing you with a link to complete the grant of rights. If your paper is eligible for Open Access, our Author Services team will also be in touch regarding any additional information that may be required to arrange payment for your article.

Please note that *Nature Cell Biology* is a Transformative Journal (TJ). Authors may publish their research with us through the traditional subscription access route or make their paper immediately open access through payment of an article-processing charge (APC). Authors will not be required to make a final decision about access to their article until it has been accepted. Find out more about Transformative Journals

Please use the following link for uploading these materials:
[Redacted]

Best regards,

Kendra Donahue
Staff
Nature Cell Biology

On behalf of

Melina Casadio, PhD
Senior Editor, Nature Cell Biology
ORCID ID: <https://orcid.org/0000-0003-2389-2243>

Reviewer #1:

Remarks to the Author:

The authors have addressed my concerns.

Reviewer #2:

Remarks to the Author:

The authors have addressed most questions from the reviewers. Some minor issues about new data provided in the revision:

1. Fig. S7g: Dead cell number (Y axis) only 1-2 per well under all treatments? Is this mislabeling?
2. Fig. S2e: in BECN-KO cells, although authors claim a statistically significant increase of cell death caused by siWIPI4, visually there is no appreciable difference whatsoever caused by siWIPI4 (bar #3 vs bar #4). As this is an important claim, reproducible data of better quality is needed.

ADDITIONAL POINTS FROM REV#2 ABOUT THE AUTHORS' RESPONSES TO THE ABOVE:

1. Understood. Suggestion: since Y axis in Fig. S7g is arbitrary number instead of absolute number as used for most other plots in the paper, then it should be labeled as "...ratio" instead of "... number" to avoid confusion. The preferred way is to use absolute cell number instead of arbitrary number for the Y-axis in Fig. S7g, so that the labeling of this plot is consistent with other similar experiments presented in the paper.
2. Fig. S2e: this reviewer understood that the authors claim here a statistically significant difference, based on statistical calculation. However, as WIPI4 is the centerpiece of the paper, it is strange that they show here a "statistically significant" difference caused by siWIPI4 but that cannot even be easily appreciated visually (or did I get it wrong?). With that said, it is up to the authors to decide if they want to use current data or obtain new data that is reproducible and with more profound difference caused by siWIPI4. Using the current very modest data may make readers wonder how much relevance their WIPI4 mechanism has.

Reviewer #3:

Remarks to the Author:

The authors generally addressed my concerns successfully and in its present form, the study meets NCB merit.

The authors should check the data presented in figure 3D of the revised manuscript as TMEM41b shows six lanes while the other only has five.

Final Decision Letter:

Dear Dr Rubinsztein,

I am pleased to inform you that your manuscript, "Loss of WIPI4 in neurodegeneration causes autophagy-independent ferroptosis", has now been accepted for publication in Nature Cell Biology.

Please note that *Nature Cell Biology* is a Transformative Journal (TJ). Authors may publish their research with us through the traditional subscription access route or make their paper immediately open access through payment of an article-processing charge (APC). Authors will not be required to make a final decision about access to their article until it has been accepted. Find out more about Transformative Journals

If you have not already done so, we strongly recommend that you upload the step-by-step protocols used in this manuscript to the Protocol Exchange (www.nature.com/protocolexchange), an open online resource established by Nature Protocols that allows researchers to share their detailed experimental know-how. All uploaded protocols are made freely available, assigned DOIs for ease of citation and are fully searchable through nature.com. Protocols and Nature Portfolio journal papers in which they are used can be linked to one another, and this link is clearly and prominently visible in the online versions of both papers. Authors who performed the specific experiments can act as primary authors for the Protocol as they will be best placed to share the methodology details, but the Corresponding Author of the present research paper should be included as one of the authors. By uploading your Protocols to Protocol Exchange, you are enabling researchers to more readily reproduce or adapt the methodology you use, as well as increasing the visibility of your protocols and papers. You can also establish a dedicated page to collect your lab Protocols. Further information can be found at www.nature.com/protocolexchange/about

With kind regards,

Melina Casadio, PhD
Senior Editor, Nature Cell Biology
ORCID ID: <https://orcid.org/0000-0003-2389-2243>